# What Matters for Adversarial Imitation Learning?

**Manu Orsini**,* **Anton Raichuk**,* **Léonard Hussenot**\*†,
**Damien Vincent, Robert Dadashi, Sertan Girgin,**
**Matthieu Geist, Olivier Bachem, Olivier Pietquin, Marcin Andrychowicz**‡

Google Research, Brain Team

## Abstract

Adversarial imitation learning has become a popular framework for imitation in continuous control. Over the years, several variations of its components were proposed to enhance the performance of the learned policies as well as the sample complexity of the algorithm. In practice, these choices are rarely tested all together in rigorous empirical studies. It is therefore difficult to discuss and understand what choices, among the high-level algorithmic options as well as low-level implementation details, matter. To tackle this issue, we implement more than 50 of these choices in a generic adversarial imitation learning framework and investigate their impacts in a large-scale study (>500k trained agents) with both synthetic and human-generated demonstrations. We analyze the key results and highlight the most surprising findings.

## 1   Introduction

Reinforcement Learning (RL) has shown its ability to perform complex tasks in contexts where clear reward functions can be set-up (e.g. +1 for winning a chess game) [15, 37, 40, 43] but for many real-world applications, designing a correct reward function is either tedious or impossible [20], while demonstrating a correct behavior is often easy and cheap. Therefore, imitation learning (IL, [4, 7]) might be the key to unlock the resolution of more complex tasks, such as autonomous driving, for which reward functions are much harder to design.

The simplest approach to IL is Behavioral Cloning (BC, [2]) which uses supervised learning to predict the expert's action for any given state. However, BC is often unreliable as prediction errors compound in the course of an episode. Adversarial Imitation Learning (AIL, [14]) aims to remedy this using inspiration from Generative Adversarial Networks (GANs, [9]) and Inverse RL [3, 5, 6]: the policy is trained to generate trajectories that are indistinguishable from the expert's ones. As in GANs, this is formalized as a two-player game where a discriminator is co-trained to distinguish between the policy and expert trajectories (or states). See App. C for a brief introduction to AIL.

A myriad of improvements over the original AIL algorithm were proposed over the years [17, 31, 41, 44, 46], from changing the discriminator's loss function [17] to switching from on-policy to off-policy agents [31]. However, their relative performance is rarely studied in a controlled setting, and never these changes have never been compared simultaneously. The performance of these high-level choices may also depend on low-level implementation details which might be silenced in the original publications [19, 29, 36, 42], as well as the hyperparameters (HPs) used. Thus, assessing whether the proposed changes are the reason for the presented improvements becomes extremely

---

*Equal contribution.

†Univ. de Lille, CNRS, Inria Scool, UMR 9189 CRIStAL.

‡Corresponding author. E-mail: `marcina@google.com`.

35th Conference on Neural Information Processing Systems (NeurIPS 2021).

difficult. This lack of proper comparisons slows down the overall research in imitation learning and the industrial applicability of these methods.

We investigate such high- and low-level choices in depth and study their impact on the algorithm performance. Hence, as **our key contributions**, we (**1**) implement a highly-configurable generic AIL algorithm, with various axes of variation (>50 HPs), including 4 different RL algorithms and 7 regularization schemes for the discriminator, (**2**) conduct a large-scale (>500k trained agents) experimental study on 10 continuous-control tasks and (**3**) analyze the experimental results to provide practical insights and recommendations for designing novel and using existing AIL algorithms. We release this generic AIL agent, implemented in JAX [25] as part of the Acme [49] framework: `https://github.com/deepmind/acme/blob/master/acme/agents/jax/ail` .

**Most surprising finding #1: regularizers.**   While many of our findings confirm common practices in AIL research, some of them are surprising or even contradict prior work. In particular, we find that standard regularizers from Supervised Learning — dropout [10] and weight decay [1] often perform similarly to the regularizers designed specifically for adversarial learning like gradient penalty [18]. Moreover, for easier environments (which were often the only ones used in prior work), we find that it is possible to achieve excellent results without using any explicit discriminator regularization.

**Most surprising finding #2: human demonstrations.**   Not only does the performance of AIL heavily depend on whether the demonstrations were collected from a human operator or generated by an RL algorithm, but the relative performance of algorithmic choices also depends on the demonstration source. Our results suggest that artificial demonstrations are not a good proxy for human data and that the very common practice of evaluating IL algorithms only with synthetic demonstrations may lead to algorithms which perform poorly in the more realistic scenarios with human demonstrations.

## 2   Experimental design

**Environments.**   We focus on continuous-control tasks as robotics appears as one of the main potential applications of IL and a vast majority of the IL literature thus focuses on it. In particular, we run experiments with five widely used environments from OpenAI Gym [13]: `HalfCheetah-v2`, `Hopper-v2`, `Walker2d-v2`, `Ant-v2`, and `Humanoid-v2` and three manipulation environments from Adroit [21]: `pen-v0`, `door-v0`, and `hammer-v0`. The Adroit tasks consist in aligning a pen with a target orientation, opening a door and hammering a nail with a 5-fingered hand.

**Demonstrations.**   For the Gym tasks, we generate demonstrations with a SAC [28] agent trained on the environment reward. For the Adroit environments, we use the "expert" and "human" datasets from D4RL [45], which are, respectively, generated by an RL agent and collected from a human operator. As far as we know, our work is the first to solve these tasks with human datasets in the imitation setup (most of the prior work concentrated on Offline RL). For all environments, we use 11 demonstration trajectories. Following prior work [14, 31, 46], we subsample expert demonstrations by only using every $20^{\text{th}}$ state-action pair to make the tasks harder.

**Adversarial Imitation Learning algorithms.**   We researched prior work on AIL algorithms and made a list of commonly used design decisions like policy objectives or discriminator regularization techniques. We also included a number of natural options which we have not encountered in literature (e.g. dropout [10] in the discriminator or clipping rewards bigger than a threshold). All choices are listed and explained in App. D. Then, we implemented a single highly-configurable AIL agent which exposes all these choices as configuration options in the Acme framework [49] using JAX [25] for automatic differentiation and Flax [47] for neural networks computation. The configuration space is so wide that it covers the whole family of AIL algorithms, in particular, it mostly covers the setups from AIRL [17] and DAC [31]. We plan to open source the agent implementation.

**Experimental design.**   We created a large HP sweep (57 HPs swept, >120k agents trained) in which each HP is sampled uniformly at random from a discrete set and independently from the other HPs. We manually ensured that the sampling ranges of all HPs are appropriate and cover the optimal values. Then, we analyzed the results of this initial experiment (called *wide*, detailed description and results in App. G), removed clearly suboptimal options and ran another experiment with the pruned sampling

ranges (called *main*, 43 HPs swept, >250k agents trained, detailed description and results in App. H). The latter experiment serves as the basis for most of the conclusions drawn in this paper but we also run a few additional experiments to investigate some additional questions (App. I and App. J).

This pruning of the HP space guarantees that we draw conclusions based on training configurations which are highly competitive (training curves can be found in Fig. 24) while using a large HP sweep (including, for example, multiple different RL algorithms) ensures that our conclusions are robust and valid not only for a single RL algorithm and specific values of HPs, but are more generally applicable. Moreover, many choices may have strong interactions with other related choices, for example we find a surprisingly strong interaction between the discriminator regularization scheme and the discriminator learning rate (Sec. 4). This means that such choices need to be tuned together (as it is the case in our study) and experiments where only a single choice is varied but the interacting choices are kept fixed may lead to misleading conclusions.

**Performance measure.** For each HP configuration and each of the 10 environment-dataset pairs we train a policy and evaluate it 10 times through the training by running it for 50 episodes and computing the average undiscounted return using the environment reward. We then average these scores to obtain a single performance score which approximates the area under the learning curve. This ensures we assign higher scores to HP configurations that learn quickly.

**Analysis.** We consider two different analyses for each choice[4]:

***Conditional 95th percentile***: For each potential value of that choice (e.g., `RL Algorithm = PPO`), we look at the performance distribution of sampled configurations with that value. We report the 95th percentile of the performance as well as error bars based on bootstrapping.[5] This corresponds to an estimate of the performance one can expect if all other choices were tuned with random search and a limited budget of roughly 13 HP configurations[6]. All scores are normalized so that $0$ corresponds to a random policy and $1$ to the expert performance (expert scores can be found in App. F).

***Distribution of choice within top 5% configurations.*** We further consider for each choice the distribution of values among the top 5% HP configurations. In particular, we measure the ratio of the frequency of the given value in the top 5% of HP configurations with the best performance to the frequency of this value among all HP configurations. If certain values are over-represented in the top models (ratio higher than 1), this indicates that the specific choice is important for good performance.

We release the raw results of our experiments[7] along with a Notebook allowing to load and study it[8].

## 3 What matters for the agent training?

**Summary of key findings.** The AIRL reward function perform best for synthetic demonstrations while $-\ln(1 - D)$ is better for human demonstrations. Using explicit absorbing state is crucial in environments with variable length episodes. Observation normalization strongly affects the performance. Using an off-policy RL algorithm is necessary for good sample complexity while replaying expert data and pretraining with BC improves the performance only slightly.

**Implicit reward function.** In this section, we investigate choices related to agent training with AIL, the most salient of which is probably the choice of the implicit reward function. Let $D(s, a)$ be the probability of classifying the given state-action pair as *expert* by the discriminator. In particular, we run experiments with the following reward functions: $r(s, a) = -\log(1 - D(s, a))$ (used in the original GAIL paper [14]), $r(s, a) = \log D(s, a) - \log(1 - D(s, a))$ (called the AIRL reward [17]), $r(s, a) = \log D(s, a)$ (a natural choice we have not encountered in literature), and the FAIRL [46] reward function $r(s, a) = -h(s, a) \cdot e^{h(s,a)}$, where $h(s, a)$ is the discriminator logit. It can be

---

[4]This analysis is based on a similar type of study focused on on-policy RL algorithms [42].

[5]We compute each metric 20 times based on a randomly selected half of all training runs, and then report the mean of these 20 measurements while the error bars show mean-std and mean+std.

[6]The probability that all 13 configurations score worse than the 95th percentile is equal $0.95^{13} \approx 50\%$.

[7]`https://storage.googleapis.com/what-matters-in-imitation-learning/data.json`

[8]`https://storage.googleapis.com/what-matters-in-imitation-learning/analysis_colab.ipynb`

shown that, under the assumption that all episodes have the same length, maximizing these reward functions corresponds to the minimization of different divergences between the marginal state-action distribution of the expert and the policy. See [46] for an in-depth discussion on this topic. We also consider clipping the rewards with absolute values bigger than a threshold which is a HP.

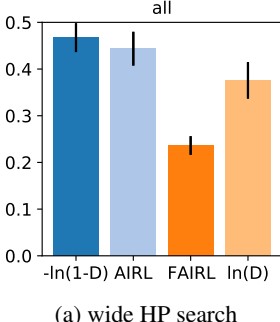
(a) wide HP search

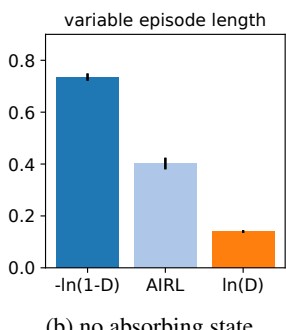
(b) no absorbing state

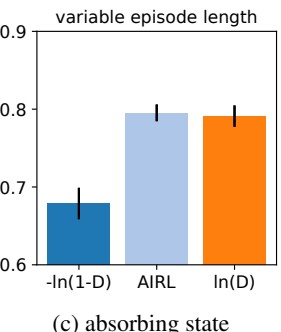
(c) absorbing state

Figure 1: Comparison of different reward functions. The bars show the 95th percentile across HPs sampling of the *average* policy performance during training. Plot (a) shows the results averaged across all 10 tasks. Plots (b) and (c) show the performance on the subset of environments with variable length episodes when the absorbing state is disabled (b) or enabled (c). See Fig. 12 and Fig. 72 for the individual results in all environments.

The FAIRL reward performed much worse than all others in the initial wide experiment (Fig. 1a) and therefore was not included in our main experiment. This is mostly caused by its inferior performance with off-policy RL algorithms (Fig. 22). Moreover, reward clipping significantly helps the FAIRL reward (Fig. 23) while it does not help the other reward functions apart from some small gains for $-\ln(1-D)$ (Fig. 82). Therefore, we suspect that the poor performance of the FAIRL reward function may be caused by its exponential term which may have very high magnitudes. Moreover, the FAIRL paper [46] mentions that the FAIRL reward is more sensitive to HPs than other reward functions which could also explain its poor performance in our experiments.

Fig. 25 shows that the $\ln(D)$ reward functions performs a bit worse than the other two reward functions in the main experiment. Five out of the ten tasks used in our experiments have variable length episodes with longer episodes correlated with better behaviour (`Hopper`, `Walker2d`, `Ant`, `Humanoid`, `pen`) — on these tasks we can notice that $r(s, a) = -\ln(1 - D(s, a))$ often performs best and $r(s, a) = \ln D(s, a)$ worst. This can be explained by the fact that $-\ln(1 - D(s, a)) > 0$ and $\ln D(s, a) < 0$ which means that the former reward encourages longer episodes and the latter one shorter ones [31]. Absorbing state (described in App. D.2) is a technique introduced in the DAC paper [31] to mitigate the mentioned bias and encourage the policy to generate episodes of similar length to demonstrations. In Fig. 1b-c we show how the performance of different reward functions compares in the environments with variable length episodes depending on whether the absorbing state is used. We can notice that without the absorbing state $r(s, a) = -\ln(1 - D(s, a)) > 0$ performs much better in the environments with variable episode length which suggests that the learning is driven to a large extent by the reward bias and not actual imitation of the expert behaviour [31]. This effect disappears when the absorbing state is enabled (Fig. 1c).

Fig. 72 shows the performance of different reward functions in all environments conditioned on whether the absorbing state is used. If the absorbing state is used, the AIRL reward function performs best in all the environments with RL-generated demonstrations, and $\ln(D)$ performs only marginally worse. The $-\ln(1 - D)$ reward function underperforms on the `Humanoid` and `pen` tasks while performing best with human datasets. We provide some hypothesis for this behaviour in Sec. 5, where we discuss human demonstrations in more details.

**Observation normalization.** We consider observation normalization which is applied to the inputs of all neural networks involved in AIL (policy, critic and discriminator). The normalization aims to transform the observations so that that each observation coordinate has mean $0$ and standard deviation $1$. In particular, we consider computing the normalization statistics either using only the expert demonstrations so that the normalization is *fixed* throughout the training, or using data from the policy

being trained (called *online*). See App. D.6 for more details. Fig. 26 shows that input normalization significantly influences the performance with the effects on performance being often much larger than those of algorithmic choices like the reward function or RL algorithm used. Surprisingly, normalizing observations can either significantly improve or diminish performance and whether the fixed or online normalization performs better is also environment dependent.

**Replaying expert data.**    When demonstrations as well as external rewards are available, it is common for RL algorithms to sample batches for off-policy updates from the demonstrations in addition to the replay buffer [30, 39]. We varied the ratio of the policy to expert data being replayed but found only very minor gains (Fig. 83). Moreover, in the cases when we see some benefits, it is usually best to replay 16–64 times more policy than expert data. On some tasks (`Humanoid`) replaying even a single expert transitions every 256 agent ones significantly hurts performance. We suspect that, in contrast to RL with demonstrations, we see little benefit from replaying expert data in the setup with learned rewards because (1) replaying expert data mostly helps when the reward signal is sparse (not the case for discriminator-based rewards), and (2) discriminator may overfit to the expert demonstrations which could result in incorrectly high rewards being assigned to expert transitions.

**Pretraining with BC.**    We also experiment with pretraining a policy with Behavioral Cloning (BC, [2]) at the beginning of training. Despite starting from a much better policy than a random one, we usually observe that the policy quality deteriorates quickly at the beginning of training (see the `pen` task in Fig. 3) due to being updated using randomly initialized critic and discriminator networks, and the overall gain from pretraining is very small in most environments (Fig. 27).

**RL algorithms.**    We run experiments with four different RL algorithms, three of which are off-policy algorithms (SAC [28], TD3 [26] and D4PG [24]), as well as PPO [22] which is nearly on-policy. Fig. 7 shows that the sample complexity of PPO is significantly worse than that of the off-policy algorithms while all off-policy algorithms perform overall similarly.

**RL algorithms HPs.**    Fig. 8 shows that the discount factor is one of the most important HPs with the values of $0.97 - 0.99$ performing well on all tasks. Fig. 29 shows that in most environments it is better not to erase any data from the RL replay buffer and always sample from all the experience encountered so far. It is common in RL to use a noise-free version of the policy during evaluation and we observe that it indeed improves the performance (Fig. 30). The policy MLP size does not matter much (Figs. 31-32) while bigger critic networks perform significantly better (Figs. 9-10). Regarding activation functions, relu performs on par or better than tanh in all environments apart from `door` in which tanh is significantly better (Fig. 33). Our implementation of TD3 optionally applies gradient clipping but it does not affect the performance much (Fig. 34). D4PG can use n-step returns, this improves the performance on the Adroit tasks but hurts on the Gym suite (Fig. 35).

## 4   What matters for the discriminator training?

**Summary of key findings.**    MLP discriminators perform on par or better than AIL-specific architectures. Explicit discriminator regularization is only important in more complicated environments (`Humanoid` and harder ones). Spectral norm is overall the best regularizer but standard regularizers from supervised learning often perform on par. Optimal learning rate for the discriminator may be 2–2.5 orders of magnitude lower than the one for the RL agent.

**Discriminator input.**    In this section we look at the choices related to the discriminator training. Fig. 49 shows how the performance depends on the discriminator input. We can observe that while it is beneficial to feed actions as well as states to the discriminator, the state-only demonstrations perform almost as well. Interestingly, on the `door` task with human data, it is better to ignore the expert actions. We explore the results with human demonstrations in more depth in Sec. 5.

**Discriminator architecture.**    Regarding the discriminator network, our basic architecture is an MLP but we also consider two modifications introduced in AIRL [17]: a reward shaping term and a $\log \pi(a|s)$ logit shift which introduces a dependence on the current policy (only applicable to RL algorithms with stochastic policies, which in our case are PPO and SAC). See App. D.3 for a detailed description of these techniques. Fig. 13 shows that the logit shift significantly hurts the performance.

This is mainly due to the fact that it does not work well with SAC which is off-policy (Fig. 21). Fig. 50 shows that the shaping term does not affect the performance much. While the modifications from AIRL does not improve the sample complexity in our experiments, it is worth mentioning that they were introduced for another purpose, namely the recovery of transferable reward functions.

Regarding the size of the discriminator MLP(s), the best results on all tasks are obtained with a single hidden layer (Fig. 51), while the size of the hidden layer is of secondary importance (if it is not very small) with the exception of the tasks with human data where fewer hidden units perform significantly better (Fig. 52). All tested discriminator activation functions perform overall similarly while sigmoid performs best with human demonstrations (Fig. 53).

**Discriminator training.**    Fig. 54 shows that it is best to use as large as possible replay buffers for sampling negative examples (i.e. agent transitions). Prior work has claimed the initialization of the last *policy* layer can significantly influence the performance in RL [42], thus we tried initializing the last *discriminator* layer with smaller weights but it does not make much difference (Fig 55).

**Discriminator regularization.**    An overfit or too accurate discriminator can make agent's training challenging, and therefore it is common to use additional regularization techniques when training the AIL discriminator (or GANs in general). We run experiments with a number of regularizers commonly used with AIL, namely Gradient Penalty [18] (GP, used e.g. in [31]), spectral norm [35] (e.g. in [44]), Mixup [23] (e.g. in [52]), as well as using the PUGAIL loss [41] instead of the standard cross entropy loss to train the discriminator. Apart from the above regularizers, we also run experiments with regularizers commonly used in Supervised Learning, namely dropout [10], the weight decay [1] variant from AdamW [33] as well as the entropy bonus of the discriminator output treated as a Bernoulli distribution. The detailed description of all these regularization techniques can be found in App. D.5.

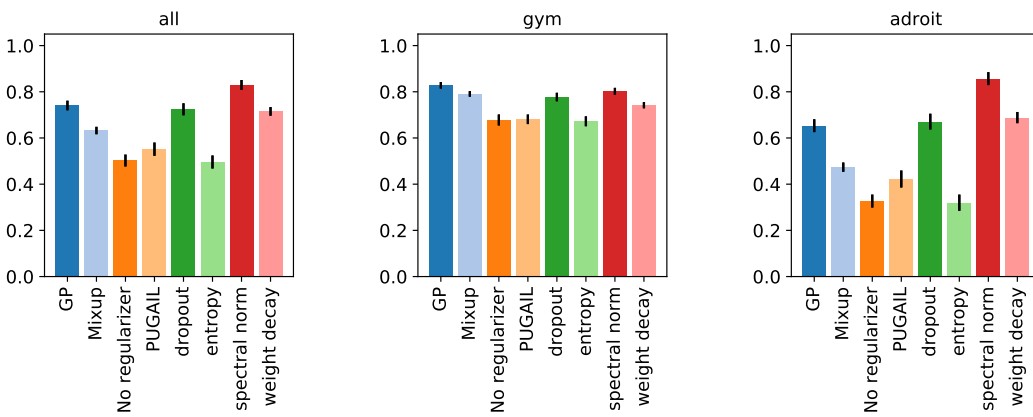

Figure 2: The 95th percentile of performance for different discriminator regularizers. The central plot shows the average performance across 5 tasks from OpenAI Gym and the right one the average performance across 5 tasks from the Adroit suite. See Fig. 56 for the plots for individual environments.

Fig. 2 shows how the performance depends on the regularizer. Spectral normalization performs overall best, while GP, dropout and weight decay all perform on par with each other and only a bit worse than spectral normalization. We find this conclusion to be quite surprising given that we have not seen dropout or weight decay being used with AIL in literature. We also notice that the regularization is generally more important on harder tasks like `Humanoid` or the tasks in the Adroit suite (Fig. 56).

Most of the regularizers investigated in this section have their own HPs and therefore the comparison of different regularizers depends on how these HPs are sampled. As we randomly sample the regularizer-specific HPs in this analysis, our approach favours regularizers that are not too sensitive to their HPs. At the same time, there might be regularizers that are sensitive to their HPs but for which good settings may be easily found. Fig. 67 shows that even if we condition on choosing the optimal HPs for each regularizer, the relative ranking of regularizers does not change.

Moreover, there might be correlations between the regularizer and other HPs, therefore their relative performance may depend on the distribution of all other HPs. In fact, we have found two such

surprising correlations. Fig. 73 shows the performance conditioned on the regularizer used *as well as* the discriminator learning rate. We notice that for PUGAIL, entropy and no regularization, the performance significantly increases for lower discriminator learning rates and the best performing discriminator learning rate ($10^{-6}$) is in fact 2-2.5 orders of magnitude lower than the best learning rate for the RL algorithm (0.0001–0.0003, Figs. 14, 38, 39, 41, 47).[9] On the other hand, the remaining regularizers are not too sensitive to the discriminator learning rate. This means that the performance gap between PUGAIL, entropy and no regularization and the other regularizers is to some degree caused by the fact that the former ones are more sensitive to the learning rate and may be smaller than suggested by Fig. 2 if we adjust for the appropriate choice of the discriminator learning rate. We can notice that PUGAIL and entropy are the only regularizers which only change the discriminator loss but do not affect the internals of the discriminator neural network. Given that they are the only two regularizers benefiting from very low discriminator learning rate, we suspect that it means that a very low learning rate can play a regularizing role in the absence of an explicit regularization inside the network.

Another surprising correlation is that in some environments, the regularizer interacts strongly with observation normalization (described App. D.6) employed on discriminator inputs (see Fig. 74 for an example on `Ant`). These two correlations highlight the difficulty of comparing regularizers, and algorithmic choices more broadly, as their performance significantly depends on the distribution of other HPs.

We also supplement our analysis by comparing the performance of different regularizers for the *best* found HPs. More precisely, we choose the best value for each HP in the main experiment (listed in App. E) and run them with different regularizers. To account for the mentioned correlations with the discriminator learning rate and observation normalization, we also include these two choices in the HP sweep and choose the best performing variant (as measure by the area under the learning curve) for each regularizer and each environment.

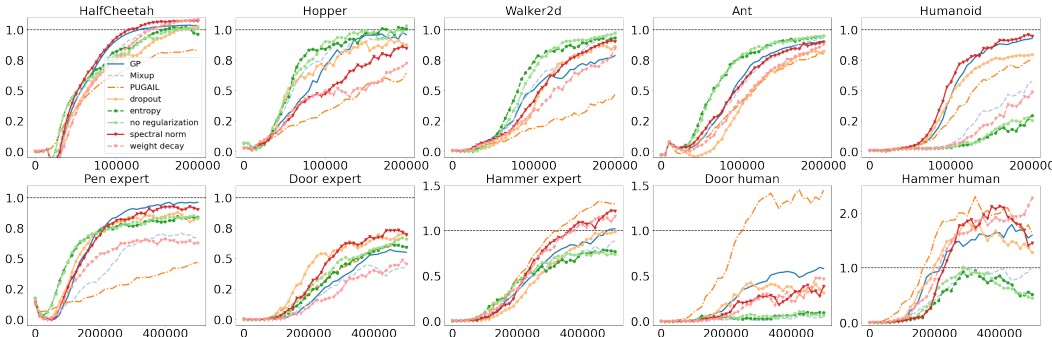

Figure 3: Learning curves for different discriminator regularizers when the other HPs are set to the best performing value across all tasks. The y-axis shows the average policy return normalized so that 0 corresponds to a random policy and 1 to the expert. See App. E for the HPs used. The plots shows the averages across 30 random seeds. Best seen in color.

While it is not guaranteed that the performance is going to be good at all because we greedily choose the best performing value for each HP and there might be some unaccounted HP correlations, we find that the performance is very competitive (Fig. 3). Notice that we use the same HPs in *all* environments and the performance can be probably improved by varying some HPs between the environments, or at least between the two environment suites.

We notice that on the four easiest tasks (`HalfCheetah`, `Hopper`, `Walker2d`, `Ant`), investigated discriminator regularizers provide no, or only minor performance improvements and excellent results can be achieved without them. On the tasks where regularization is beneficial, we usually see that there are multiple regularizers performing similarly well, with spectral normalization being one of the best regularizers in all tasks apart from the two tasks with human data where PUGAIL performs better.

**Regularizers-specific HPs.** For GP, the target gradient norm of 1 is slightly better in most environments but the value of 0 is significantly better in `hammer-human` (Fig. 57), while the penalty strength of 1 performs best overall (Fig. 58). For dropout, it is important to apply it not only to hidden layers

---

[9]The optimal learning rate for those regularizers was the smallest one included in the main experiment. We also run an additional sweep with smaller rates but found that even lower ones do not perform better (Fig. 81).

but also to inputs (Fig. 59) and the best results are obtained for 50% input dropout and 75% hidden activations dropout (Figs. 59, 60 and 67). For weight decay, the optimal decay coefficient in the AIL setup is much larger than the values typically used for Supervised Learning, the value $\lambda = 10$ performs best in our experiments (Fig. 61). For Mixup, $\alpha = 1$ outperforms the other values on almost all tested environments (Fig. 62). For PUGAIL, the unbounded version performs much better on the Adroit suite, while the bounded version is better on the Gym tasks (Fig. 63), and positive class prior of $\eta = 0.7$ performs well on most tasks (Fig. 64). For the discriminator entropy bonus, the values around 0.03 performed best overall (Fig. 65). All experiments with spectral normalization enforce the Lipschitz constant of 1 for each weight matrix.

**How to train efficiently?** So far we have analysed how HPs affect the sample complexity of AIL algorithms. For the analysis of the HPs which influence sample complexity as well as the computational cost of running an algorithm see App. A. In particular, we describe there a simple code optimization relying on processing multiple batches at once which makes training 2-3x faster in wall clock time without affecting the sample complexity (Fig. 5).

## 5 Are synthetic demonstrations a good proxy for human data?

**Summary of key findings** Human demonstrations significantly differ from synthetic ones. Learning from human demonstrations benefits more from discriminator regularization and may work better with different discriminator inputs and reward functions than RL-generated demonstrations.

Using a dataset of human demonstrations comes with a number of additional challenges. Compared to synthetic demonstrations, the human policy can be multi-modal in that for a given state different decisions might be chosen. A typical example occurs when the human demonstrator remains idle for some time (for example to think about the next action) before taking the actual relevant action: we have two modes in that state, the relevant action has a low probability while the idle action has a very high probability. The human policy might not be exactly markovian either. Those differences are significant enough that the conclusions on synthetic datasets might not hold anymore.

In this section, we focus on the Adroit `door` and `hammer` environments for which we run experiments with human as well as synthetic demonstrations. [10] Note that on top of the aforementioned challenges, the setup with the Adroit environments using human demonstrations exhibits a few additional specifics. The demonstrations were collected letting the human decide when the task is completed: said in a different way, the demonstrator is offered an additional action to jump directly to a terminal state and this action is not available to the agent imitating the expert. The end result is a dataset of demonstrations of variable length while the agent can only generate episodes consisting of exactly 200 transitions. Note that there was no time limit imposed on the demonstrator and some of the demonstrations have a length greater than 200 transitions. Getting to the exact same state distribution as the human expert may be impossible, and imitation learning algorithms may have to make some trade-offs. The additional specificity of that setup is that the reward of the environment is not exactly what the human demonstrator optimized. In the `door` environment, the reward provided by the environment is the highest when the door is *fully* opened while the human might abort the task slightly before getting the highest reward. However, overall, we consider the reward provided by the environment as a reasonable metric to assess the quality of the trained policies. Moreover, in the `hammer` environment, some demonstrations have a low return and we suspect those are not successful demonstrations.[11]

**Discriminator regularization.** When comparing the results for RL-generated (`adroit-expert`[12]) and human demonstrations (`adroit-human`) we can notice differences on a number of HPs related to the discriminator training. Human demonstrations benefit more from using discriminator regularizers (Fig. 56) and they also work better with smaller discriminator networks (Fig. 52) trained with lower learning rates (Fig. 66). The increased need for regularization suggest that it is easier to overfit to the idiosyncrasies of human demonstrations than to those of RL policies.

---

[10] For `pen`, we only use the "expert" dataset, the "human" one consists of a single (yet very long) trajectory.

[11] D4RL datasets [45] contain only the policy observations and not the simulator states and therefore it is not straightforward to visualize the demonstrations.

[12] We do not include `pen` in the `adroit-expert` plots so that both `adroit-expert` and `adroit-human` show the results averages across the `door` and `hammer` tasks and differ only in the demonstrations used.

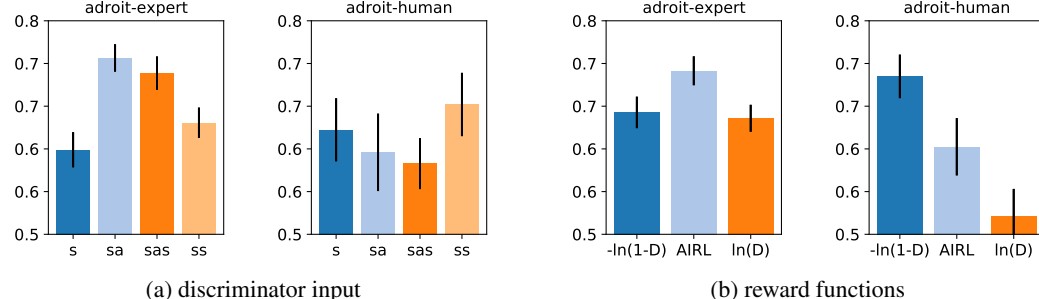

(a) discriminator input          (b) reward functions

Figure 4: Comparison of discriminator inputs (a) and reward functions (b) for environments with human demonstrations. See Fig. 49 and Fig. 25 for the individual results in all environments.

**Discriminator input.** Fig. 4a shows the performance given the discriminator input depending on the demonstration source. For most tasks with RL-generated demonstrations, feeding actions as well as states improves the performance (Fig. 49). Yet, the opposite holds when human demonstrations are used. We suspect that it might be caused by the mentioned issue with demonstrations lengths which forces the policy to repeat a similar movement but with a different speed than the demonstrator.

**Reward functions.** Finally, we look at how the relative performance of different reward functions depends on the demonstration source. Fig. 4b shows that for RL-generated demonstrations the best reward function is AIRL while $-\ln(1-D)$ performs better with human demonstrations. Under the assumption that the discriminator is optimal, these two reward functions correspond to the minimization of different divergences between the state (or state-action depending on the discriminator input) occupancy measures of the policy (denoted $\pi$) and the expert (denoted $E$).

The reward function performing best with human demonstrations ($-\ln(1-D)$) corresponds to the minimization of the Jensen-Shannon divergence (proof in [14]). Interestingly, this divergence is symmetric ($D_{\mathrm{JS}}(\pi||E) = D_{\mathrm{JS}}(E||\pi)$) and bounded ($0 \leq D_{\mathrm{JS}}(\pi||E) \leq \ln(2)$). For AIL, the symmetry means that it penalizes the policy for doing things the expert never does with exactly the same weight as for not doing some of the things the expert does while the boundedness means that the penalty for not visiting a single state is always finite. We suspect that this boundedness is beneficial for learning with human demonstrations because it may not be possible to exactly match the human distribution for the reasons explained earlier.

In contrast to Jensen-Shannon, the $D_{\mathrm{KL}}(\pi||E)$ divergence which is optimized by the AIRL reward (proof in [46]) is neither symmetric, nor bounded — it penalizes the policy much more heavily for doing the things the expert never does that for not doing all the things the expert does and the penalty for visiting a single state the expert never visits is infinite (assuming a perfect discriminator).

While it is hard to draw any general conclusions only from the two investigated environments for which we had access to human demonstrations, our analysis shows that the differences between synthetic and human-generated demonstrations can influence the relative performance of different algorithmic choices. This suggests that RL-generated data are not a good proxy for human demonstrations and that the very common practice of evaluating IL only with synthetic demonstrations may lead to algorithms which perform poorly in the more realistic scenarios with human demonstrations.

# 6  Related work

The most similar work to ours is probably [44] which compares the performance of different discriminator regularizers and concludes that gradient penalty is necessary for achieving good performance with off-policy AIL algorithms. In contrast to [44], which uses a single HP configuration, we run large-scale experiments with very wide HP sweeps which allows us to reach more robust conclusions. In particular, we are able to achieve excellent sample complexity on all the environments used in [44] without using any explicit discriminator regularizer (Fig. 3).

The methodology of our study is mostly based on [42] which analyzed the importance of different choices for on-policy actor-critic methods. Our work is also similar to other large-scale studies done in other fields of Deep Learning, e.g. model-based RL [38], GANs [34], NLP [50], disentangled representations [32] and convolution network architectures [51].

## 7    Conclusions

In this empirical study, we investigate in depth many aspects of the AIL framework including discriminator architecture, training and regularization as well as many choices related to the agent training. Our key findings can be divided into three categories: (1) Corroborating prior work, e.g. for the underlying RL problem, off-policy algorithms are more sample efficient than on-policy ones; (2) Adding nuances to previous studies, e.g. while the regularization schemes encouraging Lipschitzness improve the performance, more classical regularizers like dropout or weight decay often perform on par; (3) Raising concerns: we observe a high discrepancy between the results for RL-generated and human data. We hope this study will be helpful to anyone using or designing AIL algorithms.

Additionally we released the [1] unified AIL agent we implemented in JAX within the Acme framework as well as [2] the raw data of our experiment, along with [3] a Notebook that allows to load and study them.

[1] `https://github.com/deepmind/acme/tree/master/acme/agents/jax/ail`

[2] `https://storage.googleapis.com/what-matters-in-imitation-learning/data.json`

[3] `https://storage.googleapis.com/what-matters-in-imitation-learning/analysis_` `colab.ipynb`

## Acknowledgments

We thank Kamyar Ghasemipour for the discussions related to the FAIRL reward function and Lucas Beyer for the feedback on an earlier version of the manuscript.

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
