**Batch size and replay ratio.** One of the main factors influencing the throughput of a particular imitation algorithm is the number of times each transition is replayed on average and the batch size used.[13] See App. D.1 for the detailed description of the HPs involved. Fig. 76 shows that smaller batches perform overall better (given a fixed replay ratio) and increasing the replay ratio improves the performance, at least up to some threshold depending on the environment (Fig. 77). There is a very strong correlation between the two HPs — Fig. 80 shows that for most batch sizes, the optimal replay ratio is equal to the batch size, which corresponds to replaying exactly one batch of data per environment step. If we compare different batch sizes under the ratio of batches to environment steps fixed to one, the performance is mostly independent of the batch size (Fig. 80).

While in most of our experiment the discriminator and the RL agent are trained with exactly the same number of batches, we also tried doubling the number of discriminator batches. Fig. 78 shows that it improves the performance slightly on the Adroit suite.

**Combining multiple batches.** We also consider processing multiple batches at once for improved accelerator (GPU or TPU) utilization. In particular, we sample an $N$-times larger batch from a replay buffer, split it back into $N$ smaller/proper batches on an accelerator, and process them sequentially. In order to keep the replay ratio unaffected, we decrease the frequency of updates accordingly, e.g. instead of performing one gradient update for every environment step, we perform $N$ gradients updates every $N$ environment steps. We apply this technique to the discriminator as well as the RL agent training. The effect on the sample complexity of the algorithm can be seen in Fig. 79. There is a small negative effect for values larger or equal to 16. The effect of this parameter on the throughput of our system could be observed in Fig. 5. The value of 8 provides a good compromise: almost no noticeable sample complexity regression while decreasing the training time by 2–3 times.

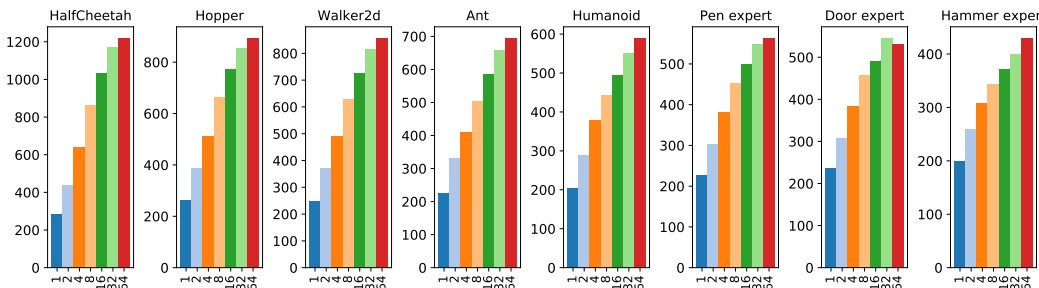

Figure 5: Training speed (in terms of environment steps per second) for combining multiple batches. The x-axis denotes the number of batches combined. Other HPs are set to the best performing value across all tasks (listed in App. E). The plots shows the averages across 10 random seeds.

# B Reinforcement Learning Background

We consider the standard reinforcement learning formalism consisting of an agent interacting with an environment. To simplify the exposition we assume in this section that the environment is fully observable. An environment is described by a set of states $\mathcal{S}$, a set of actions $\mathcal{A}$, a distribution of initial states $p(s_0)$, a reward function $r : \mathcal{S} \times \mathcal{A} \to \mathbb{R}$, transition probabilities $p(s_{t+1}|s_t, a_t)$ ($t$ is a timestep index explained later), termination probabilities $T(s_t, a_t)$ and a discount factor $\gamma \in [0, 1]$.

---

[13]We use the same batch size for the policy and actor networks while the discriminator batch size is effectively two times larger because its batches contain always `batch size (C7)` demonstration transitions and `batch size (C7)` policy transitions. The replay ratio is the same for all networks with the exception of the discriminator which can have its replay ratio doubled depending on the value of `discriminator to RL updates ratio (C44)`. See App. D.4 for details.

A policy $\pi$ is a mapping from state to a distribution over actions. Every episode starts by sampling an initial state $s_0$. At every timestep $t$ the agent produces an action based on the current state: $a_t \sim \pi(\cdot|s_t)$. In turn, the agent receives a reward $r_t = r(s_t, a_t)$ and the environment's state is updated. With probability $T(s_t, a_t)$ the episode is terminated, and otherwise the new environments state $s_{t+1}$ is sampled from $p(\cdot|s_t, a_t)$. The discounted sum of future rewards, also referred to as the *return*, is defined as $R_t = \sum_{i=t}^{\infty} \gamma^{i-t} r_i$. The agent's goal is to find the policy $\pi$ which maximizes the expected return $\mathbb{E}_{\pi}[R_0|s_0]$, where the expectation is taken over the initial state distribution, the policy, and environment transitions accordingly to the dynamics specified above.

## C   Adversarial Imitation Learning Background

See App. B for a very brief introduction to RL and the notation used in this section.

Drawing inspiration from Inverse Reinforcement Learning [5, 8] and Generative Adversarial Networks (GANs, [9]), adversarial imitation learning [14] aims at learning a behavior similar to that of the expert given a set of expert demonstrations $\mathcal{D}_{expert}$ and the ability to interact with the environment.

To do so, the agent with policy $\pi$ is initialized randomly and interacts with the environment. A discriminator network $D$ is trained to distinguish between samples coming from the agent $(s_t, a_t, s_{t+1}) \sim \mathcal{D}_{\pi}$ and samples coming from the expert dataset $(s_t, a_t, s_{t+1}) \sim \mathcal{D}_{expert}$ with a cross-entropy loss. A reward function for the policy is then defined based on the discriminator prediction, e.g. $r(s, a) = -\ln(1 - D(s, a))$, where $D(s, a)$ denotes the probability of classifying the state-action pair as expert by the discriminator. The agent is then trained with an RL algorithm to maximize this reward and thus fool the discriminator. As in GANs, the training of the discriminator and that of the agent (here playing the role of the *generator*) are interleaved. Therefore, at the high level, the algorithm repeats the following steps in a loop: (1) interact with the environment using the current policy and store the experience in a replay buffer, (2) update the discriminator, (3) perform an RL update accordingly to the RL algorithm used.

## D   List of Investigated Choices

In this section we list all algorithmic choices which we consider in our experiments. See App. C for an introduction to adversarial imitation and the notation used in this section. For convenience, we mark each of the choices with a number (e.g., C8) and a fixed name (e.g. `RL Algorithm (C8)`) that can be easily used to find a description of the choice in this section.

### D.1   Reinforcement Learning algorithms

In all experiments we use MLPs for the policy and critic/value networks and sample the following HPs controlling the networks architectures: `policy MLP depth (C1)` (the number of *hidden* layers), `policy MLP width (C2)`, `critic MLP depth (C3)`, `critic MLP width (C4)`, `RL activation (C5)`, as well as `discount` $\gamma$ `(C6)` and `batch size (C7)`. All networks are optimized with the Adam [11] optimizer.

We sample `RL Algorithm (C8)` from the following options:

**Proximal Policy Optimization (PPO, [22])**   For PPO, `batch size (C7)` denotes the number of experience fragments, each of consisting `PPO unroll length (C9)` transitions, collected in each policy update step. In each policy update step, we perform `PPO number of epochs (C10)` passes over the gathered data when in each pass the data is split into `PPO number of minibatches (C11)` minibatches. We use the PPO loss with the clipping threshold set by `PPO clipping` $\epsilon$ `(C12)` and add an entropy loss with the coefficient specified by `PPO entropy cost (C13)`. We also sample `PPO learning rate (C14)`, and the GAE [12] returns mixing coefficient `GAE` $\lambda$ `(C15)`.

**Soft Actor Critic (SAC, [28])**   We use a version of SAC with a policy entropy constraint [27]. In particular, we choose `SAC entropy per dimension (C16)` and that set the entropy constraint so that the policy entropy is not lower than the number of action dimensions times this value. We also sweep `SAC learning rate (C17)` and the target network polyak averaging coefficient `SAC polyak` $\tau$ `(C18)` (the target network is updates after each minibatch).

**Twin Delayed Deep Deterministic Policy Gradient (TD3, [26])** For TD3, we sweep `TD3 policy learning rate (C19)` and `TD3 critic learning rate (C20)` separately, as well as sample `behavioral policy noise (C21)`. Following the original publication, we update the actor only using every other minibatch while the critic networks uses all minibatches. The target network is updated after every minibatch with the polyak coefficient fixed to $0.005$. Following DAC [31], we clip actor gradients with magnitudes bigger than `TD3 gradient clipping (C22)`.

**Distributed Distributional Deterministic Policy Gradients (D4PG, [24])** This algorithm is similar to TD3 but uses a distributional C51-style critic [16] outputting distributions over `number of atoms (C23)` atoms spaced equally between `-VMax (C24)` and `VMax (C24)` as well as `N-step returns (C25)` returns. In contrast to the original D4PG [24], we use a single actor and do not use prioritized replay. The target network is fully updated every 100 training batches. As usual, we also sweep `D4PG learning rate (C26)`.

Moreover, for off-policy algorithm (SAC, TD3 and D4PG) we sample `replay ratio (C27)` which denotes the average number of times each transition is replayed. This is achieved in the following way — if `replay ratio (C27)` $\geq$ `batch size (C7)` than we replay `replay ratio (C27)` / `batch size (C7)` batches (each with `batch size (C7)` transitions) after every environment step. If `batch size (C7)` > `replay ratio (C27)`, we replay a single batch every `batch size (C7)` / `replay ratio (C27)` transitions. The transitions for replay are sampled uniformly from a FIFO replay buffer of size `RL replay buffer size (C28)` and we start training whenever we have at least 10k transition in the buffer.

For the RL algorithms which train stochastic policies (PPO and SAC) we use a Gaussian distribution followed by tanh to squash actions into the $[-1, 1]$ range.[14] More precisely, the policy network output is split into two parts — $\mu$ and $\rho$, and the action distribution used during training is $\tanh(\mathcal{N}(\mu, \mathrm{softplus}(\rho) + 0.001))$. For policy evaluation, we choose `evaluation behavior policy type (C29)` from the following options:

- *stochastic*: sample from the distribution (same as behavioral policy used during training),
- *mode*: use the mode of the Gaussian instead of sampling,
- *average*: sample five action from the distribution and take the average of them.

### D.2 Imitation-specific changes to RL

**Reward function** Let $D$ denote the probability that a state-action pair $(s, a)$ is classified as *expert* by the discriminator while $h$ is the discriminator logit, i.e. $D = \sigma(h)$ where $\sigma$ denotes the sigmoid function. Depending on the value of `reward function (C30)` we use one of the following reward functions (for completeness we write the formulas as a function of $D$ as well as $h$):

- $r(s, a) = -\ln(1 - D) = \mathrm{softplus}(h)$ (used in the original GAIL paper[15] [14]),
- $r(s, a) = \ln D - \ln(1 - D) = h$ (introduced in AIRL [17]).
- $r(s, a) = \ln D = -\mathrm{softplus}(-h)$,
- $r(s, a) = -he^h$ (introduced in FAIRL [46]).

We also clip rewards with the absolute values higher than `max reward magnitude (C31)`.

**Absorbing state** We optionally (if `absorbing state (C32)=True`) apply the absorbing state technique from DAC [31]. This technique encourages the agent to generate episodes of similar length to the ones of the expert. In particular, the demonstration and agent episodes are processed in the following way: for each terminal transition, we replace it with a non-terminal transition to a special absorbing state[16] and also add a transition from the absorbing state to itself with a zero action.

---

[14]The action coordinates are scaled to $[-1, 1]$ regardless of the RL algorithm used.

[15]The GAIL paper uses the inverse convention in which $D$ denotes the probability as being classified as *non-expert*.

[16]In practice, this is done by adding a special bit to every observation which is set to zero for normal observations and one for the absorbing state. The remaining bits of the absorbing state are all zeros.

**Replaying demonstrations** For off-policy RL algorithms, we optionally (if `policy-to-expert replay ratio (C33)` $\neq \infty$) sample batches for RL training not only from the replay buffer, but also from the demonstrations. In particular, the ratio of policy to expert data in each minibatch is equal to `policy-to-expert replay ratio (C33)`.

**Initialization with behavior cloning** We optionally (if `BC pretraining (C34)=True`) pre-train the policy network offline at the beginning of training using Behavior Cloning [2]. In particular, we perform 100k gradient steps with Adam on the MSE loss, using learning rate $10^{-4}$ and batch size 256.

### D.3 Discriminator parameterization

Depending on the value of `discriminator input (C35)`, the discriminator is fed single states, state-action pairs, state-state pairs or state-action-state tuples.

Our basic discriminator architecture is an MLP with `discriminator MLP depth (C36)` hidden layers, each of size `discriminator MLP width (C37)` with the activation function specified by `discriminator activation (C38)`. Its output is interpreted as the logit of the probability of being classified as expert, i.e. for a state-action-state tuple $(s, a, s')$ we have $D(s, a, s') = \sigma(f(s, a, s'))$, where $D$ is the probability of classifying the tuple $(s, a, s')$ as expert, $\sigma$ denotes the sigmoid function, and $f$ is a learnable function represented as an MLP.

We also consider two modifications introduced in the AIRL [17] paper. The first one (enabled if `reward shaping (C39)=True`) adds a reward shaping term where the $f$ function is parameterized in the following way: $f(s, a, s') = g(s, a, s') + \gamma h(s') - h(s)$ where $g$ and $h$ are MLPs parameterized as described above, and $\gamma$ is the RL discount factor.[17] The second modification (enabled if `subtract log-pi (C40)=True`) parameterizes the discriminator as $D(s, a, s') = \frac{\exp(f(s,a,s'))}{\exp(f(s,a,s'))+\pi(a|s)}$, where $\pi$ is the current agent policy. It can be easy shown that it is equivalent to $D(s, a, s') = \sigma((f(s, a, s') - \log \pi(a|s))$ so this just shifts the logits by $\log \pi(a|s)$.

### D.4 Discriminator training

All discriminator weight matrices use the `lecun_uniform` initializer from JAX [25]. The last discriminator layer initialization is additionally multiplied by `discriminator last layer init scale (C41)`.

The discriminator is trained with the Adam [11] optimizer, the learning rate specified by `discriminator learning rate (C42)` and the cross-entropy loss. Each data batch contains exactly `batch size (C7)` expert transitions and `batch size (C7)` policy transitions. The policy transitions are sampled uniformly from a FIFO replay buffer of size `discriminator replay buffer size (C43)`.

We perform `discriminator to RL updates ratio (C44)` discriminator gradient steps for each RL gradient step. More precisely, after each environment step, we compute the number of RL gradient steps as described in App. D.1, and perform `discriminator to RL updates ratio (C44)` that many discriminator gradient steps *before* performing the RL update.

### D.5 Discriminator regularization

Depending on the value of `discriminator regularizer (C45)`, we optionally apply one of the following regularizers to the discriminator:

**Gradient Penalty (GP, [18])** Gradient penalty is parameterized with `gradient penalty k (C46)` and `gradient penalty λ (C47)`. This regularizer adds an extra term in the discriminator loss that encourages the discriminator gradient to be close to $k$ on a convex combination of positive (expert) and negative (policy) data. In particular, for an expert data $x \sim \mathcal{D}_{expert}$ and policy data $\widetilde{x} \sim \mathcal{D}_{\pi}$, the gradient penalty is defined as $\lambda(||\nabla_{\hat{x}} D(\hat{x})||_2 - k)^2$, where $\hat{x}$ is a convex combination of $x$ and $\widetilde{x}$, i.e. $\hat{x} := \epsilon x + (1 - \epsilon)\widetilde{x}$ and $\epsilon$ follows a uniform distribution: $\epsilon \sim U[0, 1]$.

---

[17]The inputs fed to $g$ are specified by `discriminator input (C35)`.

In practice, $k$ is usually chosen to be 0 (penalty for high gradients) or 1 (penalty for gradients with norms far from 1). Our gradient penalty implementation uses the gradient of the discriminator logit instead of the classification probability.

**Spectral normalization [35]**  Spectral normalization guarantees that the discriminator is 1-Lipschitz: $|D(x_2) - D(x_1)| \leq ||x_2 - x_1||$. It does so by dividing each dense layer matrix by its highest eigenvalue which can be efficiently computed with the power iteration method. See [35] for details.

**Mixup [23]**  Mixup is parameterized with `mixup` $\alpha$ `(C48)` and relies on training the discriminator on a convex combination of positive (expert) and negative (policy) data. With expert data $x \sim \mathcal{D}_{expert}$ and policy data $\widetilde{x} \sim \mathcal{D}_{\pi}$, let $\epsilon$ follow a Beta distribution: $\epsilon \sim Beta(\alpha, \alpha)$. Instead of training the discriminator on $x$ and $\widetilde{x}$ separately, we only train it on the convex combination of them $\hat{x} := \epsilon x + (1 - \epsilon)\widetilde{x}$ with the label being the convex combinations of the labels, i.e. expert with probability $\epsilon$ and non-expert with probability $1 - \epsilon$, so that the loss is $-\epsilon \ln D(\hat{x}) - (1 - \epsilon) \ln(1 - D(\hat{x}))$.

**Positive Unlabeled GAIL (PUGAIL, [41])**  Normally the discriminator is trained under the assumption that expert trajectories are positive examples and policy trajectories are negative examples. The PUGAIL loss assumes instead that policy trajectories are a mix of positive and negative examples.

With `PUGAIL` $\eta$ `(C49)` denoting the assumed proportion of positive samples in the policy data and `PUGAIL` $\beta$ `(C50)` being a clipping threshold, the discriminator is trained with the following loss:

$$\eta\hat{\mathbb{E}}_{x \sim \mathcal{D}_{expert}}[-\ln(D(x))] + \max\left(-\beta, \hat{\mathbb{E}}_{x \sim \mathcal{D}_{\pi}}[-\ln(1 - D(x))] - \eta\hat{\mathbb{E}}_{x \sim \mathcal{D}_{expert}}[-\ln(1 - D(x))]\right).$$

**Dropout [10]**  We apply dropout to the hidden layers (`dropout hidden rate (C51)`) as well as inputs (`dropout input rate (C52)`). See [10] for the description of dropout.

**Weight decay [1, 33]**  Weight decay is parameterized with a parameter controlling its strength `weight decay` $\lambda$ `(C53)`. Normally, weight decay is applied by adding a sum of the squares of the network parameters to the loss. However, this may interact negatively with an adaptive gradient optimizer like Adam [11] unless the optimizer is modified appropriately [33]. In our experiments, we use a version of Adam with weight decay called AdamW [33] from the Optax library [48]. See [33] for the details.

**Entropy bonus**  Similarly to entropy bonus in RL, we also experiment with adding to the discriminator loss a term proportional to the entropy of the discriminator output treated as a Bernoulli distribution: $\lambda\left(D \ln D + (1 - D) \ln(1 - D)\right)$ where `entropy` $\lambda$ `(C54)` is a HP.

### D.6  Observation normalization

We optionally apply input normalization (choice `observation normalization (C55)`) which transforms linearly the observations to all neural networks (in the RL algorithm as well as the discriminator) so that each coordinate has approximately mean equal zero and standard deviation equal one. This is done by subtracting from each observation $\mu$ and dividing by $\max(\rho, 0.001)$, where $\mu$ and $\rho$ are the empirical mean and standard deviation of either all demonstrations (we call it *fixed* normalization because it does not change during training) or the empirical mean and standard deviation of all the observations encountered by the policy being trained so far (called *online* because it changes during training).

### D.7  Combining multiple batches

We consider processing multiple batches at once for improved accelerator (GPU or TPU) utilization (choice `number of combined batches (C56)`). In particular, we sample an $N$-times larger batch from a replay buffer, split it back into $N$ smaller/proper batches on an accelerator, and process them sequentially. In order to keep the replay ratio unaffected, we decrease the frequency of updates accordingly, e.g. instead of performing one gradient update for every environment step, we perform $N$ gradients updates every $N$ environment steps. We apply this technique to the discriminator as well as the RL agent training.

# E   Best hyperparameter values

Table 1 shows the best value found for each HP in the main experiment. See App. H for the full experimental report. The sample complexity can be slightly improved by decreasing `number of combined batches (C56)` and increasing `discriminator to RL updates ratio (C44)`. We used the suboptimal values from Table 1 because they give a good trade-off between sample complexity and runtime. `discriminator learning rate (C42)` equal $10^{-6}$ is better when PUGAIL, entropy or no discriminator regularizer is used, and $3 \cdot 10^{-5}$ is better otherwise. The performance of observation normalization schemes depends heavily on the environment *and* discriminator regularization used. For completeness, we present the best HPs for all discriminator regularizers.

Table 1: Best hyperparameter configuration.

| Choice | Name | Best value |
|---|---|---|
| C1 | policy MLP depth | 2 |
| C2 | policy MLP width | 256 |
| C3 | critic MLP depth | 2 |
| C4 | critic MLP width | 256 |
| C5 | RL activation | ReLu |
| C6 | discount $\gamma$ | 0.97 |
| C7 | batch size | 256 |
| C8 | RL Algorithm | SAC |
| C16 | SAC entropy per dimension | $-0.5$ |
| C17 | SAC learning rate | $3 \cdot 10^{-4}$ |
| C18 | SAC polyak $\tau$ | 0.01 |
| C27 | replay ratio | 256 |
| C28 | RL replay buffer size | $3 \cdot 10^6$ |
| C29 | evaluation behavior policy type | mode |
| C30 | reward function | AIRL |
| C31 | max reward magnitude | $\infty$ |
| C32 | absorbing state | True |
| C33 | policy-to-expert replay ratio | $\infty$ |
| C34 | BC pretraining | True |
| C35 | discriminator input | $(s, a)$ |
| C36 | discriminator MLP depth | 1 |
| C37 | discriminator MLP width | 64 |
| C38 | discriminator activation | ReLu |
| C39 | reward shaping | False |
| C40 | subtract log-pi | False |
| C41 | discriminator last layer init scale | 1 |
| C42 | discriminator learning rate | $10^{-6}$ or $3 \cdot 10^{-5}$ |
| C43 | discriminator replay buffer size | $3 \cdot 10^6$ |
| C44 | discriminator to RL updates ratio | 1 |
| C45 | discriminator regularizer | spectral normalization |
| C46 | gradient penalty k | 0 |
| C47 | gradient penalty $\lambda$ | 1 |
| C48 | mixup $\alpha$ | 1 |
| C49 | PUGAIL $\eta$ | 0.7 |
| C50 | PUGAIL $\beta$ | $\infty$ |
| C51 | dropout hidden rate | 75% |
| C52 | dropout input rate | 50% |
| C53 | weight decay $\lambda$ | 10 |
| C54 | entropy $\lambda$ | 0.03 |
| C55 | observation normalization | depends on the environment |
| C56 | number of combined batches | 8 |

# F Expert and random policy scores

Table 2: Expert and random policy scores used to normalize the performance for all tasks.

| Task | Random policy score | Expert score |
|---|---|---|
| HalfCheetah-v2 | -282 | 8770 |
| Hopper-v2 | 18 | 2798 |
| Walker2d-v2 | 1.6 | 4118 |
| Ant-v2 | -59 | 5637 |
| Humanoid-v2 | 123 | 9115 |
| pen-expert-v0 | 94 | 3078 |
| door-expert-v0 | -56 | 2882 |
| door-human-v0 | -56 | 796 |
| hammer-expert-v0 | -274 | 12794 |
| hammer-human-v0 | -274 | 3071 |

# G Experiment wide

## G.1 Design

For each of the 10 tasks, we sampled 12083 choice configurations where we sampled the following choices independently and uniformly from the following ranges:

- `RL Algorithm (C8)`: {d4pg, ppo, sac, td3}
  - For the case "RL Algorithm (C8) = sac", we further sampled the sub-choices:
    * `SAC learning rate (C17)`: {0.0001, 0.0003, 0.001}
    * `SAC entropy per dimension (C16)`: {-2.0, -1.0, -0.5, 0.0}
    * `SAC polyak` $\tau$ `(C18)`: {0.001, 0.003, 0.01, 0.03}
    * `subtract log-pi (C40)`: {False, True}
    * `batch size (C7)`: {256.0}
  - For the case "RL Algorithm (C8) = d4pg", we further sampled the sub-choices:
    * `D4PG learning rate (C26)`: {3e-05, 0.0001, 0.0003}
    * `behavioral policy noise (C21)`: {0.1, 0.2, 0.3, 0.5}
    * `VMax (C24)`: {150.0, 750.0, 1500.0}
    * `number of atoms (C23)`: {51.0, 101.0, 201.0, 401.0}
    * `N-step returns (C25)`: {1.0, 3.0, 5.0}
    * `batch size (C7)`: {256.0}
  - For the case "RL Algorithm (C8) = td3", we further sampled the sub-choices:
    * `TD3 policy learning rate (C19)`: {0.0001, 0.0003, 0.001}
    * `TD3 critic learning rate (C20)`: {0.0001, 0.0003, 0.001}
    * `TD3 gradient clipping (C22)`: {40.0, $\infty$}
    * `behavioral policy noise (C21)`: {0.1, 0.2, 0.3, 0.5}
    * `batch size (C7)`: {256.0}
  - For the case "RL Algorithm (C8) = ppo", we further sampled the sub-choices:
    * `PPO learning rate (C14)`: {3e-05, 0.0001, 0.0003}
    * `PPO number of epochs (C10)`: {2.0, 5.0, 10.0, 20.0}
    * `PPO entropy cost (C13)`: {0.0, 0.001, 0.003, 0.01, 0.03, 0.1}
    * `PPO number of minibatches (C11)`: {8.0, 16.0, 32.0, 64.0}
    * `PPO unroll length (C9)`: {4.0, 8.0, 16.0, 32.0}
    * `PPO clipping` $\epsilon$ `(C12)`: {0.1, 0.2, 0.3}
    * `GAE` $\lambda$ `(C15)`: {0.8, 0.9, 0.95, 0.99}
    * `subtract log-pi (C40)`: {False, True}
    * `batch size (C7)`: {64.0, 128.0, 256.0}
- `RL replay buffer size (C28)`: {300000.0, 1000000.0, 3000000.0}
- `policy MLP depth (C1)`: {1, 2, 3}
- `policy MLP width (C2)`: {64, 128, 256, 512}
- `critic MLP depth (C3)`: {1, 2, 3}
- `critic MLP width (C4)`: {64, 128, 256, 512}
- `RL activation (C5)`: {relu, tanh}
- `discount` $\gamma$ `(C6)`: {0.9, 0.97, 0.99, 0.997}
- `BC pretraining (C34)`: {False, True}
- `absorbing state (C32)`: {False, True}
- `discriminator replay buffer size (C43)`: {300000, 1000000, 3000000}
- `reward shaping (C39)`: {False, True}
- `discriminator input (C35)`: {s, sa, sas, ss}
- `discriminator MLP depth (C36)`: {1, 2, 3}
- `discriminator MLP width (C37)`: {16, 32, 64, 128, 256, 512}

- `discriminator activation (C38)`: {elu, leaky_relu, relu, sigmoid, swish, tanh}
- `discriminator last layer init scale (C41)`: {0.001, 1.0}
- `discriminator regularizer (C45)`: {GP, Mixup, No regularizer, PUGAIL, dropout, entropy, spectral norm, weight decay}
    - For the case "`discriminator regularizer (C45)` = GP", we further sampled the sub-choices:
        * `gradient penalty` $\lambda$ `(C47)`: {0.1, 1.0, 10.0}
        * `gradient penalty k (C46)`: {0.0, 1.0}
    - For the case "`discriminator regularizer (C45)` = Mixup", we further sampled the sub-choices:
        * `mixup` $\alpha$ `(C48)`: {0.1, 0.4, 1.0}
    - For the case "`discriminator regularizer (C45)` = PUGAIL", we further sampled the sub-choices:
        * `PUGAIL` $\eta$ `(C49)`: {0.25, 0.5, 0.7}
        * `PUGAIL` $\beta$ `(C50)`: {0.0, 0.7, $\infty$}
    - For the case "`discriminator regularizer (C45)` = entropy", we further sampled the sub-choices:
        * `entropy` $\lambda$ `(C54)`: {0.0003, 0.001, 0.003, 0.01, 0.03, 0.1, 0.3}
    - For the case "`discriminator regularizer (C45)` = weight decay", we further sampled the sub-choices:
        * `weight decay` $\lambda$ `(C53)`: {0.3, 1.0, 3.0, 10.0, 30.0}
    - For the case "`discriminator regularizer (C45)` = dropout", we further sampled the sub-choices:
        * `dropout input rate (C52)`: {0.0, 0.25, 0.5, 0.75}
        * `dropout hidden rate (C51)`: {0.25, 0.5, 0.75}
- `observation normalization (C55)`: {fixed, none}
- `evaluation behavior policy type (C29)`: {average, mode, stochastic}
- `discriminator learning rate (C42)`: {1e-06, 3e-06, 1e-05, 3e-05, 0.0001, 0.0003}
- `max reward magnitude (C31)`: {0.5, 1.0, 2.0, 5.0, 10.0, 50.0, $\infty$}
- `reward function (C30)`: {-ln(1-D), AIRL, FAIRL, ln(D)}
- `replay ratio (C27)`: {256}
- `discriminator to RL updates ratio (C44)`: {1}
- `number of combined batches (C56)`: {8}

## G.2  Results

For each of the sampled choice configurations we compute the performance metric as described in Section 2. We report aggregate statistics of the experiment in Tables 3–6 as well as training curves in Figure 6. We further provide per-choice analyses in Figures 7-20.

Table 3: Quantiles of the *final* agent performance across HP configurations for OpenAI Gym tasks.

|      | Ant  | HalfCheetah | Hopper | Humanoid | Walker2d |
|------|------|-------------|--------|----------|----------|
| 90%  | 0.18 | 0.80        | 0.99   | 0.06     | 0.56     |
| 95%  | 0.56 | 0.98        | 1.15   | 0.30     | 0.85     |
| 99%  | 0.92 | 1.10        | 1.20   | 0.79     | 0.99     |
| Max  | 1.10 | 1.39        | 1.32   | 1.02     | 1.06     |

Table 4: Quantiles of the *final* agent performance across HP configurations for Adroit tasks.

|  | Door expert | Door human | Hammer expert | Hammer human | Pen expert |
|---|---|---|---|---|---|
| 90% | 0.12 | 0.07 | 0.16 | 0.12 | 0.28 |
| 95% | 0.42 | 0.28 | 0.67 | 0.47 | 0.46 |
| 99% | 0.90 | 1.20 | 1.26 | 2.03 | 0.77 |
| Max | 1.11 | 2.82 | 1.42 | 5.39 | 1.12 |

Table 5: Quantiles of the *average* agent performance during training across HP configurations for OpenAI Gym tasks.

|  | Ant | HalfCheetah | Hopper | Humanoid | Walker2d |
|---|---|---|---|---|---|
| 90% | 0.13 | 0.54 | 0.62 | 0.05 | 0.31 |
| 95% | 0.31 | 0.66 | 0.80 | 0.20 | 0.49 |
| 99% | 0.62 | 0.85 | 0.98 | 0.49 | 0.71 |
| Max | 0.94 | 0.99 | 1.08 | 0.84 | 0.92 |

Table 6: Quantiles of the *average* agent performance during training across HP configurations for Adroit tasks.

|  | Door expert | Door human | Hammer expert | Hammer human | Pen expert |
|---|---|---|---|---|---|
| 90% | 0.11 | 0.11 | 0.11 | 0.15 | 0.21 |
| 95% | 0.27 | 0.24 | 0.34 | 0.33 | 0.35 |
| 99% | 0.55 | 0.61 | 0.78 | 0.85 | 0.59 |
| Max | 0.87 | 1.65 | 1.01 | 1.97 | 0.84 |

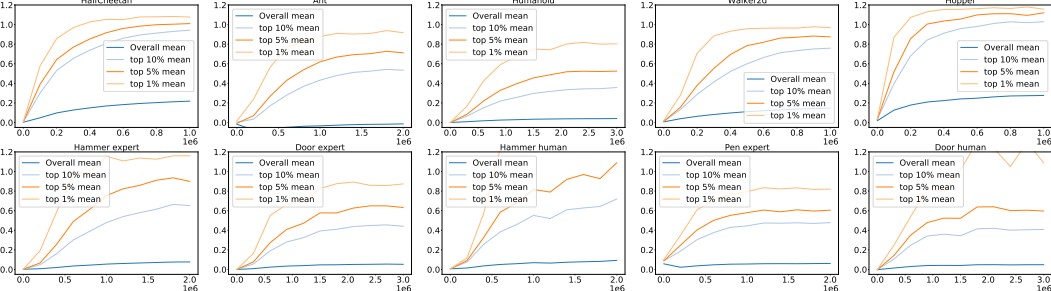

Figure 6: Training curves.

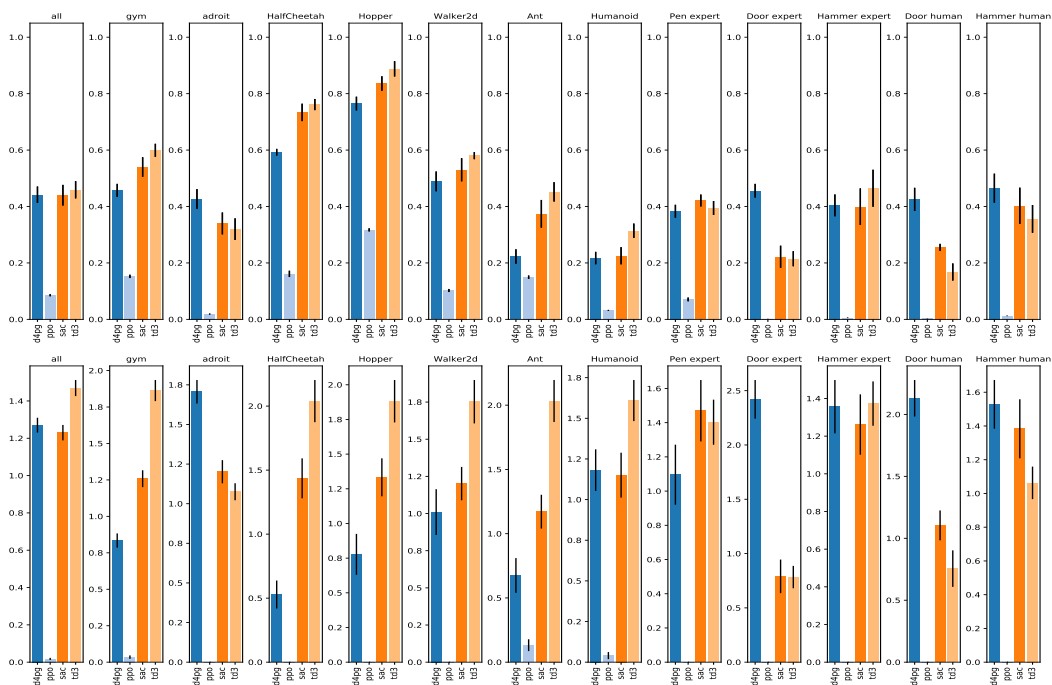

Figure 7: Analysis of choice `RL Algorithm` (C8): 95th percentile of performance scores conditioned on choice (top) and distribution of choices in top 5% of configurations (bottom).

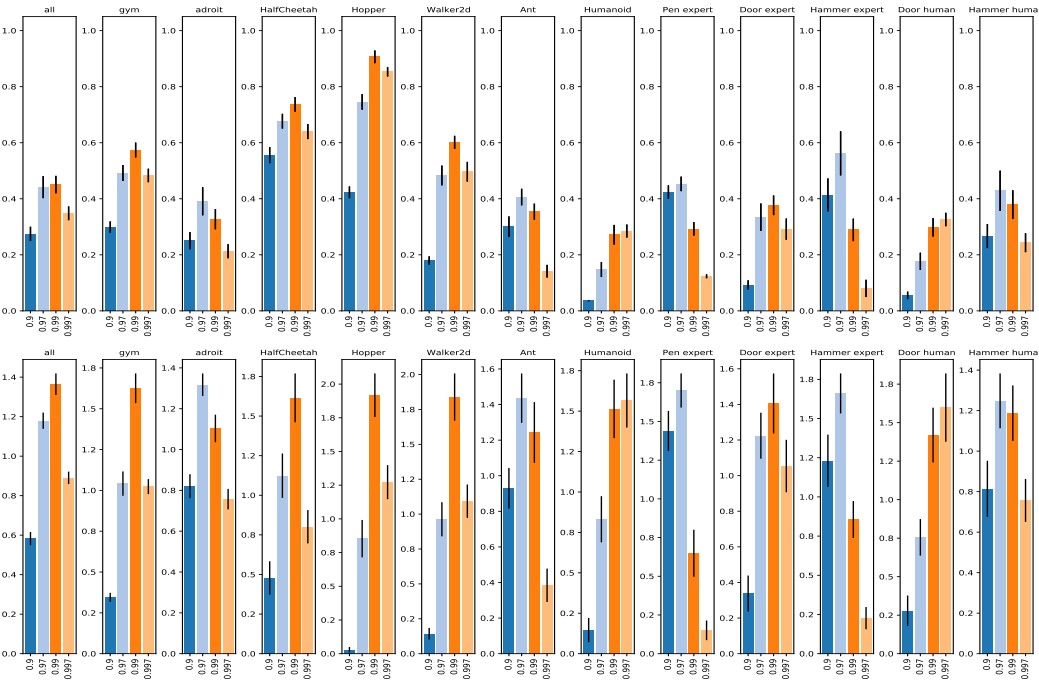

Figure 8: Analysis of choice `discount` $\gamma$ (C6): 95th percentile of performance scores conditioned on choice (top) and distribution of choices in top 5% of configurations (bottom).

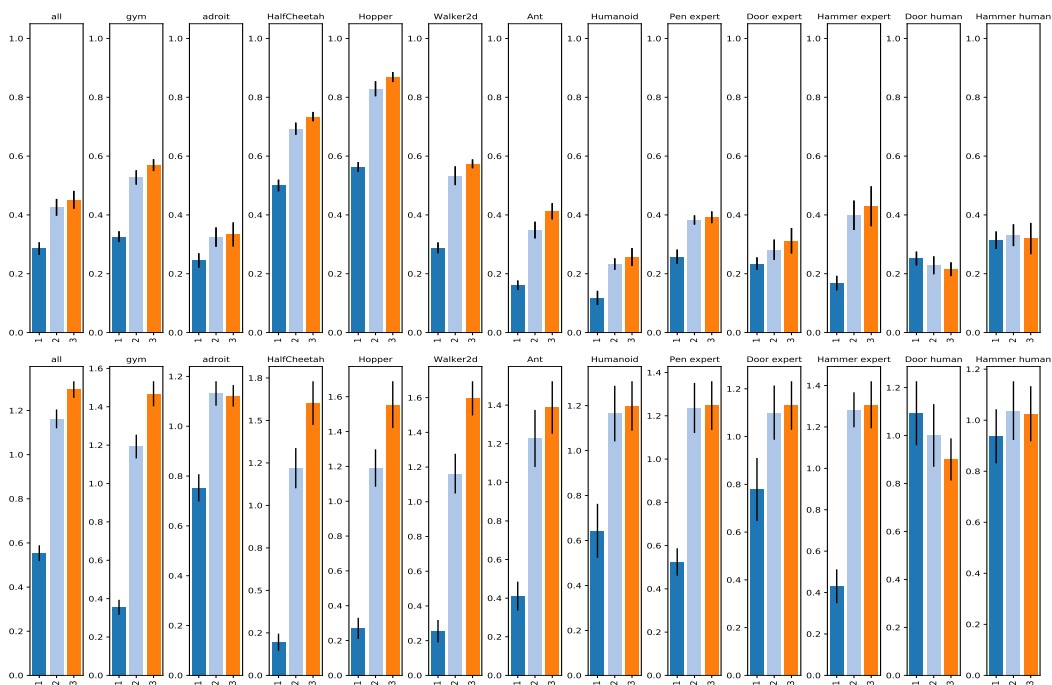

Figure 9: Analysis of choice `critic MLP depth (C3)`: 95th percentile of performance scores conditioned on choice (top) and distribution of choices in top 5% of configurations (bottom).

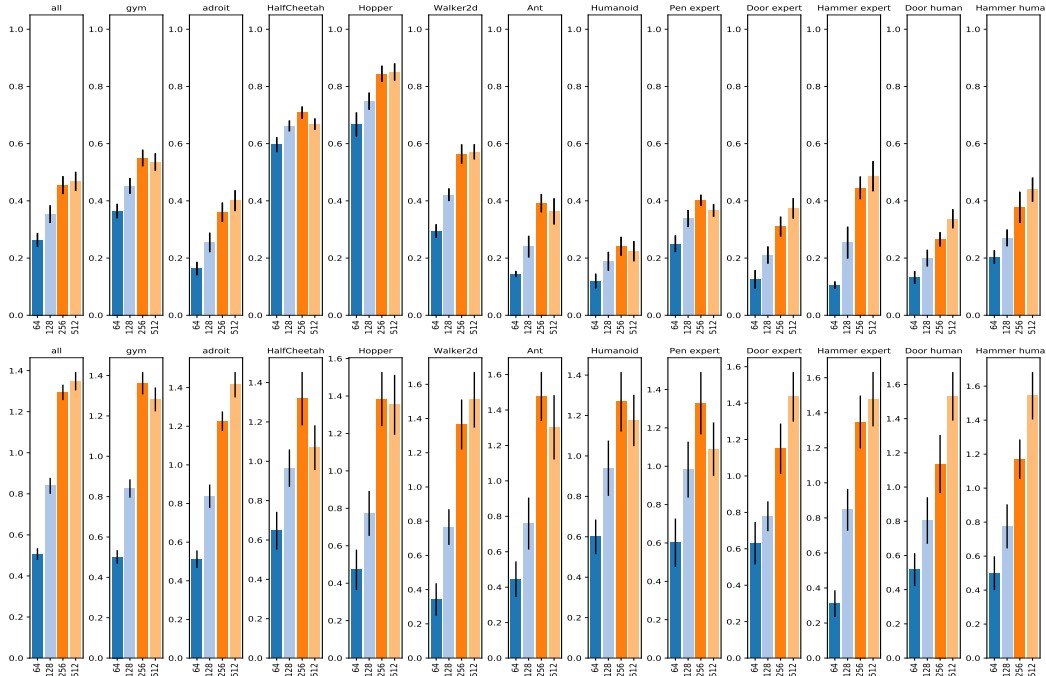

Figure 10: Analysis of choice `critic MLP width (C4)`: 95th percentile of performance scores conditioned on choice (top) and distribution of choices in top 5% of configurations (bottom).

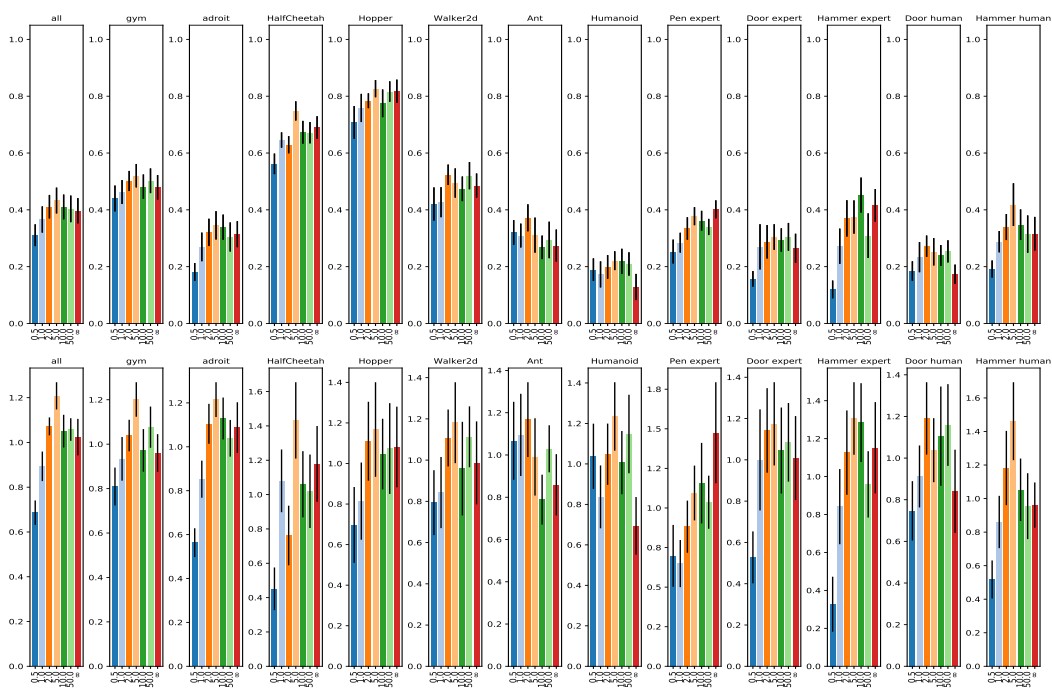

Figure 11: Analysis of choice `max reward magnitude (C31)`: 95th percentile of performance scores conditioned on choice (top) and distribution of choices in top 5% of configurations (bottom).

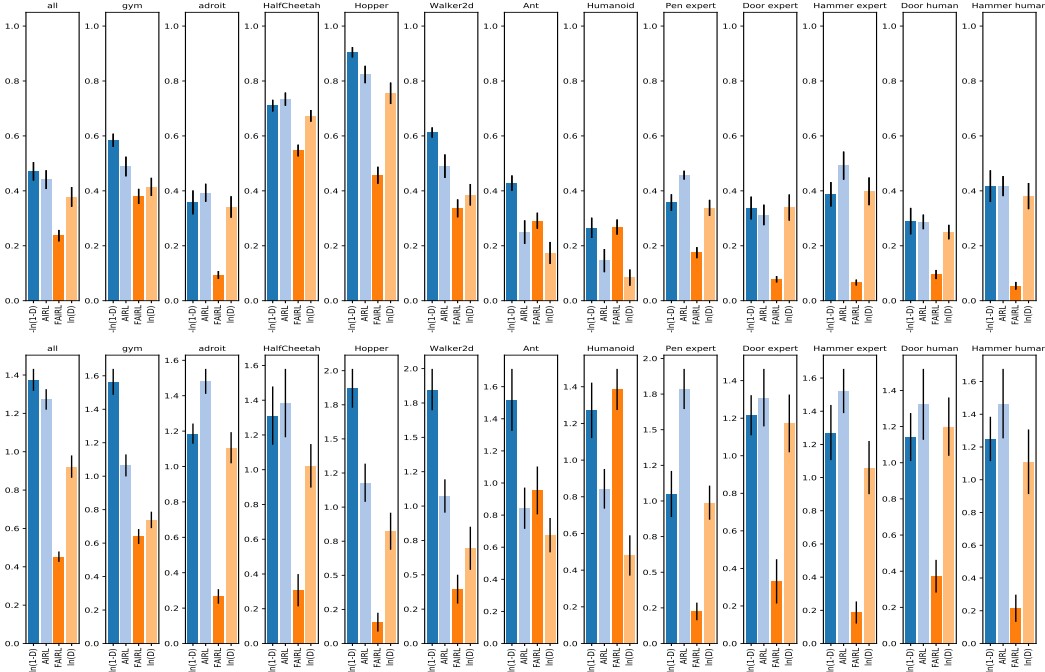

Figure 12: Analysis of choice `reward function (C30)`: 95th percentile of performance scores conditioned on choice (top) and distribution of choices in top 5% of configurations (bottom).

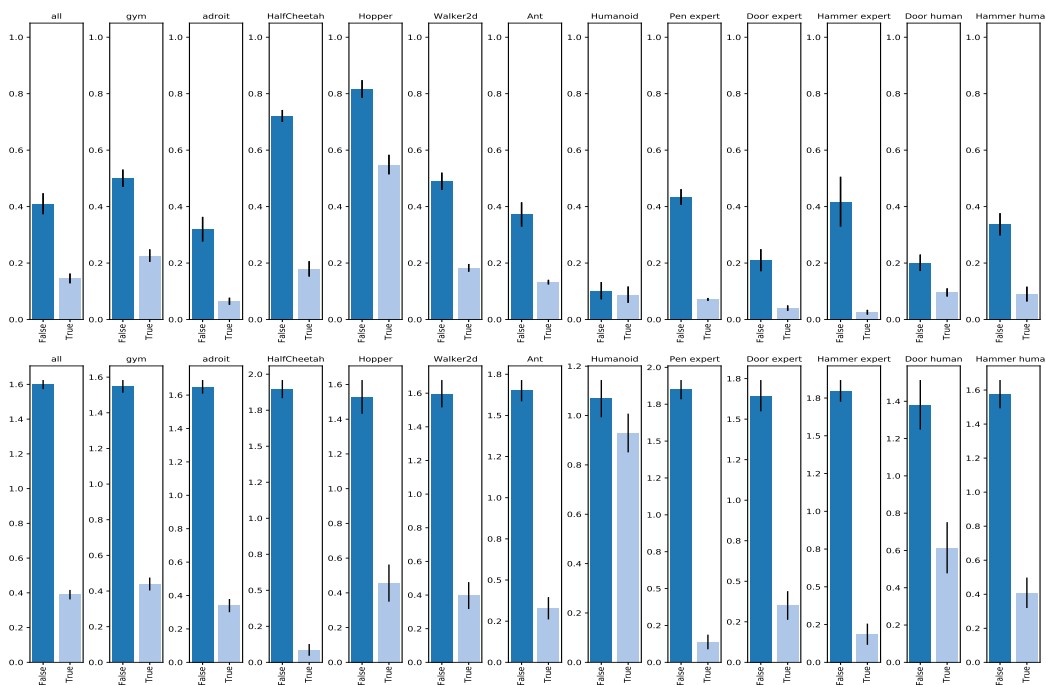

Figure 13: Analysis of choice `subtract log-pi (C40)`: 95th percentile of performance scores conditioned on choice (top) and distribution of choices in top 5% of configurations (bottom).

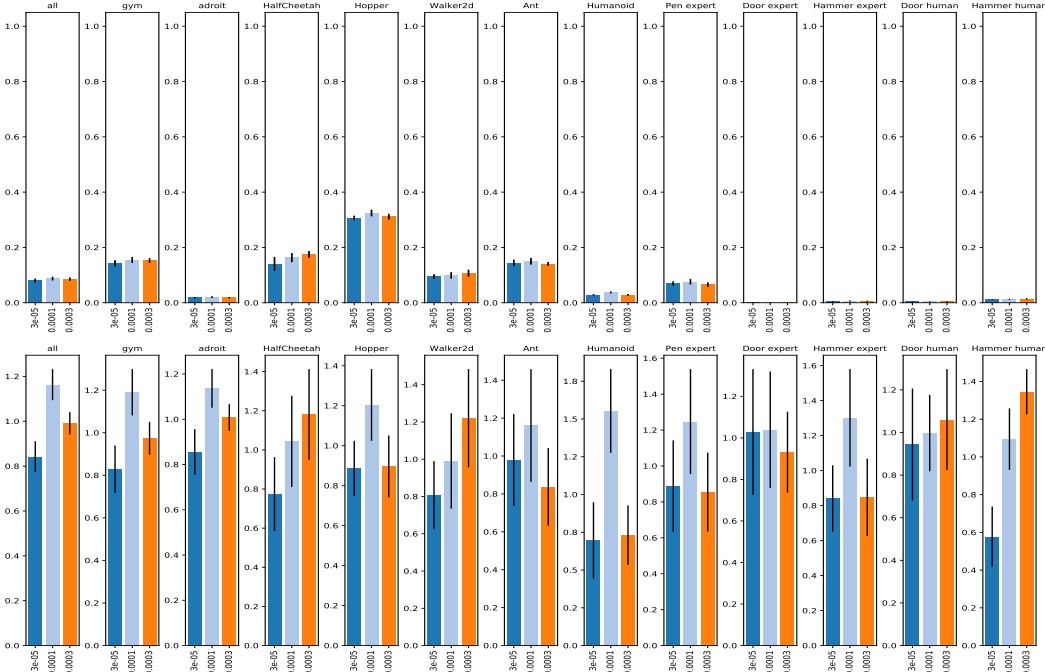

Figure 14: Analysis of choice `PPO learning rate (C14)`: 95th percentile of performance scores conditioned on choice (top) and distribution of choices in top 5% of configurations (bottom).

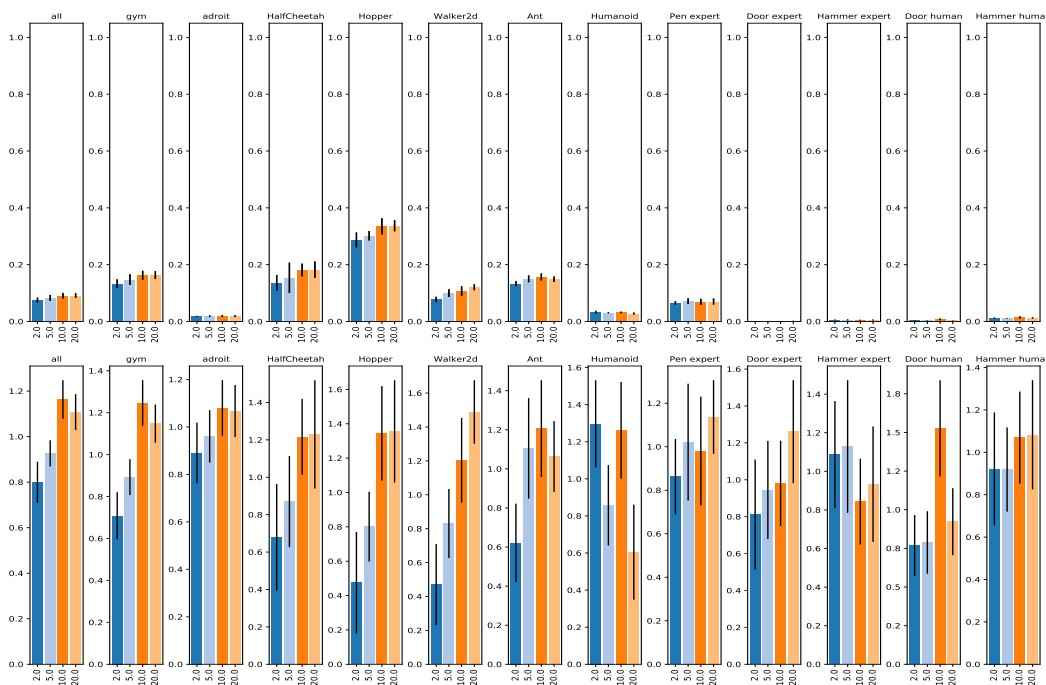

Figure 15: Analysis of choice `PPO number of epochs (C10)`: 95th percentile of performance scores conditioned on choice (top) and distribution of choices in top 5% of configurations (bottom).

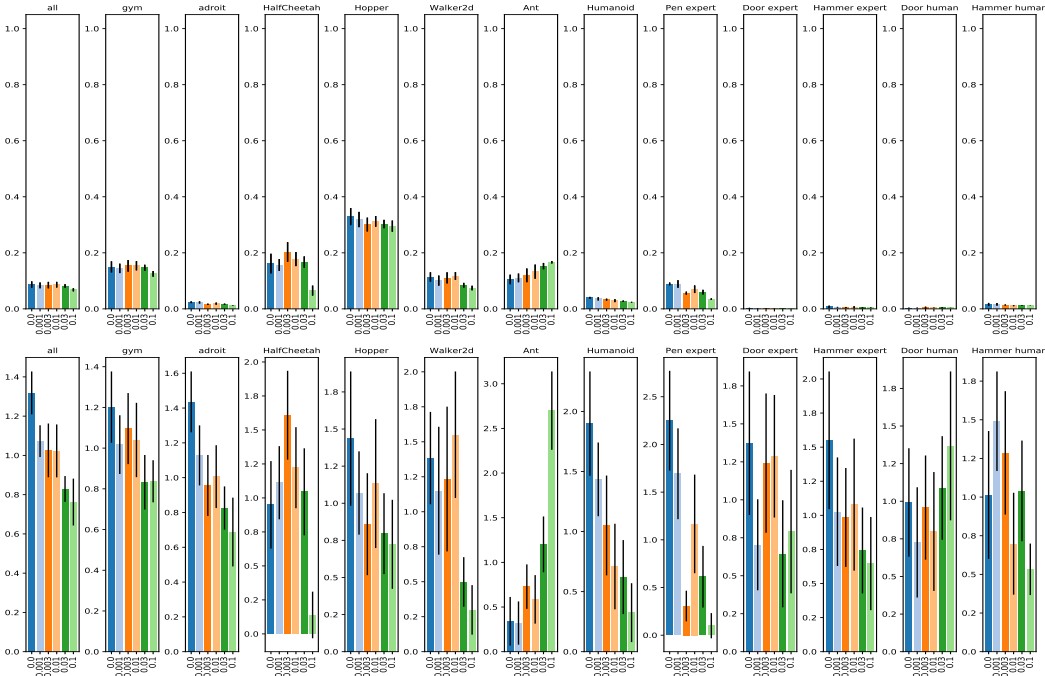

Figure 16: Analysis of choice `PPO entropy cost (C13)`: 95th percentile of performance scores conditioned on choice (top) and distribution of choices in top 5% of configurations (bottom).

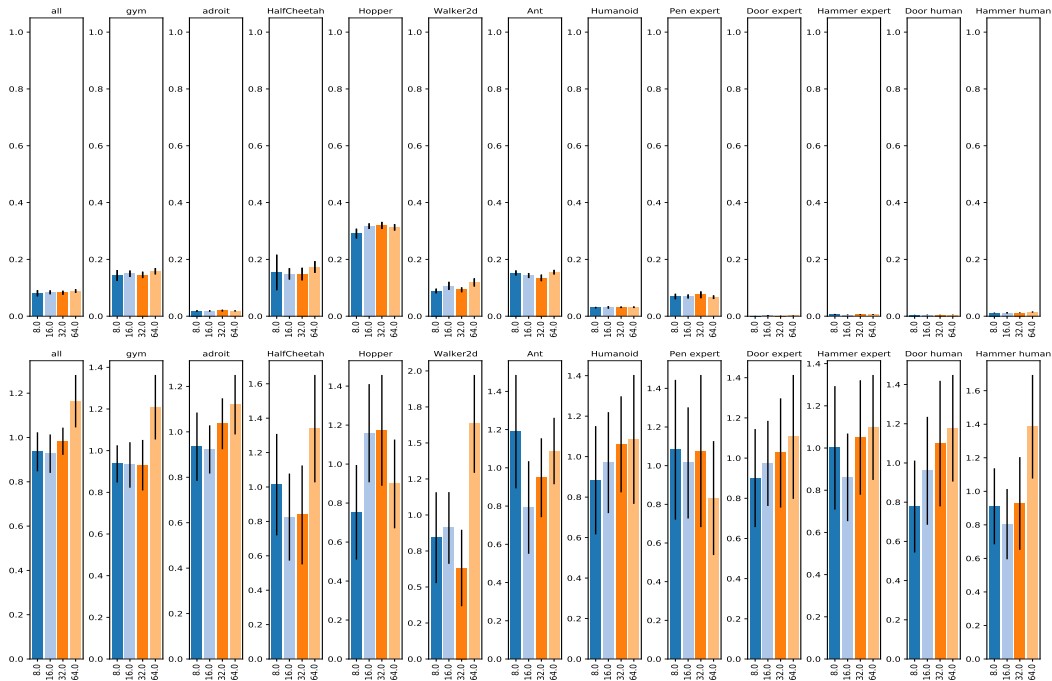

Figure 17: Analysis of choice PPO `number of minibatches (C11)`: 95th percentile of performance scores conditioned on choice (top) and distribution of choices in top 5% of configurations (bottom).

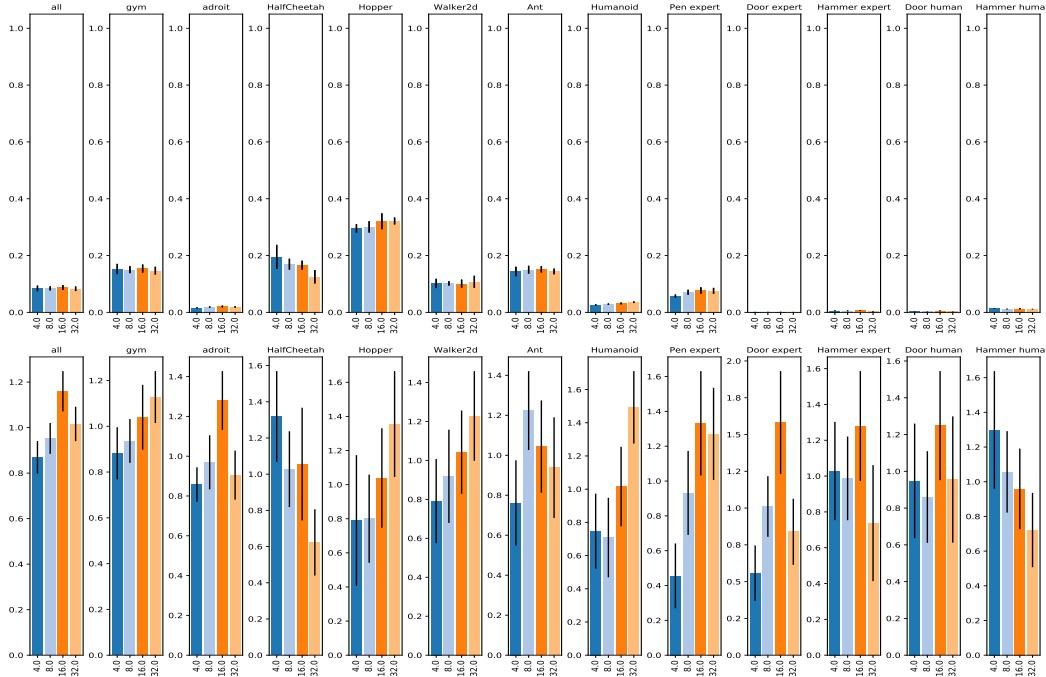

Figure 18: Analysis of choice PPO `unroll length (C9)`: 95th percentile of performance scores conditioned on choice (top) and distribution of choices in top 5% of configurations (bottom).

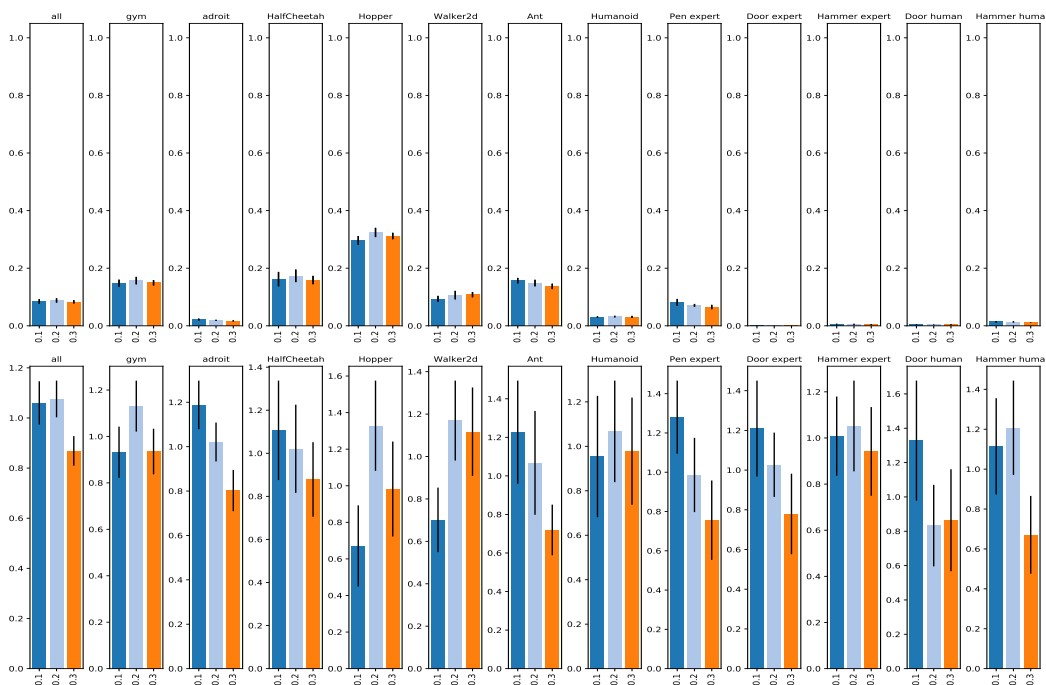

Figure 19: Analysis of choice PPO clipping $\epsilon$ (C12): 95th percentile of performance scores conditioned on choice (top) and distribution of choices in top 5% of configurations (bottom).

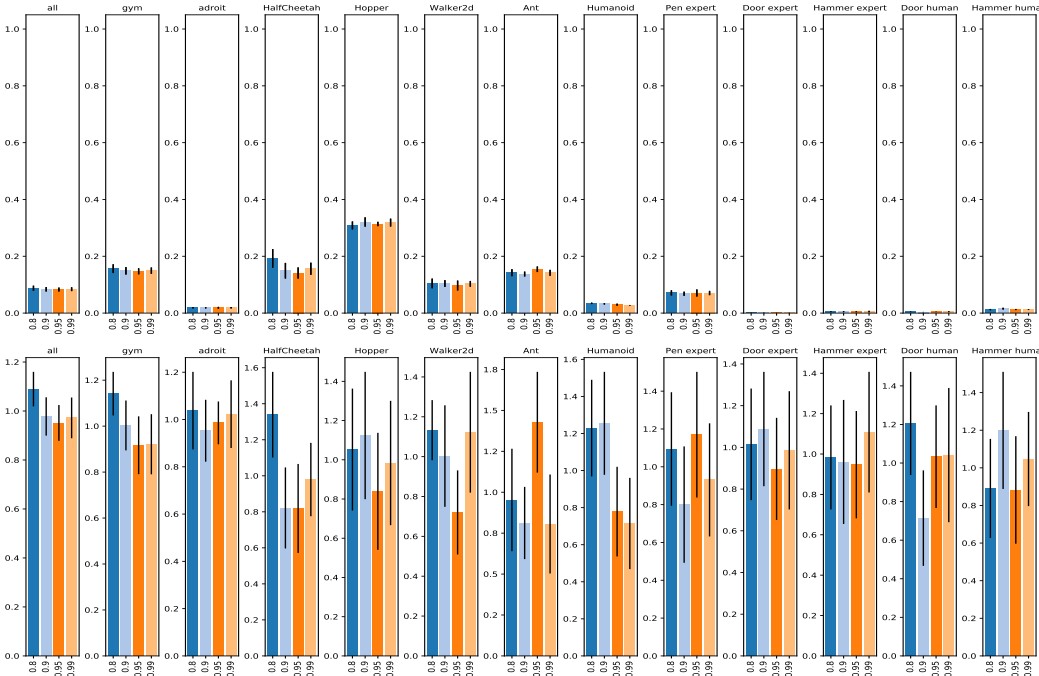

Figure 20: Analysis of choice GAE $\lambda$ (C15): 95th percentile of performance scores conditioned on choice (top) and distribution of choices in top 5% of configurations (bottom).

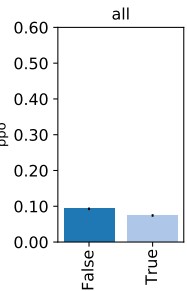 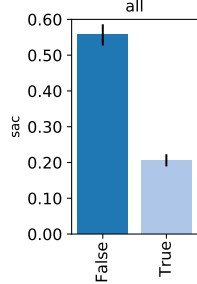

Figure 21: 95th percentile of performance scores conditioned on `RL Algorithm (C8)`(subplots) and `subtract log-pi (C40)`(bars).

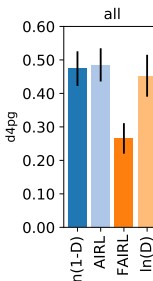 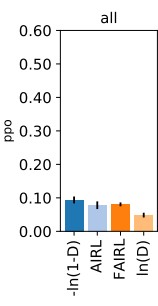 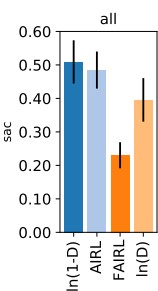 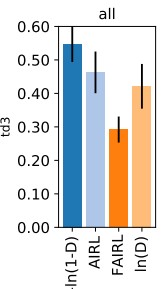

Figure 22: 95th percentile of performance scores conditioned on `RL Algorithm (C8)`(subplots) and `reward function (C30)`(bars).

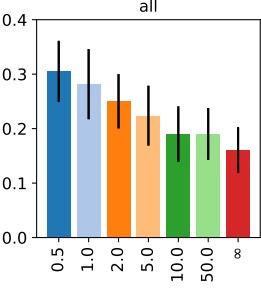

Figure 23: 95th percentile of performance scores conditioned on `max reward magnitude (C31)` and `reward function (C30)=FAIRL`.

# H Experiment main

## H.1 Design

For each of the 10 tasks, we sampled 25334 choice configurations where we sampled the following choices independently and uniformly from the following ranges:

- `RL Algorithm (C8)`: {d4pg, sac, td3}
  - For the case "`RL Algorithm (C8)` = sac", we further sampled the sub-choices:
    * `SAC learning rate (C17)`: {0.0001, 0.0003, 0.001}
    * `SAC entropy per dimension (C16)`: {-2.0, -1.0, -0.5, 0.0}
    * `SAC polyak` $\tau$ `(C18)`: {0.001, 0.003, 0.01, 0.03}
  - For the case "`RL Algorithm (C8)` = d4pg", we further sampled the sub-choices:
    * `D4PG learning rate (C26)`: {3e-05, 0.0001, 0.0003}
    * `behavioral policy noise (C21)`: {0.1, 0.2, 0.3, 0.5}
    * `VMax (C24)`: {150.0, 750.0, 1500.0}
    * `number of atoms (C23)`: {51.0, 101.0, 201.0, 401.0}
    * `N-step returns (C25)`: {1.0, 3.0, 5.0}
  - For the case "`RL Algorithm (C8)` = td3", we further sampled the sub-choices:
    * `TD3 policy learning rate (C19)`: {0.0001, 0.0003, 0.001}
    * `TD3 critic learning rate (C20)`: {0.0001, 0.0003, 0.001}
    * `TD3 gradient clipping (C22)`: {40.0, $\infty$}
    * `behavioral policy noise (C21)`: {0.1, 0.2, 0.3, 0.5}
- `RL replay buffer size (C28)`: {300000, 1000000, 3000000}
- `policy MLP depth (C1)`: {1, 2, 3}
- `policy MLP width (C2)`: {64, 128, 256, 512}
- `critic MLP depth (C3)`: {2, 3}
- `critic MLP width (C4)`: {256, 512}
- `RL activation (C5)`: {relu, tanh}
- `discount` $\gamma$ `(C6)`: {0.97, 0.99}
- `BC pretraining (C34)`: {False, True}
- `absorbing state (C32)`: {False, True}
- `discriminator replay buffer size (C43)`: {300000, 1000000, 3000000}
- `reward shaping (C39)`: {False, True}
- `discriminator input (C35)`: {s, sa, sas, ss}
- `discriminator MLP depth (C36)`: {1, 2, 3}
- `discriminator MLP width (C37)`: {16, 32, 64, 128, 256, 512}
- `discriminator activation (C38)`: {elu, leaky_relu, relu, sigmoid, swish, tanh}
- `discriminator last layer init scale (C41)`: {0.001, 1.0}
- `discriminator regularizer (C45)`: {GP, Mixup, No regularizer, PUGAIL, dropout, entropy, spectral norm, weight decay}
  - For the case "`discriminator regularizer (C45)` = GP", we further sampled the sub-choices:
    * `gradient penalty` $\lambda$ `(C47)`: {0.1, 1.0, 10.0}
    * `gradient penalty k (C46)`: {0.0, 1.0}
  - For the case "`discriminator regularizer (C45)` = Mixup", we further sampled the sub-choices:
    * `mixup` $\alpha$ `(C48)`: {0.1, 0.4, 1.0}
  - For the case "`discriminator regularizer (C45)` = PUGAIL", we further sampled the sub-choices:

- ∗ PUGAIL $\eta$ (C49): {0.25, 0.5, 0.7}
- ∗ PUGAIL $\beta$ (C50): {0.0, 0.7, $\infty$}
  - – For the case "`discriminator regularizer (C45)` = entropy", we further sampled the sub-choices:
    - ∗ `entropy` $\lambda$ (C54): {0.0003, 0.001, 0.003, 0.01, 0.03, 0.1, 0.3}
  - – For the case "`discriminator regularizer (C45)` = weight decay", we further sampled the sub-choices:
    - ∗ `weight decay` $\lambda$ (C53): {0.3, 1.0, 3.0, 10.0, 30.0}
  - – For the case "`discriminator regularizer (C45)` = dropout", we further sampled the sub-choices:
    - ∗ `dropout input rate (C52)`: {0.0, 0.25, 0.5, 0.75}
    - ∗ `dropout hidden rate (C51)`: {0.25, 0.5, 0.75}
- `observation normalization (C55)`: {fixed, none, online}
- `evaluation behavior policy type (C29)`: {average, mode, stochastic}
- `discriminator learning rate (C42)`: {1e-06, 3e-06, 1e-05, 3e-05, 0.0001, 0.0003}
- `reward function (C30)`: {-ln(1-D), AIRL, ln(D)}
- `batch size (C7)`: {256}
- `replay ratio (C27)`: {256}
- `discriminator to RL updates ratio (C44)`: {1}
- `number of combined batches (C56)`: {8}

## H.2   Results

For each of the sampled choice configurations we compute the performance metric as described in Section 2. We report aggregate statistics of the experiment in Tables 7–10 as well as training curves in Figure 24. We further provide per-choice analyses in Figures 37-71.

Table 7: Quantiles of the *final* agent performance across HP configurations for OpenAI Gym tasks.

|      | Ant  | HalfCheetah | Hopper | Humanoid | Walker2d |
|------|------|-------------|--------|----------|----------|
| 90%  | 0.90 | 1.07        | 1.18   | 0.51     | 0.99     |
| 95%  | 0.99 | 1.11        | 1.20   | 0.87     | 1.01     |
| 99%  | 1.07 | 1.17        | 1.23   | 1.01     | 1.04     |
| Max  | 1.18 | 1.37        | 1.34   | 1.06     | 1.21     |

Table 8: Quantiles of the *final* agent performance across HP configurations for Adroit tasks.

|      | Door expert | Door human | Hammer expert | Hammer human | Pen expert |
|------|-------------|------------|---------------|--------------|------------|
| 90%  | 0.72        | 0.25       | 1.08          | 0.46         | 0.74       |
| 95%  | 0.91        | 0.83       | 1.26          | 1.15         | 0.89       |
| 99%  | 1.04        | 2.29       | 1.37          | 3.04         | 1.11       |
| Max  | 1.16        | 3.73       | 1.45          | 5.55         | 1.44       |

Table 9: Quantiles of the *average* agent performance during training across HP configurations for OpenAI Gym tasks.

|      | Ant  | HalfCheetah | Hopper | Humanoid | Walker2d |
|------|------|-------------|--------|----------|----------|
| 90%  | 0.61 | 0.82        | 0.93   | 0.29     | 0.70     |
| 95%  | 0.72 | 0.87        | 0.98   | 0.53     | 0.76     |
| 99%  | 0.85 | 0.94        | 1.06   | 0.79     | 0.84     |
| Max  | 0.96 | 1.05        | 1.10   | 0.92     | 0.92     |

Table 10: Quantiles of the *average* agent performance during training across HP configurations for Adroit tasks.

|      | Door expert | Door human | Hammer expert | Hammer human | Pen expert |
|------|-------------|------------|---------------|--------------|------------|
| 90%  | 0.42        | 0.30       | 0.59          | 0.42         | 0.56       |
| 95%  | 0.57        | 0.56       | 0.77          | 0.70         | 0.66       |
| 99%  | 0.74        | 1.04       | 0.96          | 1.23         | 0.84       |
| Max  | 0.92        | 2.08       | 1.18          | 3.42         | 1.09       |

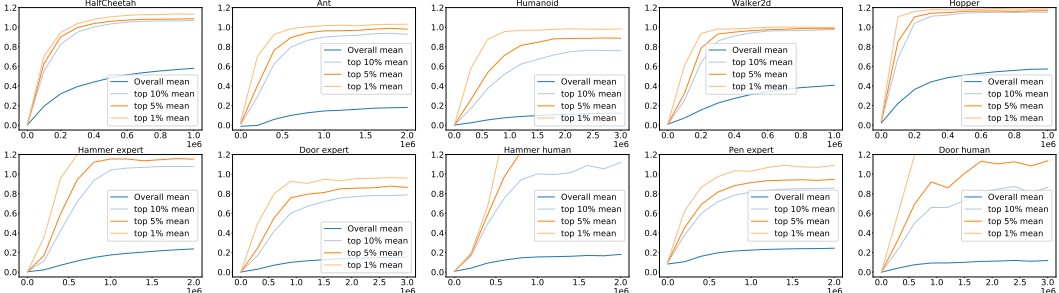

Figure 24: Training curves.

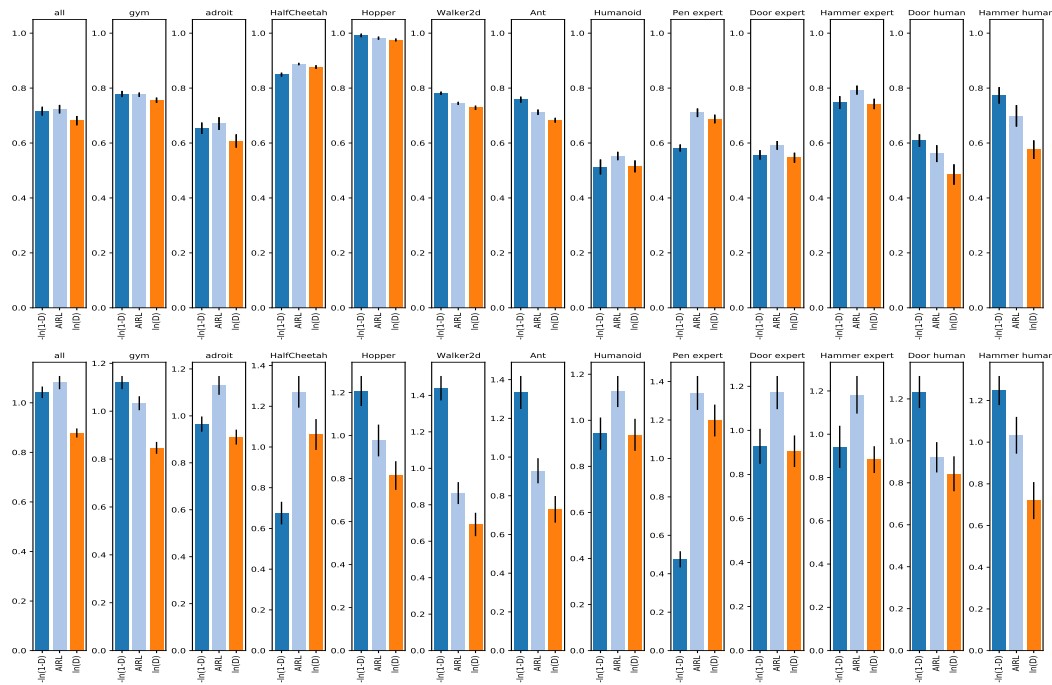

Figure 25: Analysis of choice `reward function` (C30): 95th percentile of performance scores conditioned on choice (top) and distribution of choices in top 5% of configurations (bottom).

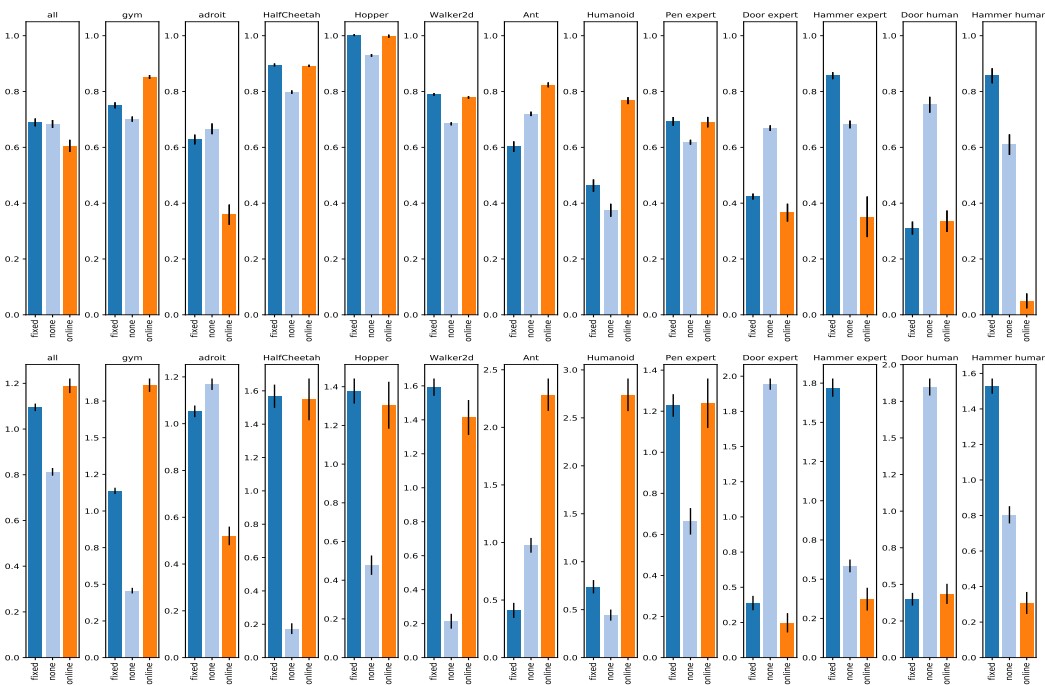

Figure 26: Analysis of choice `observation normalization` (C55): 95th percentile of performance scores conditioned on choice (top) and distribution of choices in top 5% of configurations (bottom).

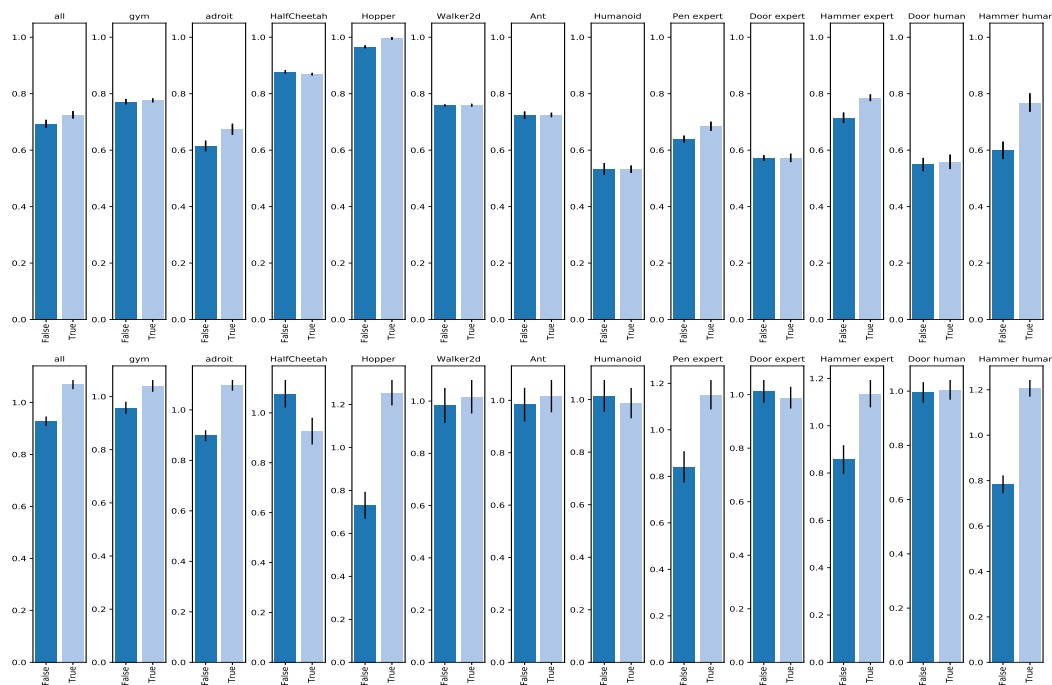

Figure 27: Analysis of choice `BC pretraining (C34)`: 95th percentile of performance scores conditioned on choice (top) and distribution of choices in top 5% of configurations (bottom).

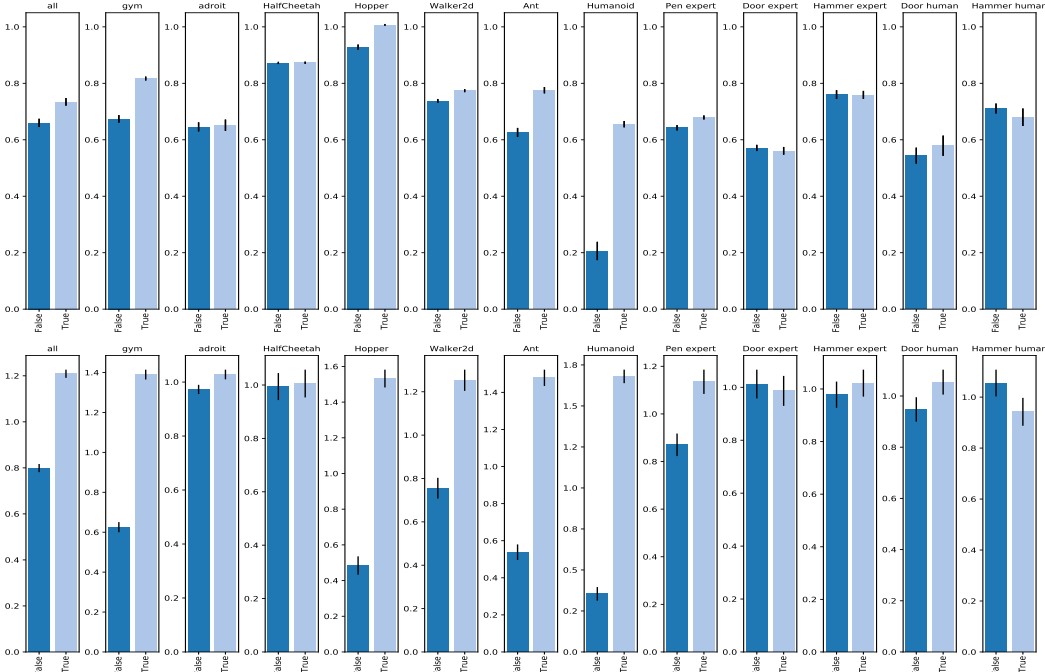

Figure 28: Analysis of choice `absorbing state (C32)`: 95th percentile of performance scores conditioned on choice (top) and distribution of choices in top 5% of configurations (bottom).

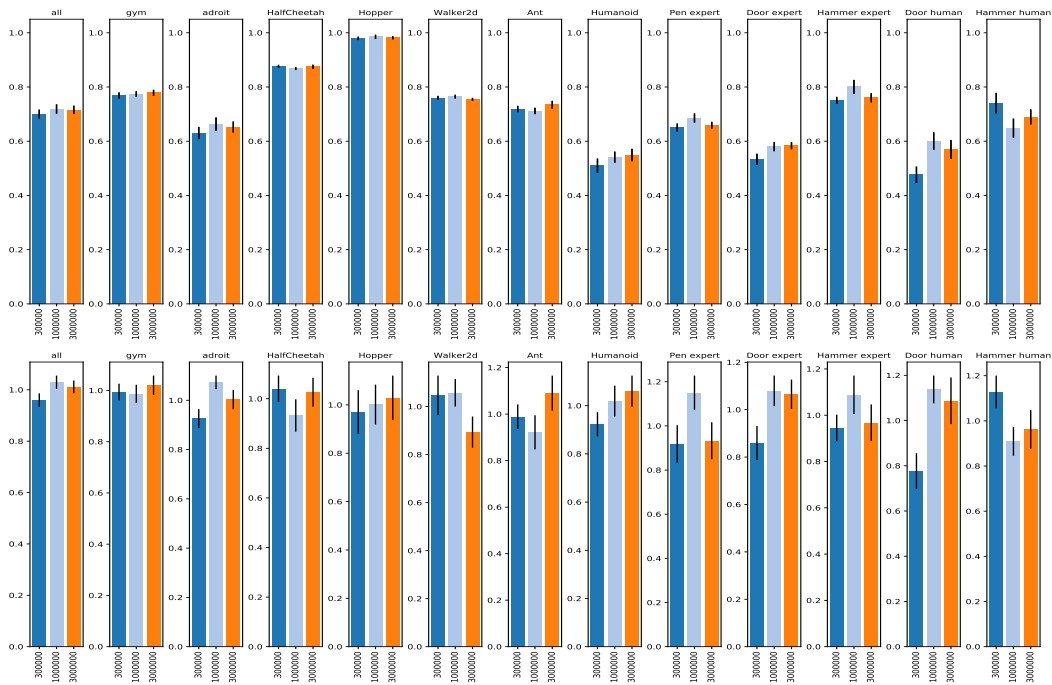

Figure 29: Analysis of choice `RL replay buffer size (C28)`: 95th percentile of performance scores conditioned on choice (top) and distribution of choices in top 5% of configurations (bottom).

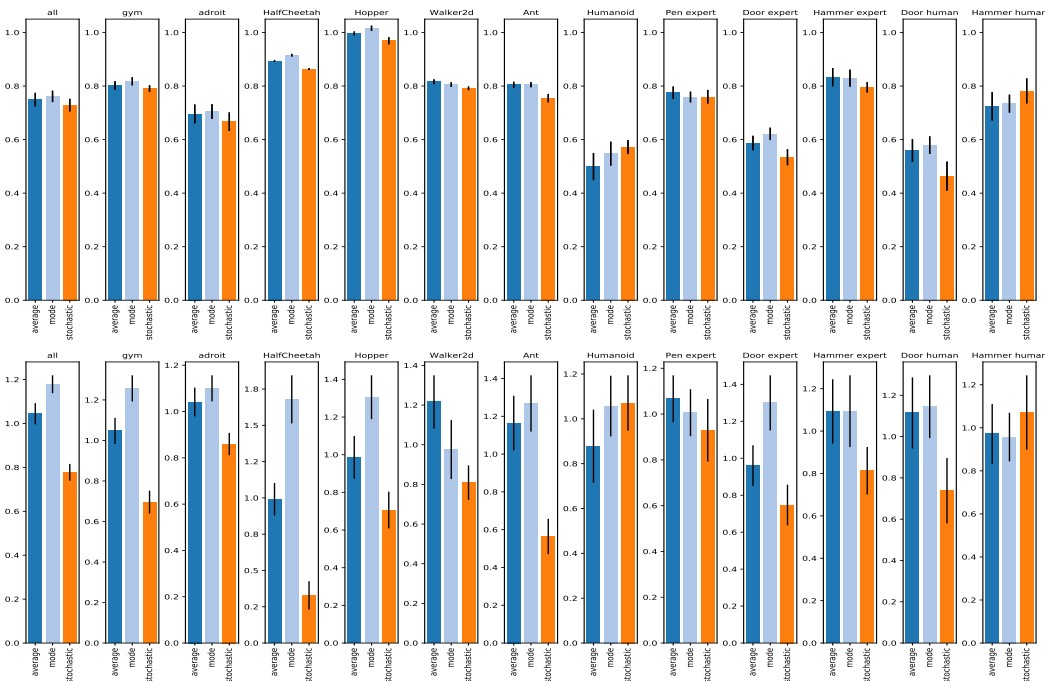

Figure 30: Analysis of choice `evaluation behavior policy type (C29)`: 95th percentile of performance scores conditioned on choice (top) and distribution of choices in top 5% of configurations (bottom).

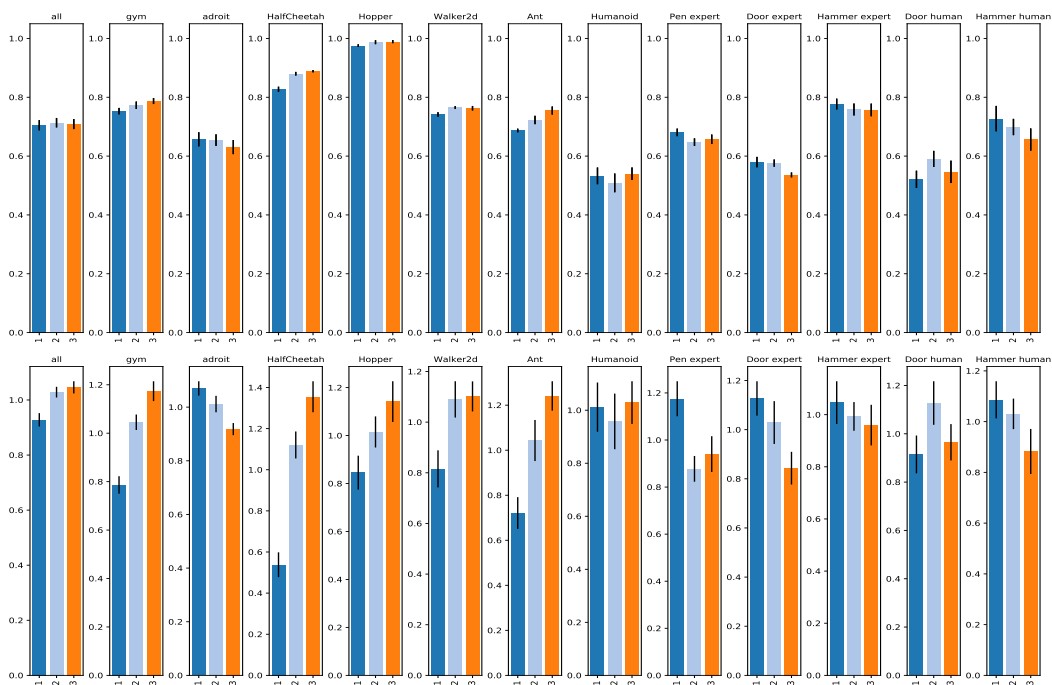

Figure 31: Analysis of choice `policy MLP depth (C1)`: 95th percentile of performance scores conditioned on choice (top) and distribution of choices in top 5% of configurations (bottom).

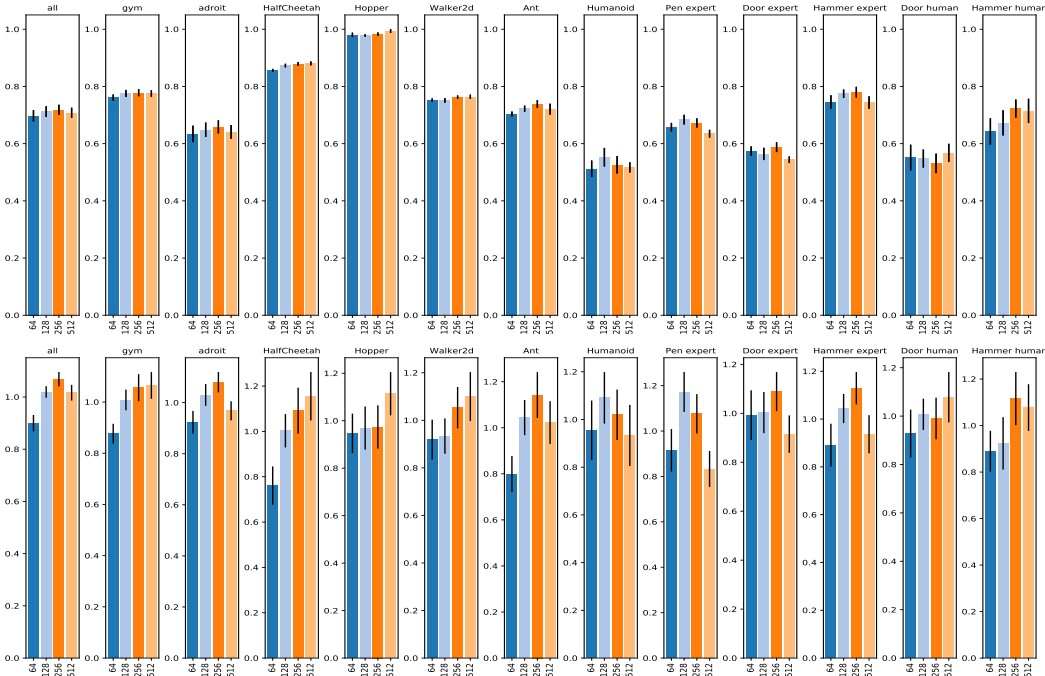

Figure 32: Analysis of choice `policy MLP width (C2)`: 95th percentile of performance scores conditioned on choice (top) and distribution of choices in top 5% of configurations (bottom).

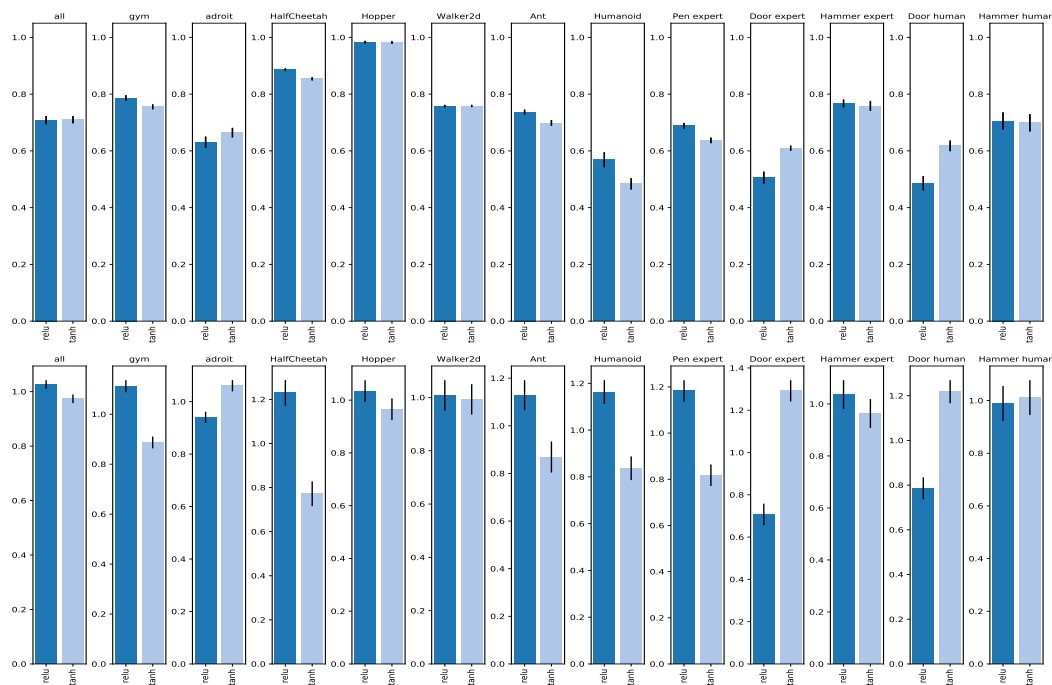

Figure 33: Analysis of choice RL `activation` (C5): 95th percentile of performance scores conditioned on choice (top) and distribution of choices in top 5% of configurations (bottom).

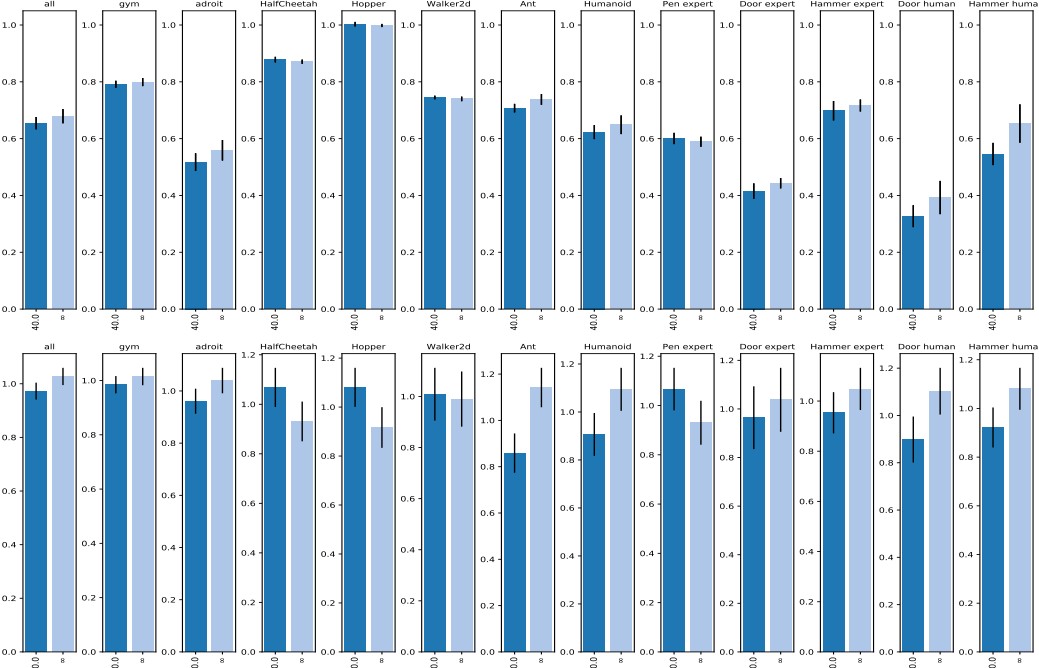

Figure 34: Analysis of choice TD3 `gradient clipping` (C22): 95th percentile of performance scores conditioned on choice (top) and distribution of choices in top 5% of configurations (bottom).

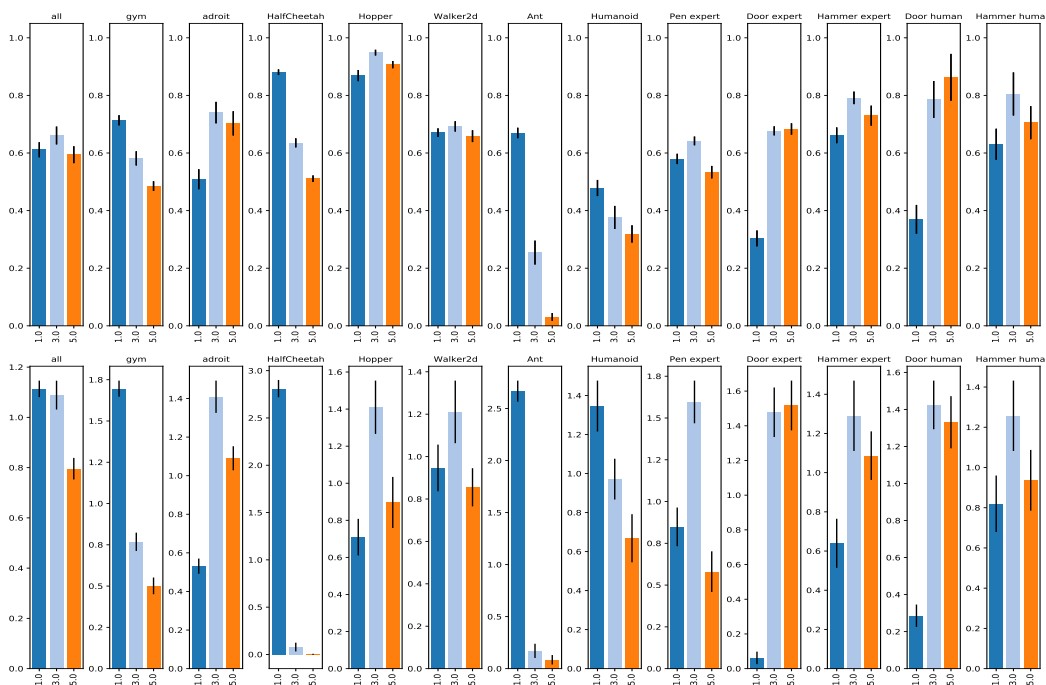

Figure 35: Analysis of choice `N-step returns` (C25): 95th percentile of performance scores conditioned on choice (top) and distribution of choices in top 5% of configurations (bottom).

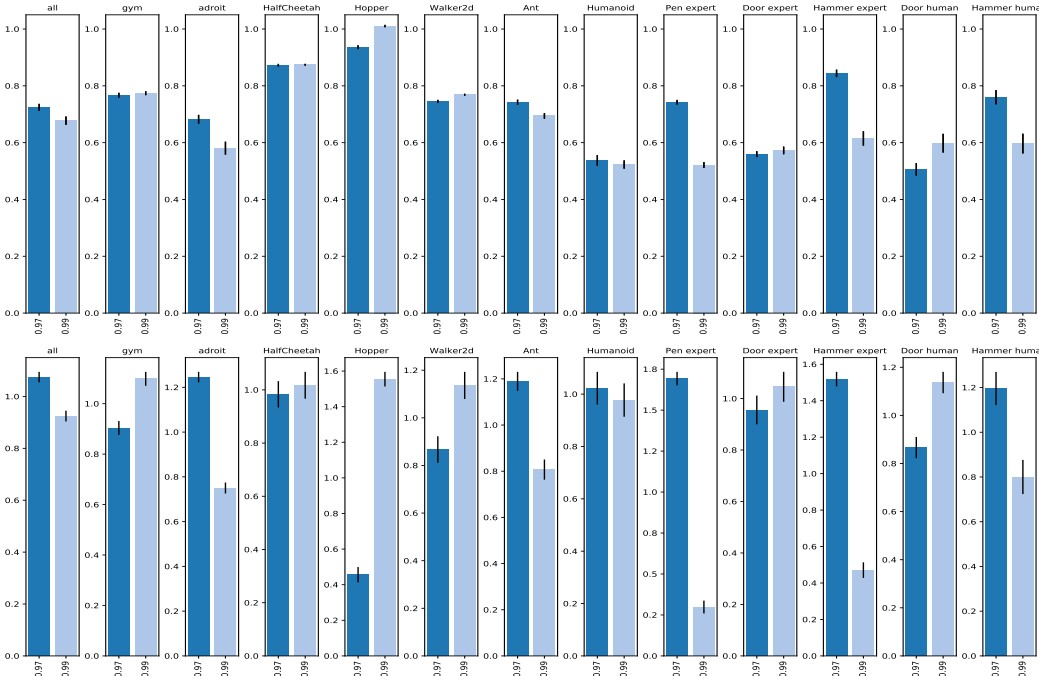

Figure 36: Analysis of choice `discount` $\gamma$ (C6): 95th percentile of performance scores conditioned on choice (top) and distribution of choices in top 5% of configurations (bottom).

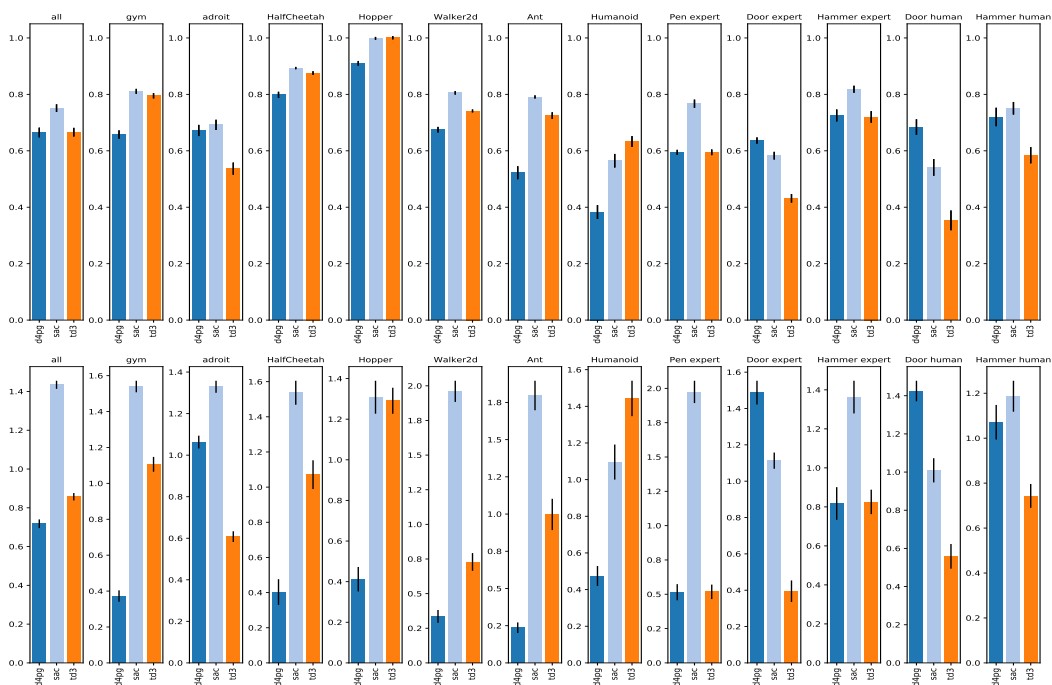

Figure 37: Analysis of choice RL Algorithm (C8): 95th percentile of performance scores conditioned on choice (top) and distribution of choices in top 5% of configurations (bottom).

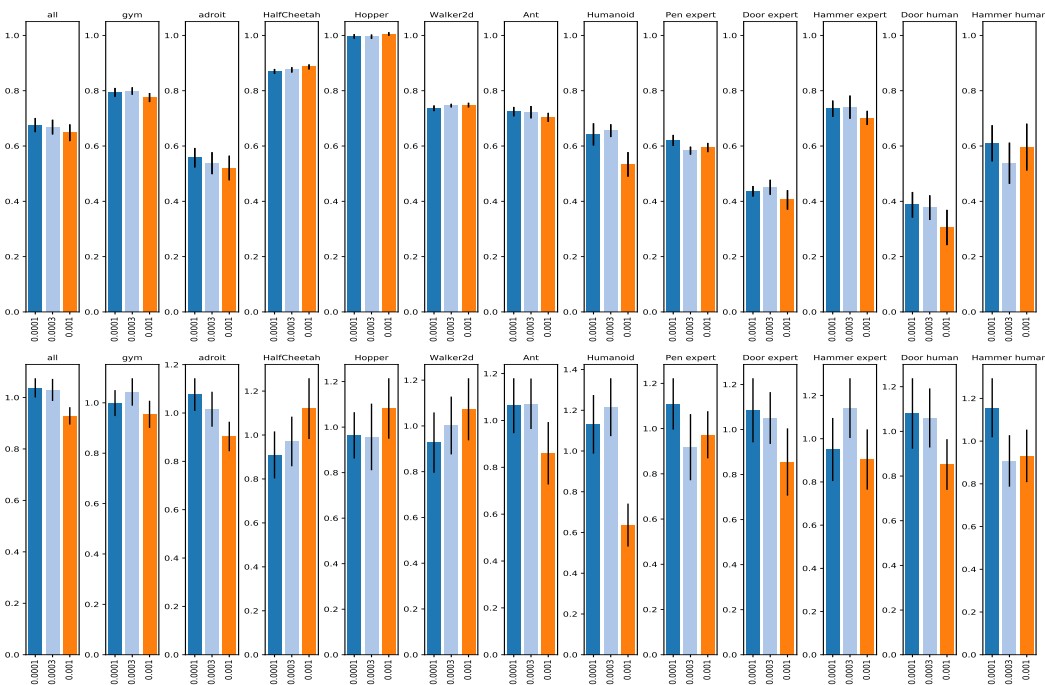

Figure 38: Analysis of choice TD3 policy learning rate (C19): 95th percentile of performance scores conditioned on choice (top) and distribution of choices in top 5% of configurations (bottom).

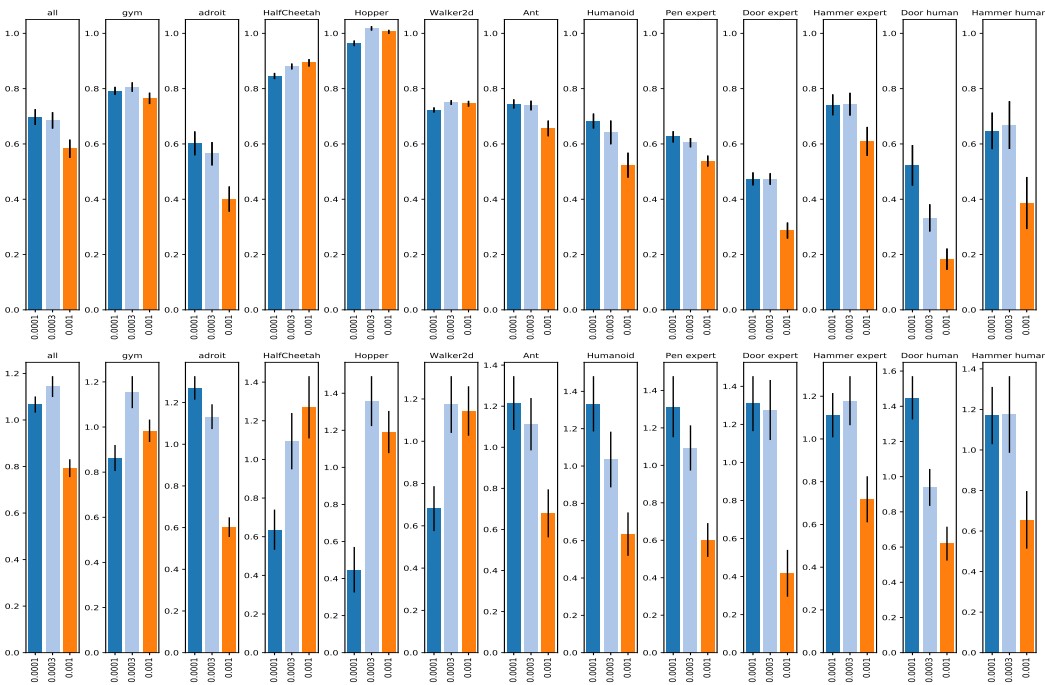

Figure 39: Analysis of choice `TD3 critic learning rate (C20)`: 95th percentile of performance scores conditioned on choice (top) and distribution of choices in top 5% of configurations (bottom).

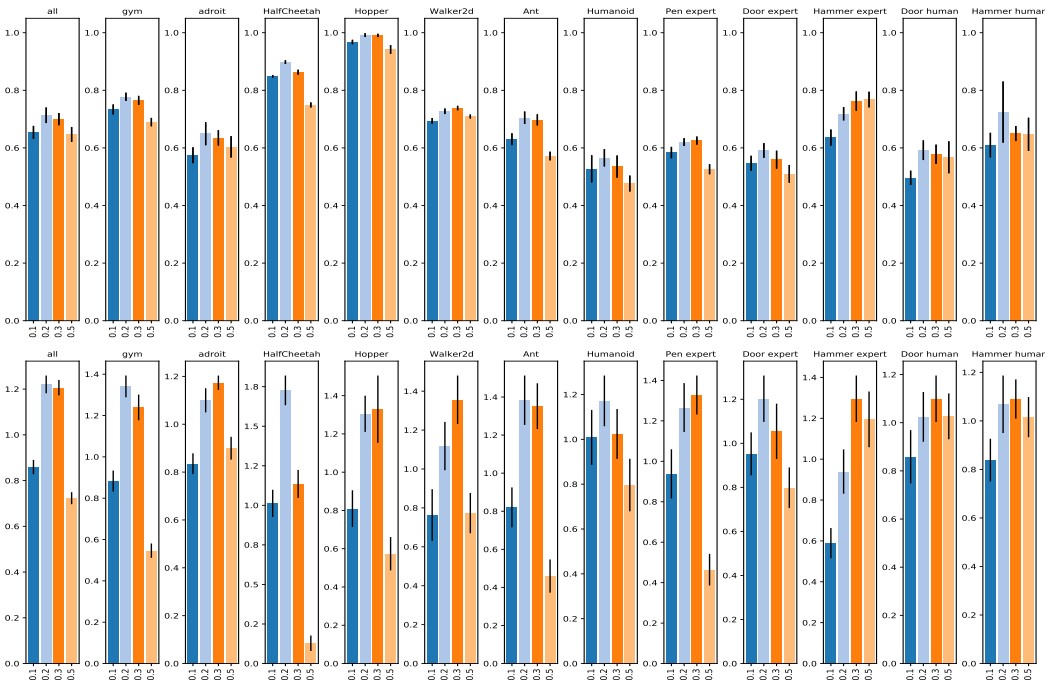

Figure 40: Analysis of choice `behavioral policy noise (C21)`: 95th percentile of performance scores conditioned on choice (top) and distribution of choices in top 5% of configurations (bottom).

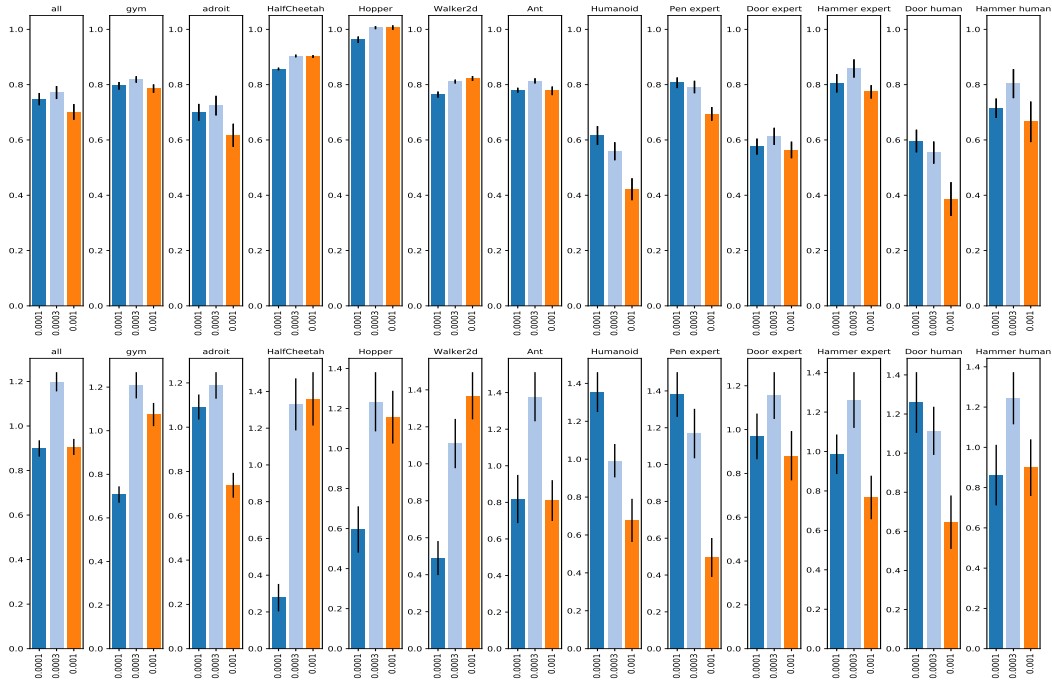

Figure 41: Analysis of choice `SAC learning rate (C17)`: 95th percentile of performance scores conditioned on choice (top) and distribution of choices in top 5% of configurations (bottom).

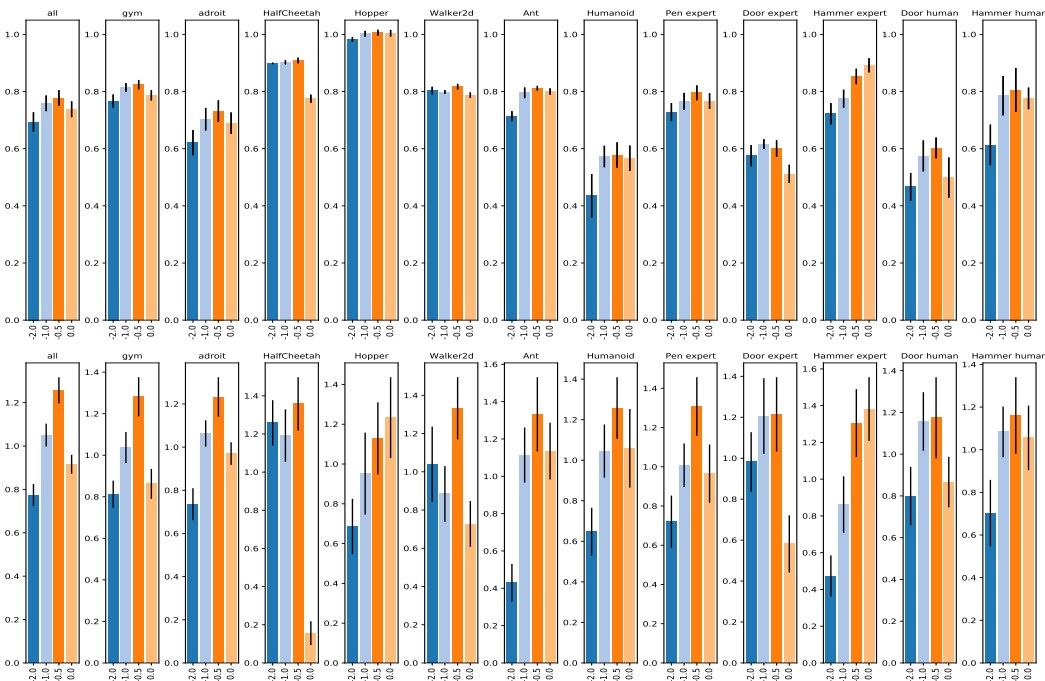

Figure 42: Analysis of choice `SAC entropy per dimension (C16)`: 95th percentile of performance scores conditioned on choice (top) and distribution of choices in top 5% of configurations (bottom).

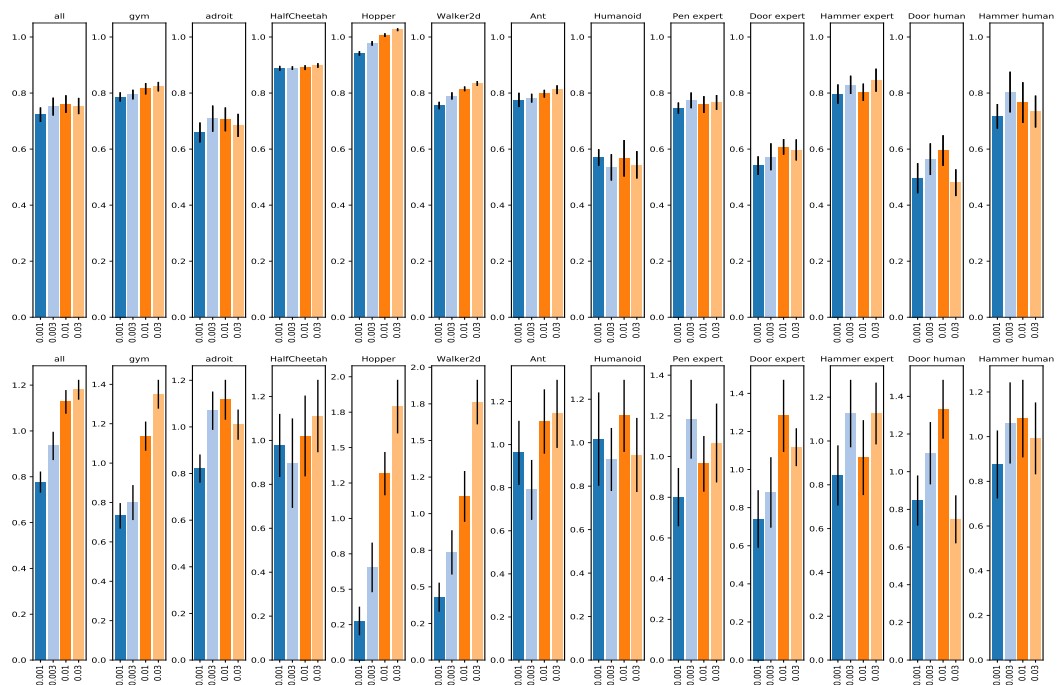

Figure 43: Analysis of choice `SAC polyak` $\tau$ `(C18)`: 95th percentile of performance scores conditioned on choice (top) and distribution of choices in top 5% of configurations (bottom).

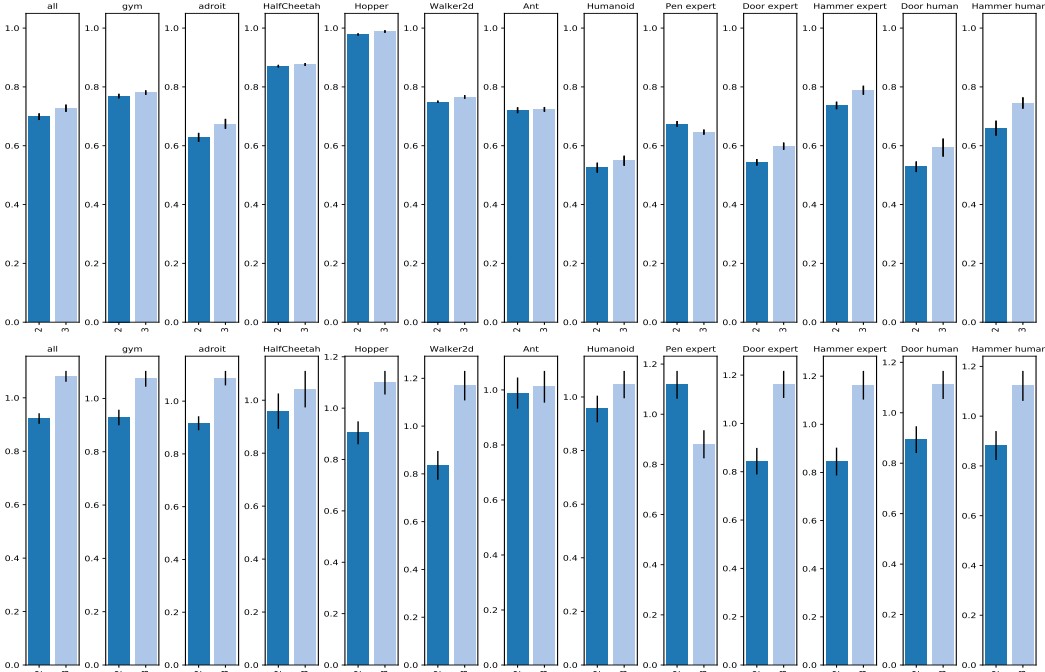

Figure 44: Analysis of choice `critic MLP depth (C3)`: 95th percentile of performance scores conditioned on choice (top) and distribution of choices in top 5% of configurations (bottom).

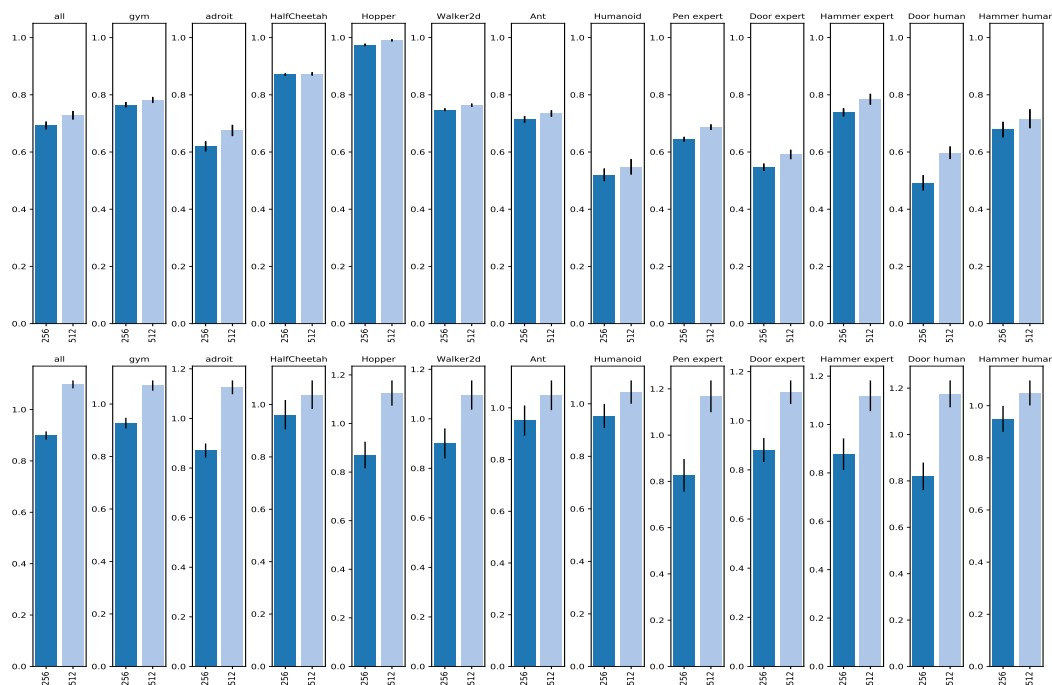

Figure 45: Analysis of choice `critic MLP width (C4)`: 95th percentile of performance scores conditioned on choice (top) and distribution of choices in top 5% of configurations (bottom).

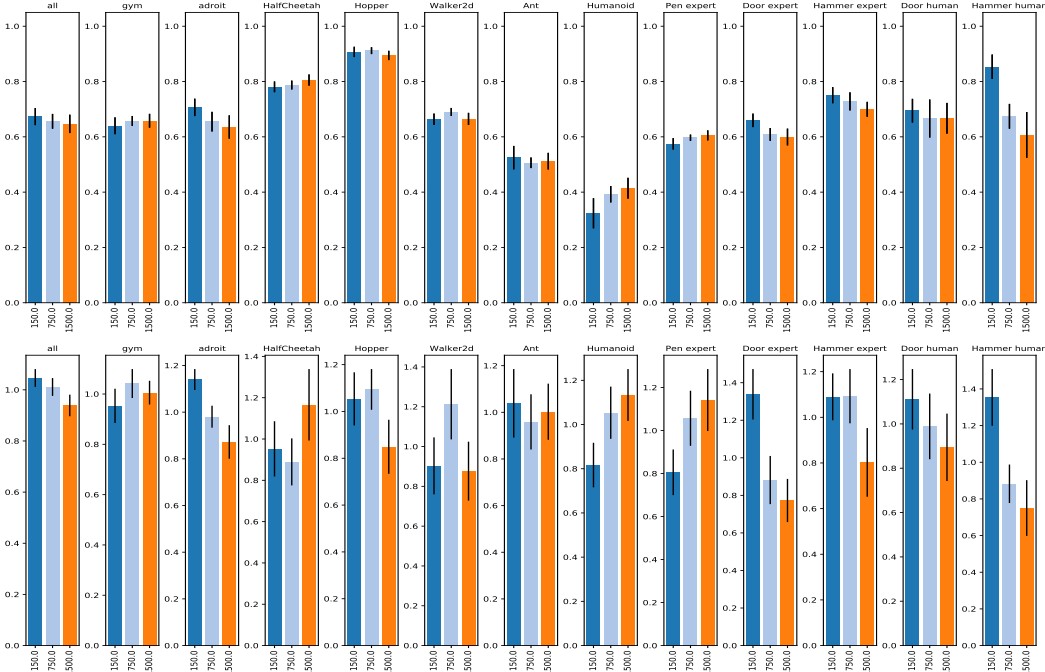

Figure 46: Analysis of choice `VMax (C24)`: 95th percentile of performance scores conditioned on choice (top) and distribution of choices in top 5% of configurations (bottom).

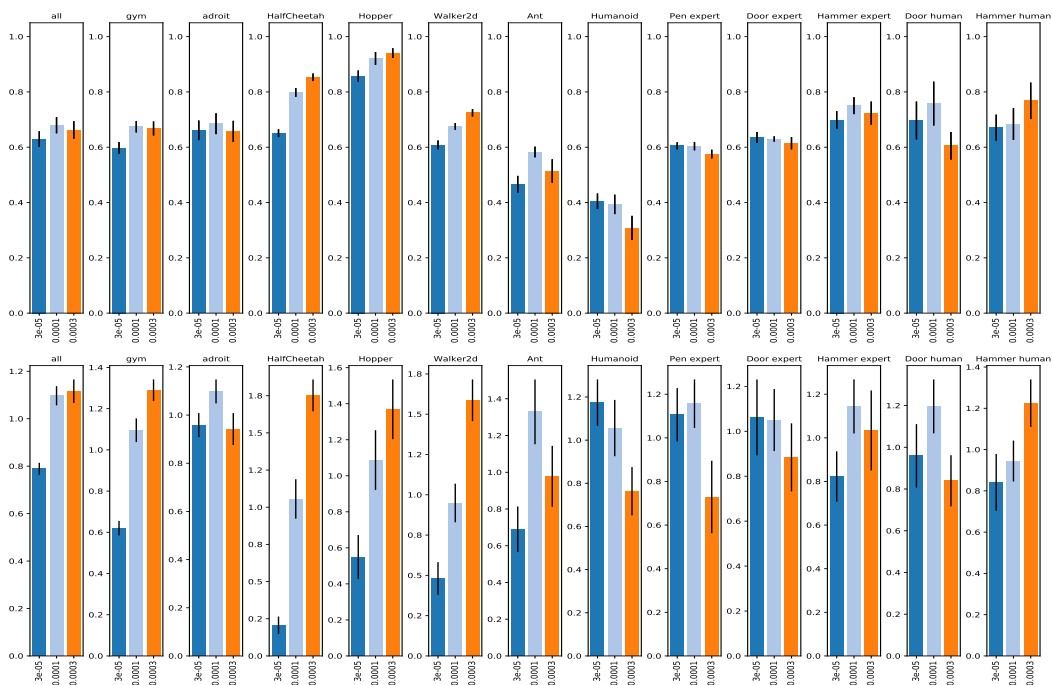

Figure 47: Analysis of choice `D4PG learning rate (C26)`: 95th percentile of performance scores conditioned on choice (top) and distribution of choices in top 5% of configurations (bottom).

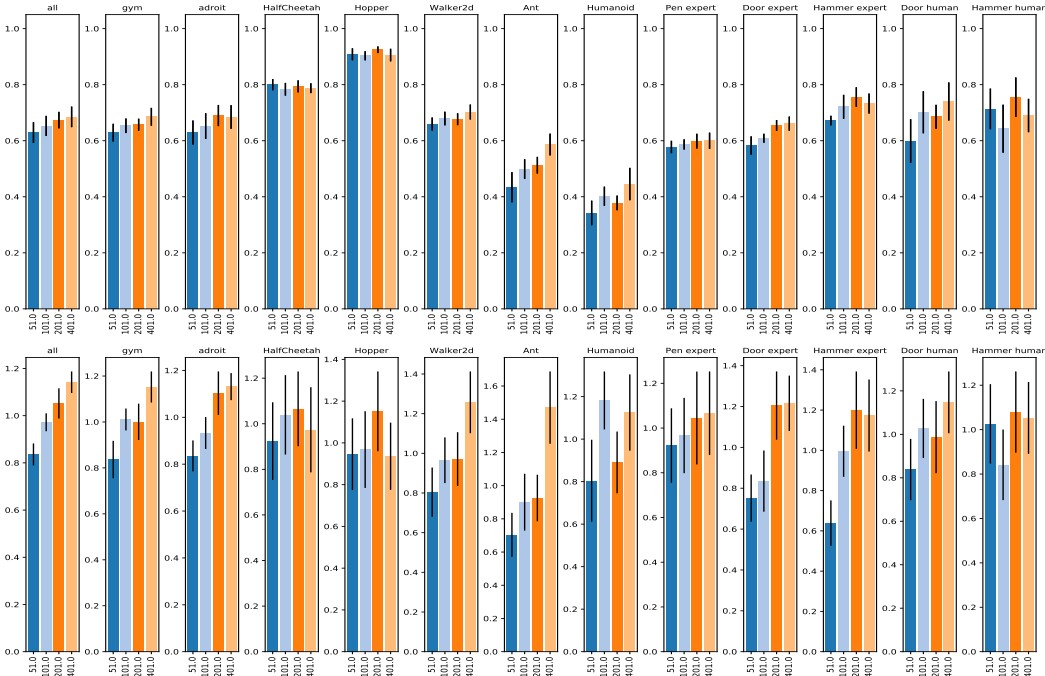

Figure 48: Analysis of choice `number of atoms (C23)`: 95th percentile of performance scores conditioned on choice (top) and distribution of choices in top 5% of configurations (bottom).

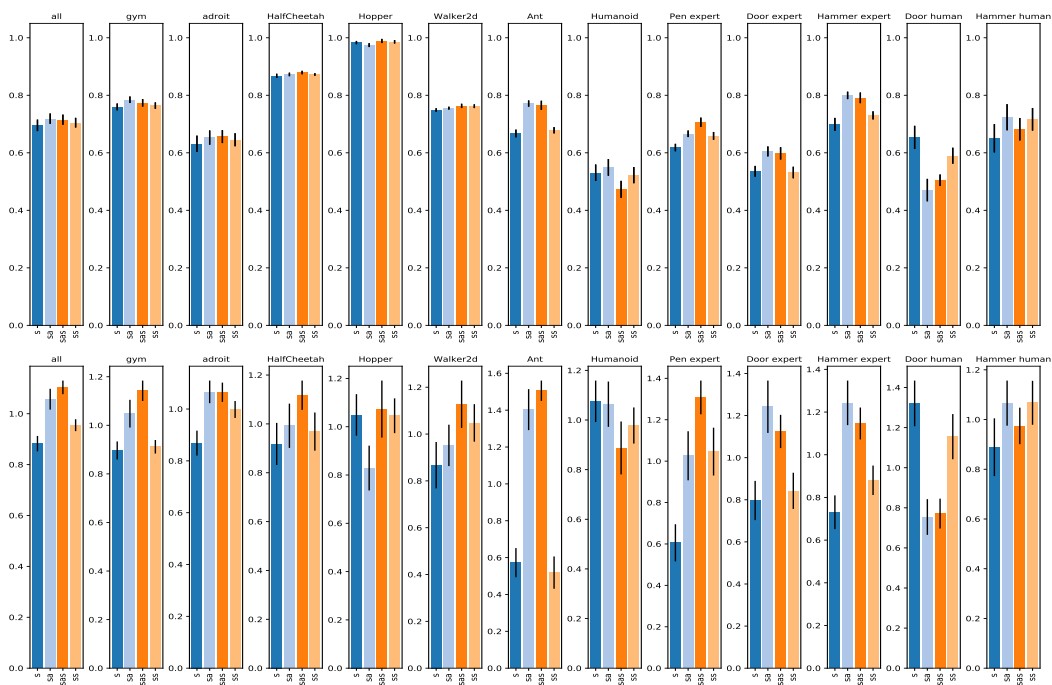

Figure 49: Analysis of choice `discriminator input (C35)`: 95th percentile of performance scores conditioned on choice (top) and distribution of choices in top 5% of configurations (bottom).

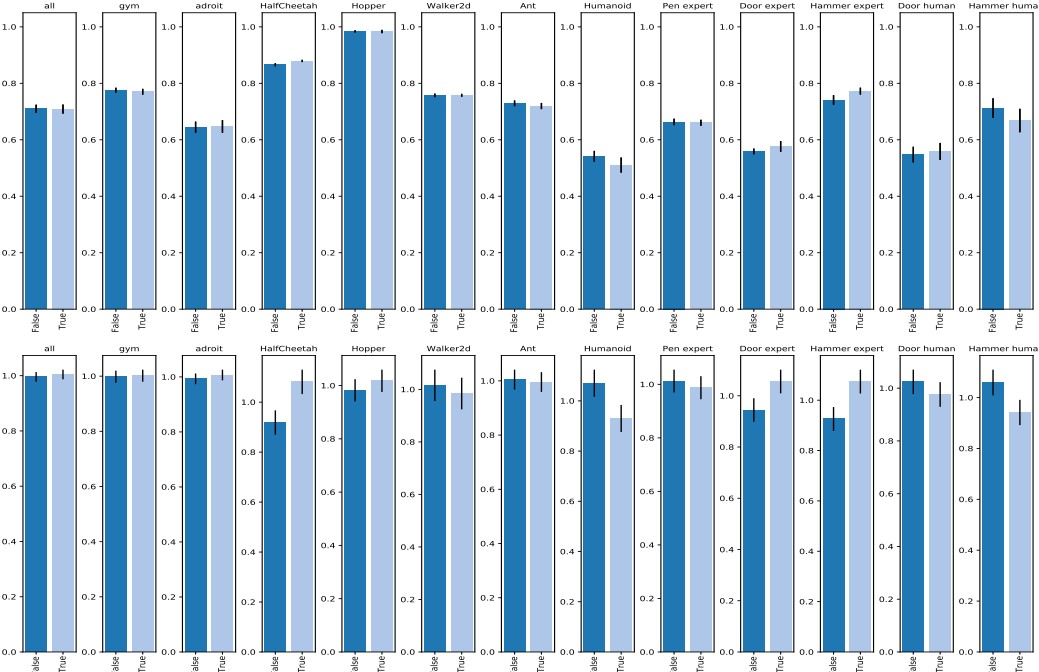

Figure 50: Analysis of choice `reward shaping (C39)`: 95th percentile of performance scores conditioned on choice (top) and distribution of choices in top 5% of configurations (bottom).

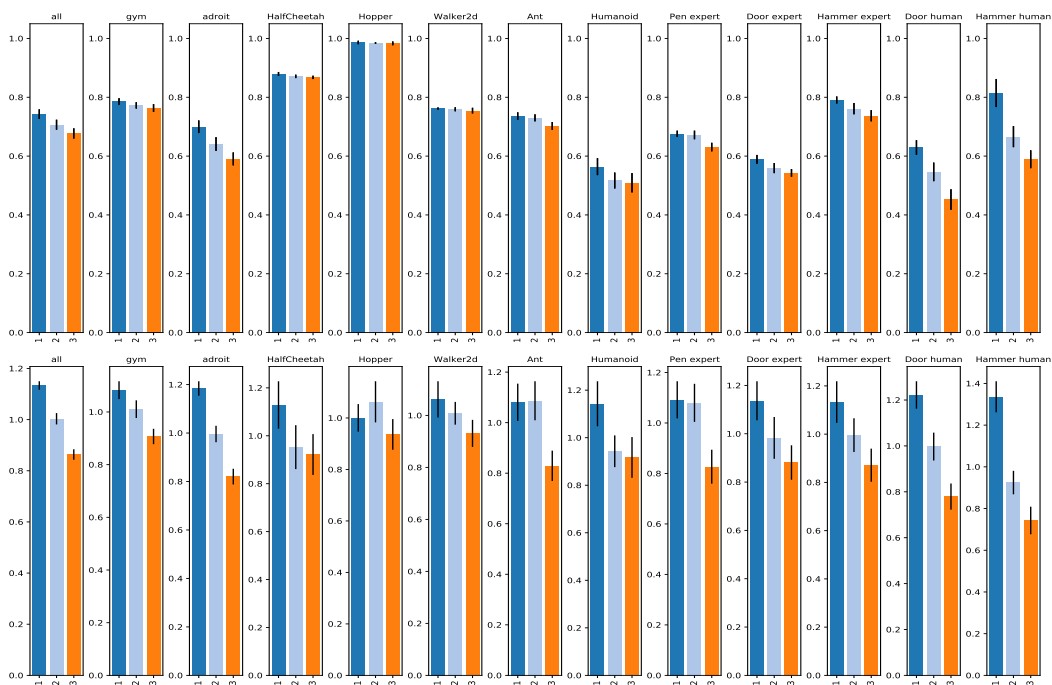

Figure 51: Analysis of choice `discriminator MLP depth (C36)`: 95th percentile of performance scores conditioned on choice (top) and distribution of choices in top 5% of configurations (bottom).

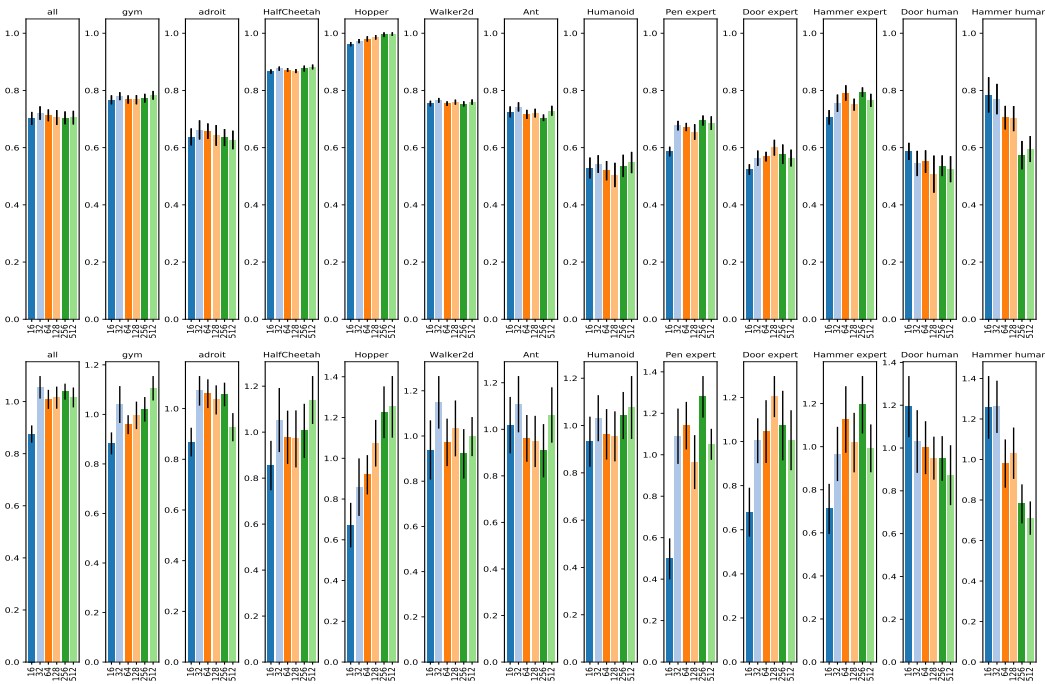

Figure 52: Analysis of choice `discriminator MLP width (C37)`: 95th percentile of performance scores conditioned on choice (top) and distribution of choices in top 5% of configurations (bottom).

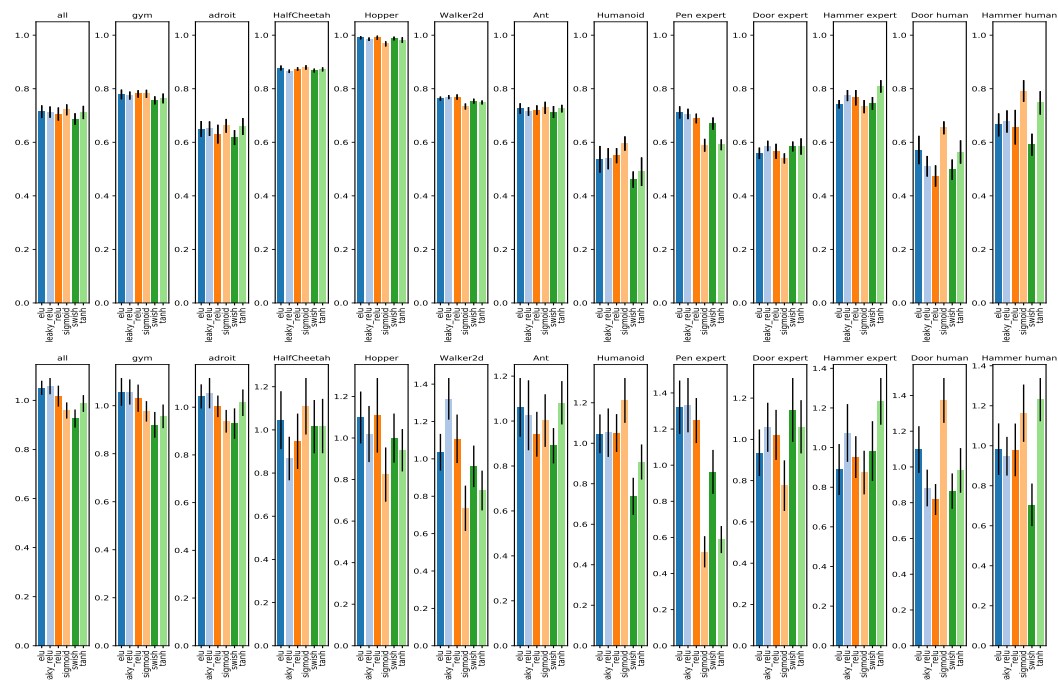

Figure 53: Analysis of choice `discriminator activation` (C38): 95th percentile of performance scores conditioned on choice (top) and distribution of choices in top 5% of configurations (bottom).

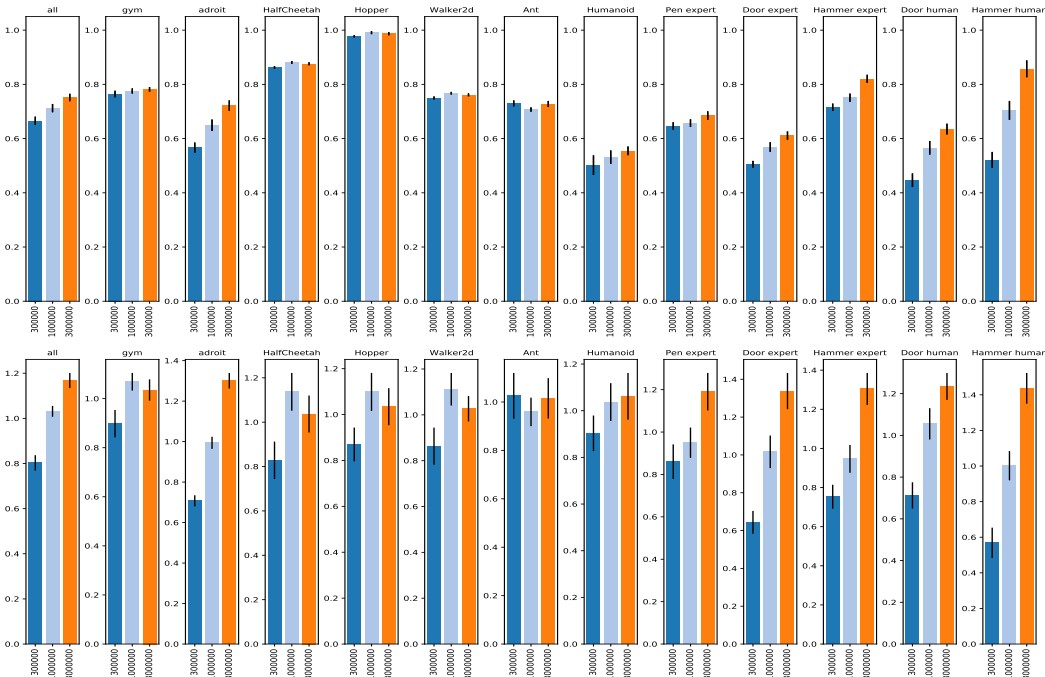

Figure 54: Analysis of choice `discriminator replay buffer size` (C43): 95th percentile of performance scores conditioned on choice (top) and distribution of choices in top 5% of configurations (bottom).

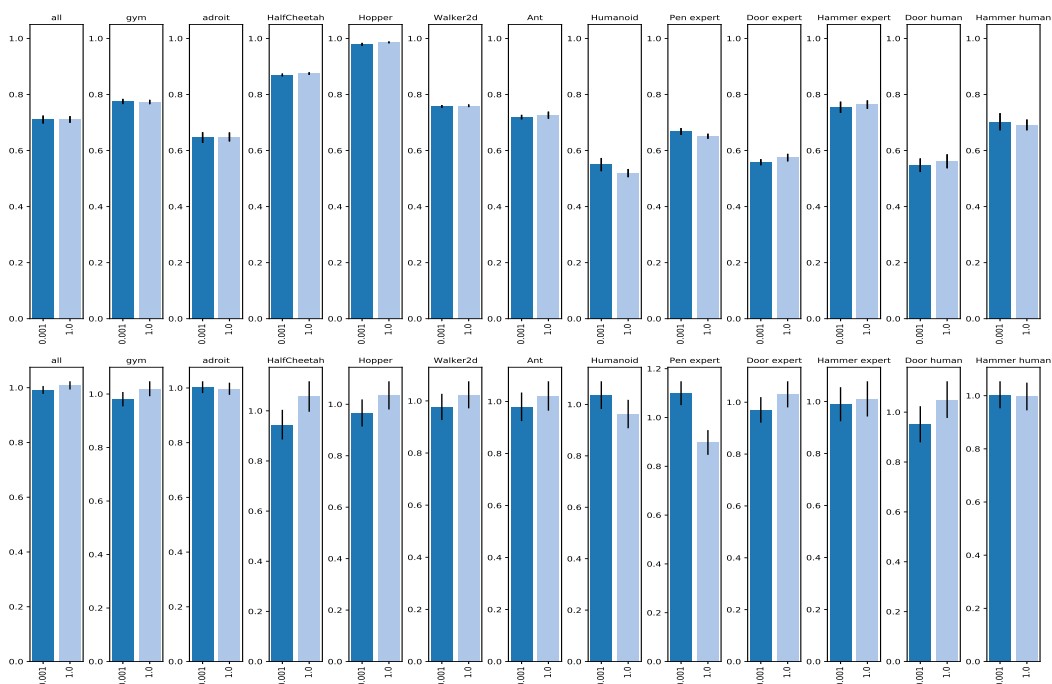

Figure 55: Analysis of choice `discriminator last layer init scale` (C41): 95th percentile of performance scores conditioned on choice (top) and distribution of choices in top 5% of configurations (bottom).

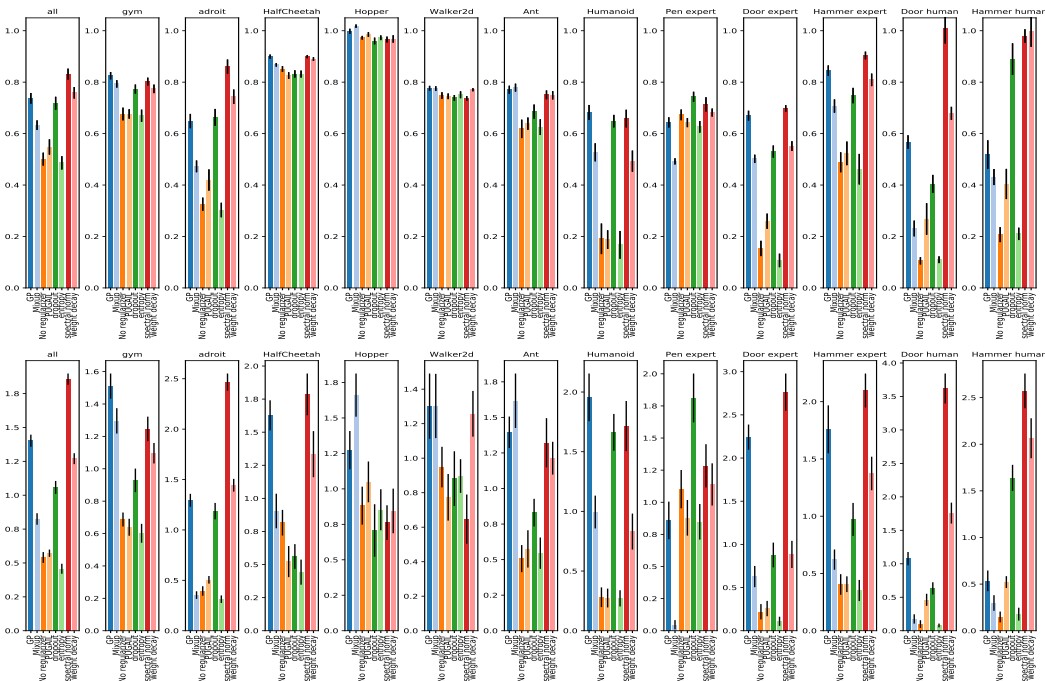

Figure 56: Analysis of choice `discriminator regularizer` (C45): 95th percentile of performance scores conditioned on choice (top) and distribution of choices in top 5% of configurations (bottom).

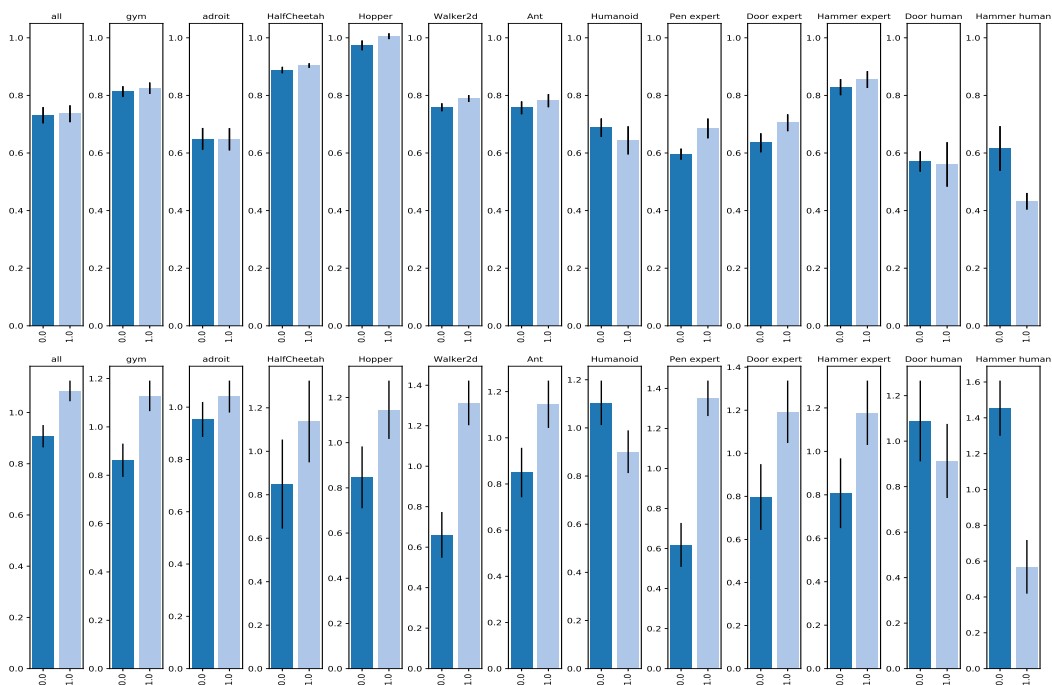

Figure 57: Analysis of choice `gradient penalty k` (C46): 95th percentile of performance scores conditioned on choice (top) and distribution of choices in top 5% of configurations (bottom).

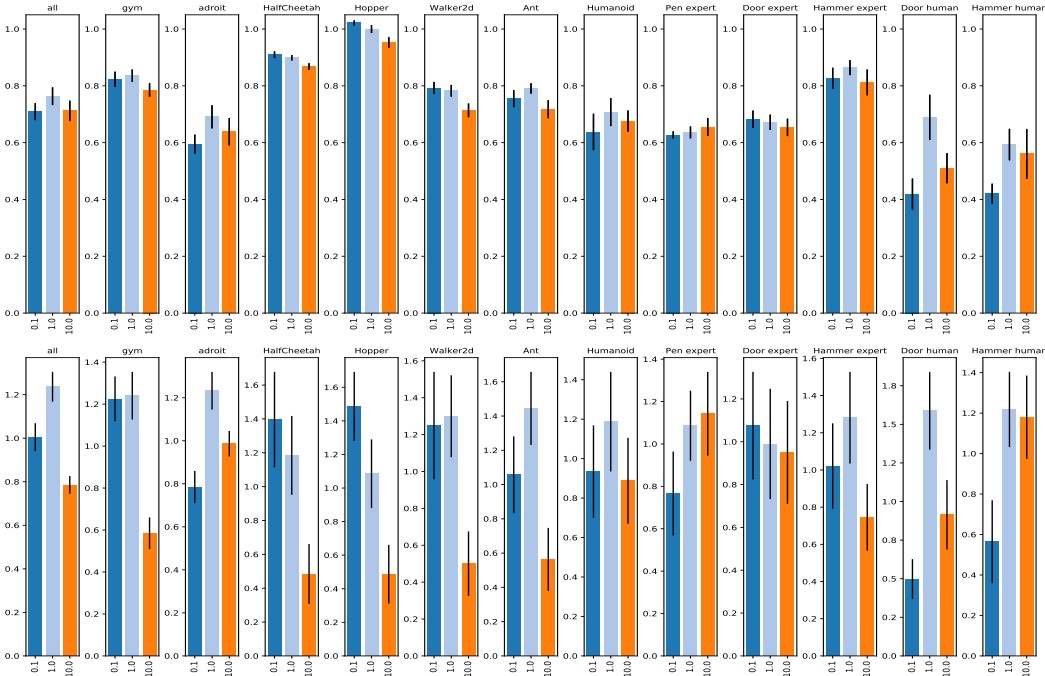

Figure 58: Analysis of choice `gradient penalty λ` (C47): 95th percentile of performance scores conditioned on choice (top) and distribution of choices in top 5% of configurations (bottom).

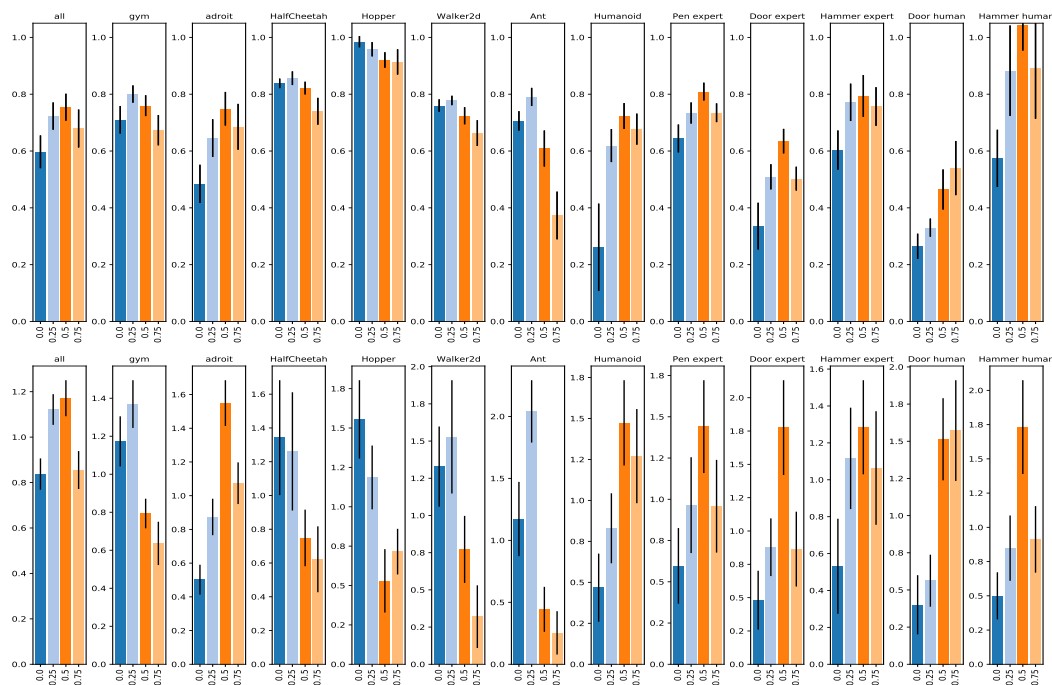

Figure 59: Analysis of choice `dropout input rate` (C52): 95th percentile of performance scores conditioned on choice (top) and distribution of choices in top 5% of configurations (bottom).

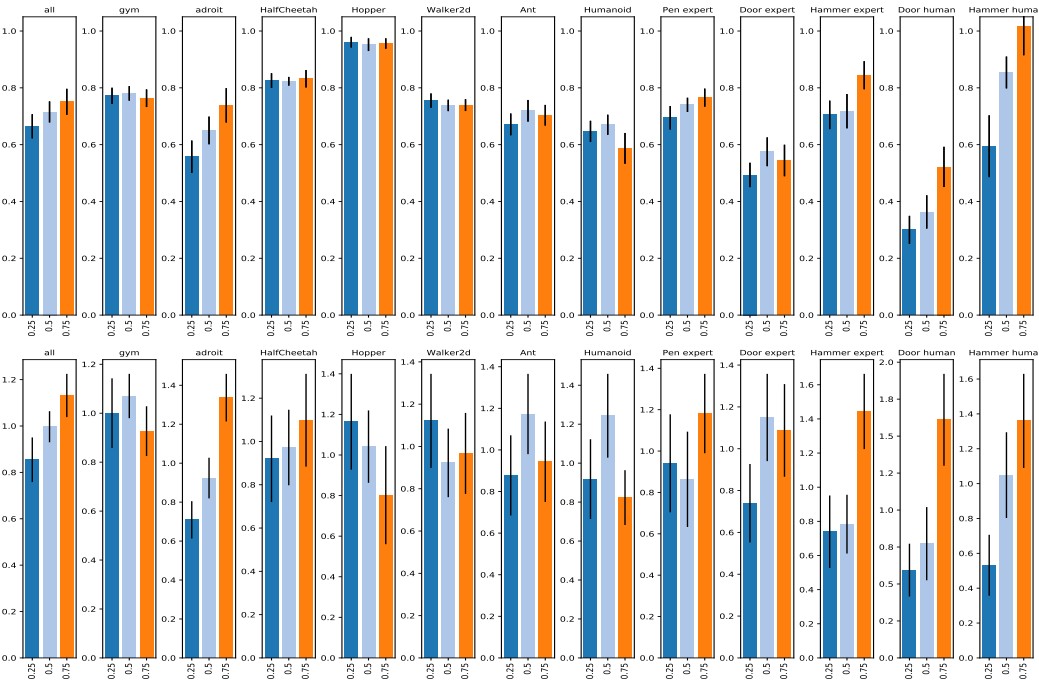

Figure 60: Analysis of choice `dropout hidden rate` (C51): 95th percentile of performance scores conditioned on choice (top) and distribution of choices in top 5% of configurations (bottom).

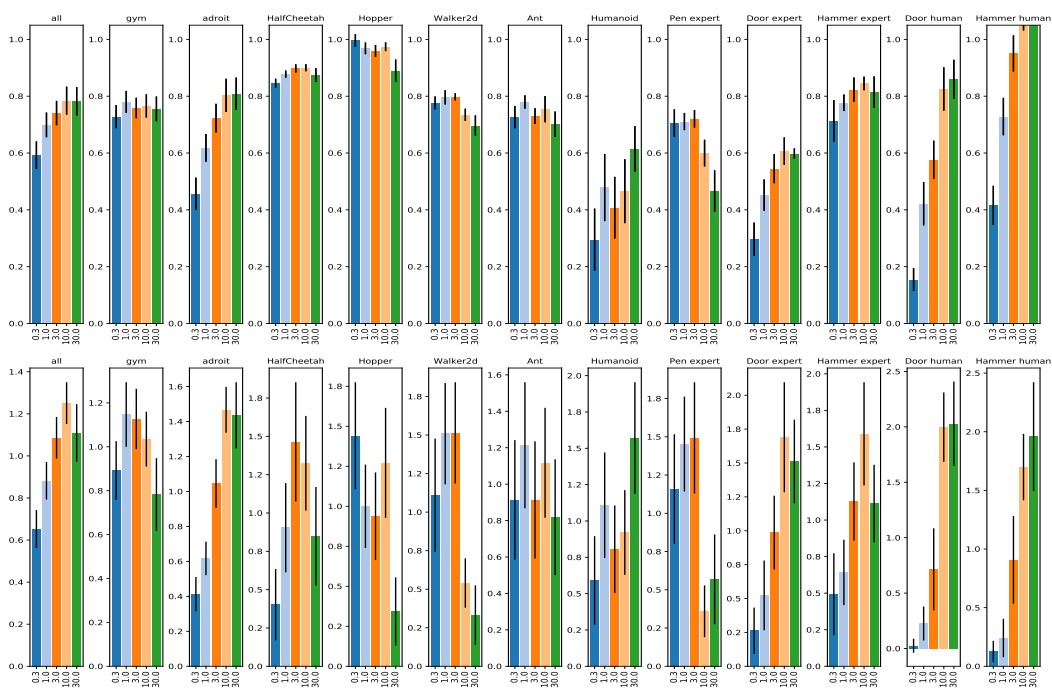

Figure 61: Analysis of choice `weight decay` $\lambda$ (C53): 95th percentile of performance scores conditioned on choice (top) and distribution of choices in top 5% of configurations (bottom).

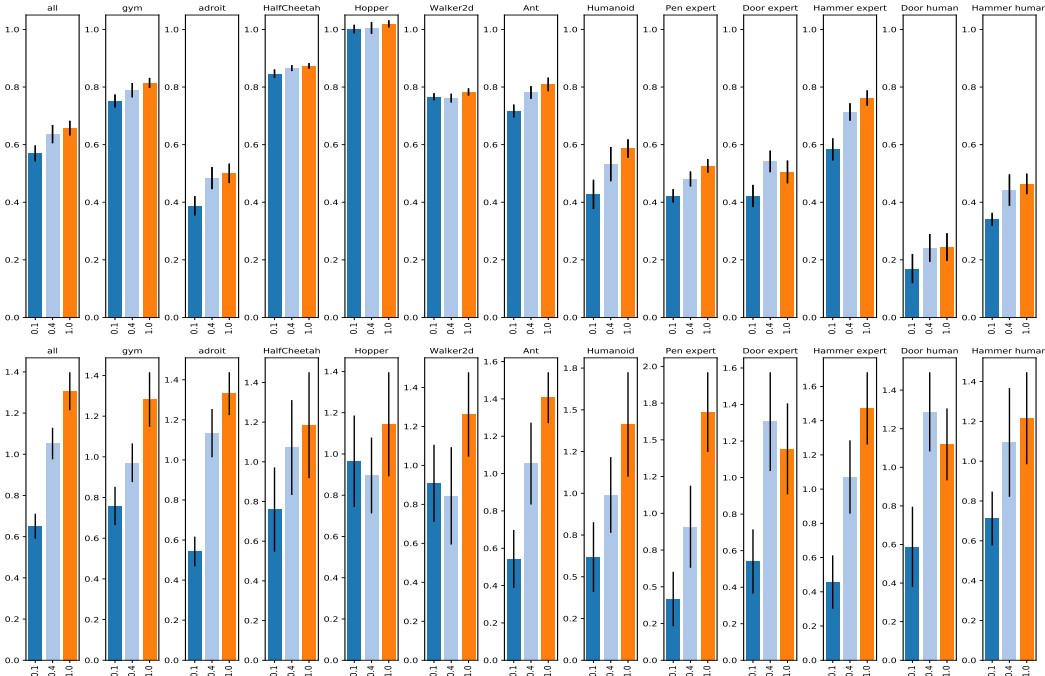

Figure 62: Analysis of choice `mixup` $\alpha$ (C48): 95th percentile of performance scores conditioned on choice (top) and distribution of choices in top 5% of configurations (bottom).

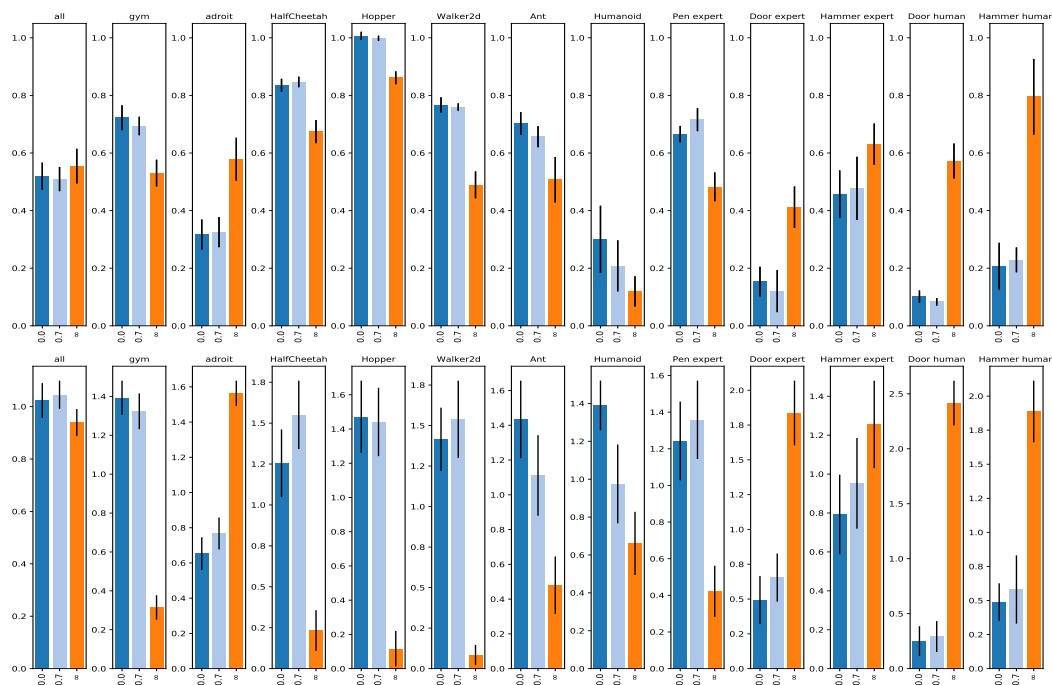

Figure 63: Analysis of choice `PUGAIL` $\beta$ `(C50)`: 95th percentile of performance scores conditioned on choice (top) and distribution of choices in top 5% of configurations (bottom).

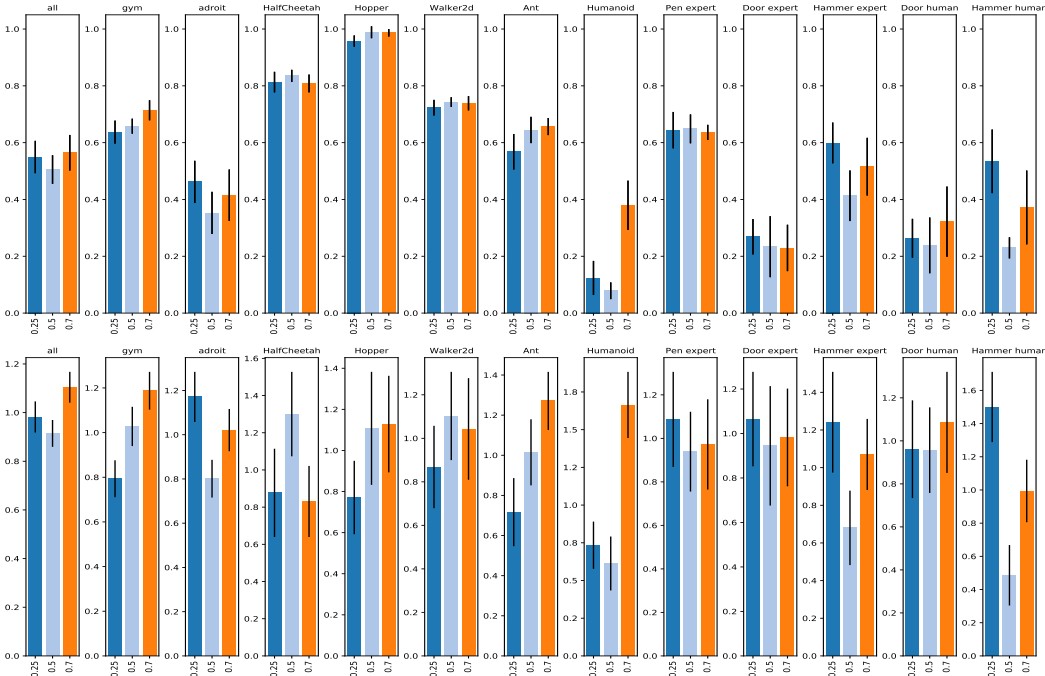

Figure 64: Analysis of choice `PUGAIL` $\eta$ `(C49)`: 95th percentile of performance scores conditioned on choice (top) and distribution of choices in top 5% of configurations (bottom).

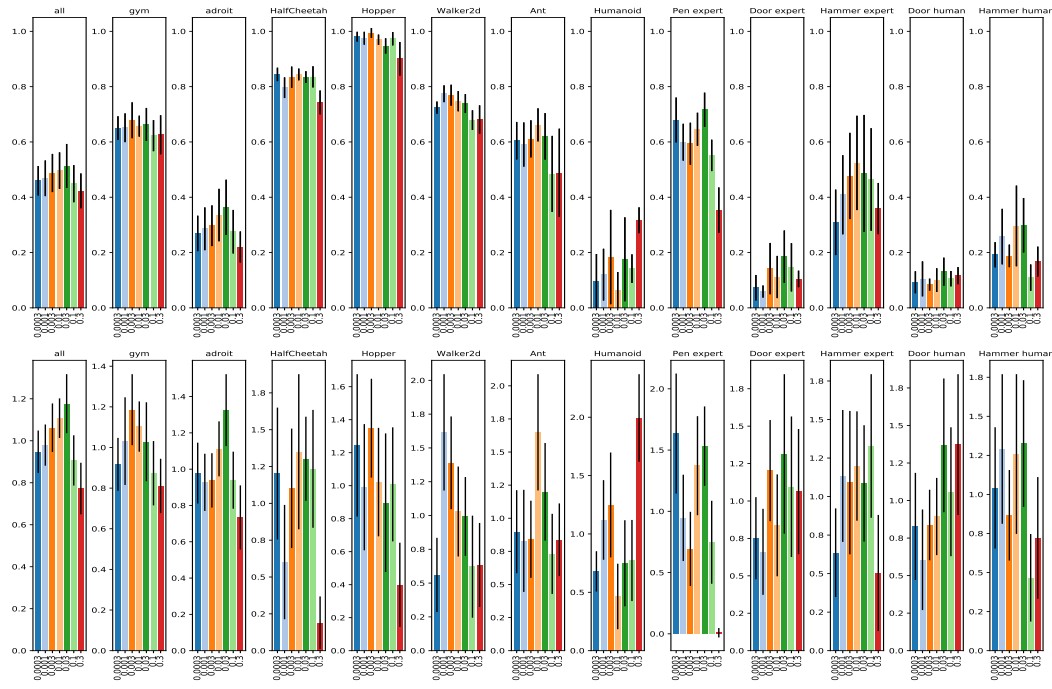

Figure 65: Analysis of choice `entropy` $\lambda$ (C54): 95th percentile of performance scores conditioned on choice (top) and distribution of choices in top 5% of configurations (bottom).

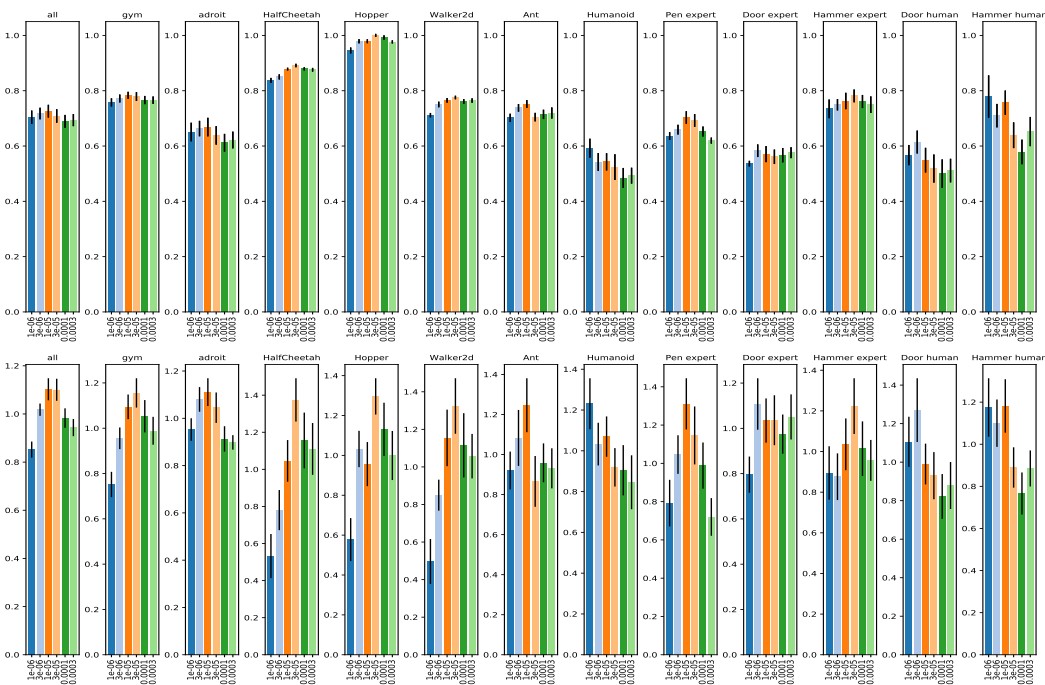

Figure 66: Analysis of choice `discriminator learning rate` (C42): 95th percentile of performance scores conditioned on choice (top) and distribution of choices in top 5% of configurations (bottom).

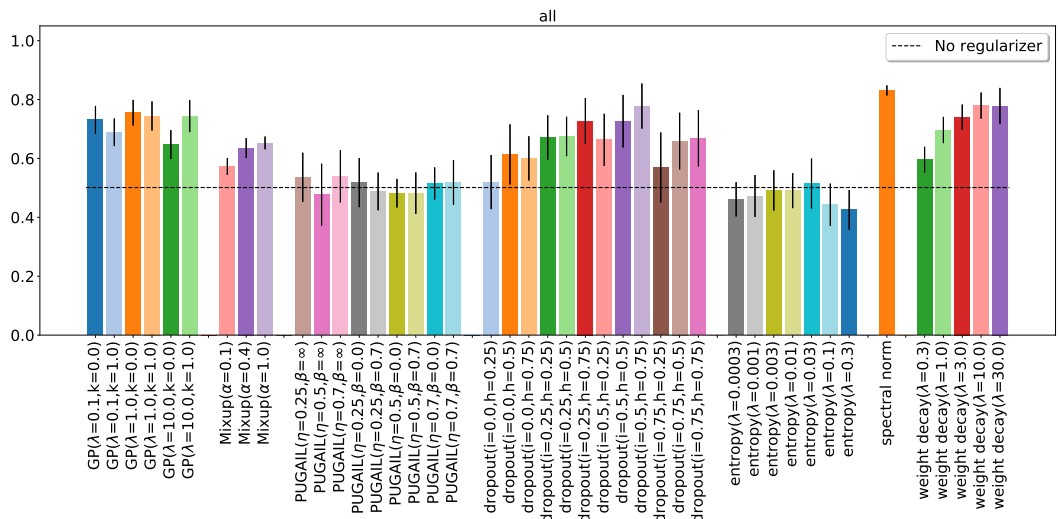

Figure 67: 95th percentile of performance scores conditioned on `discriminator regularizer` (C45) and regularizers' HPs averaged across all environments.

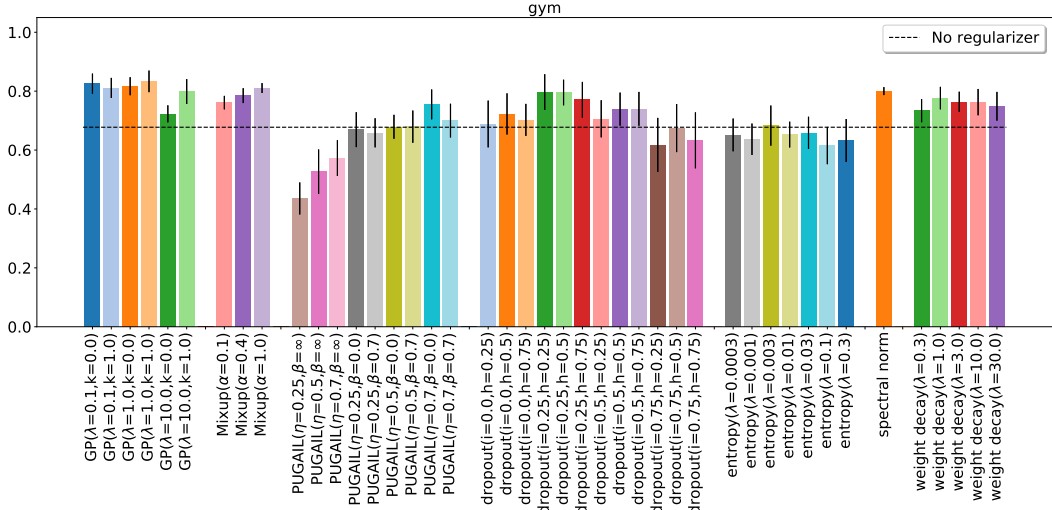

Figure 68: 95th percentile of performance scores conditioned on `discriminator regularizer` (C45) and regularizers' HPs averaged across OpenAI Gym environments.

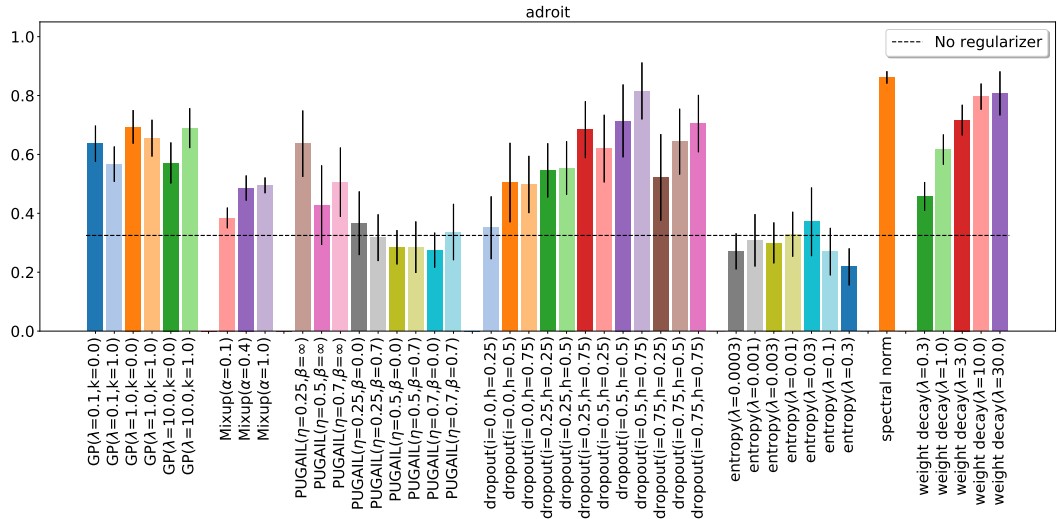

Figure 69: 95th percentile of performance scores conditioned on `discriminator regularizer` (`C45`) and regularizers' HPs averaged across Adroit environments.

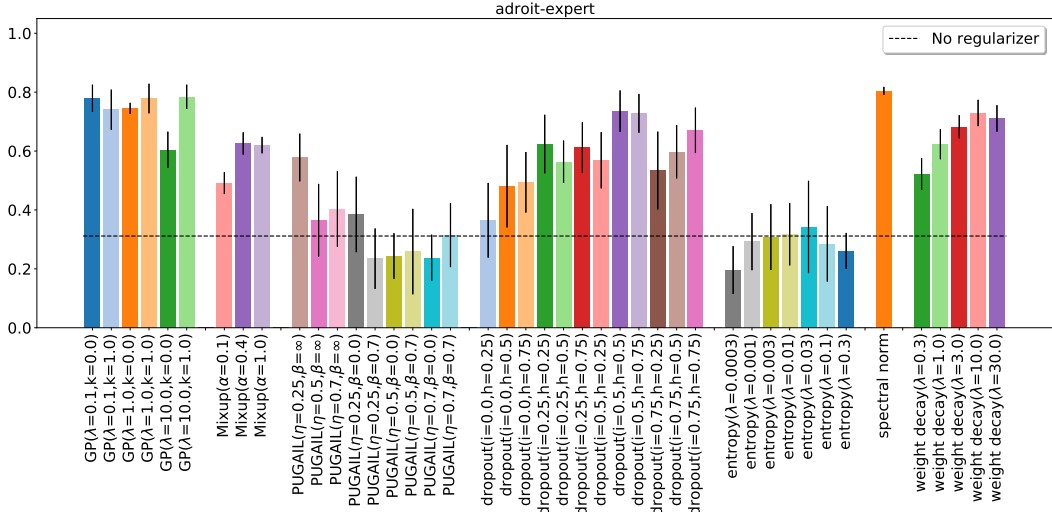

Figure 70: 95th percentile of performance scores conditioned on `discriminator regularizer` (`C45`) and regularizers' HPs averaged across `door-expert` and `hammer-expert` tasks.

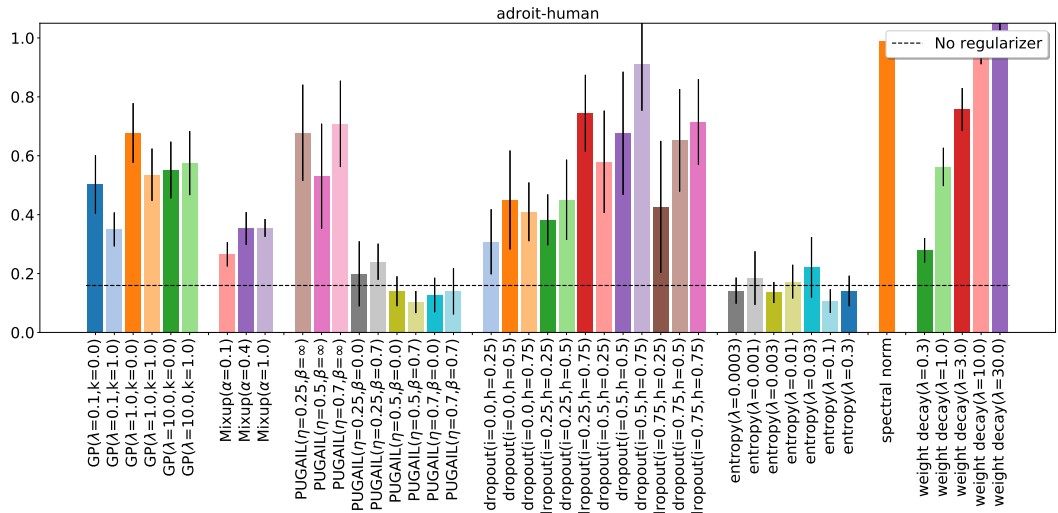

Figure 71: 95th percentile of performance scores conditioned on `discriminator regularizer` (`C45`) and regularizers' HPs averaged across `door-human` and `hammer-human` tasks.

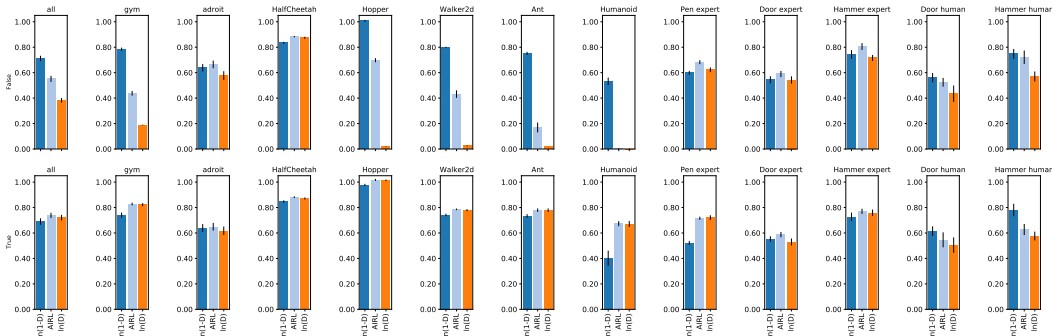

Figure 72: 95th percentile of performance scores conditioned on `absorbing state (C32)`(rows) and `reward function (C30)`(bars).

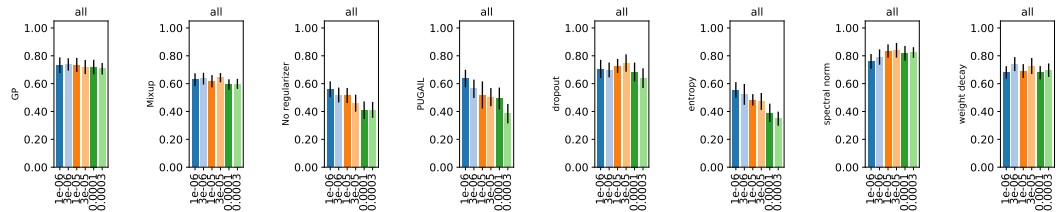

Figure 73: 95th percentile of performance scores conditioned on `discriminator regularizer (C45)`(subplots) and `discriminator learning rate (C42)`(bars).

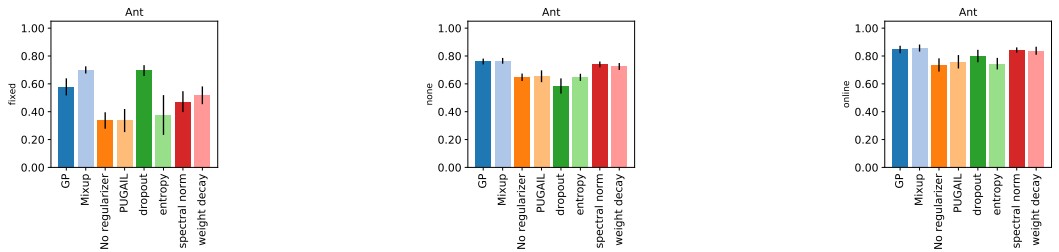

Figure 74: 95th percentile of performance scores conditioned on `observation normalization (C55)`(subplots) and `discriminator regularizer (C45)`(bars) in the `Ant` environment.

# I Experiment trade-offs

## I.1 Design

For each of the 10 tasks, we sampled 7991 choice configurations where we sampled the following choices independently and uniformly from the following ranges:

- `RL Algorithm (C8)`: {d4pg, sac, td3}
  - For the case "`RL Algorithm (C8)` = sac", we further sampled the sub-choices:
    * `SAC learning rate (C17)`: {0.0001, 0.0003, 0.001}
    * `SAC entropy per dimension (C16)`: {-2.0, -1.0, -0.5, 0.0}
    * `SAC polyak` $\tau$ `(C18)`: {0.001, 0.003, 0.01, 0.03}
  - For the case "`RL Algorithm (C8)` = d4pg", we further sampled the sub-choices:
    * `D4PG learning rate (C26)`: {3e-05, 0.0001, 0.0003}
    * `behavioral policy noise (C21)`: {0.1, 0.2, 0.3, 0.5}
    * `VMax (C24)`: {150.0, 750.0, 1500.0}
    * `number of atoms (C23)`: {51.0, 101.0, 201.0, 401.0}
    * `N-step returns (C25)`: {1.0, 3.0, 5.0}
  - For the case "`RL Algorithm (C8)` = td3", we further sampled the sub-choices:
    * `TD3 policy learning rate (C19)`: {0.0001, 0.0003, 0.001}
    * `TD3 critic learning rate (C20)`: {0.0001, 0.0003, 0.001}
    * `TD3 gradient clipping (C22)`: {40.0, $\infty$}
    * `behavioral policy noise (C21)`: {0.1, 0.2, 0.3, 0.5}
- `RL replay buffer size (C28)`: {300000, 1000000, 3000000}
- `policy MLP depth (C1)`: {1, 2, 3}
- `policy MLP width (C2)`: {64, 128, 256, 512}
- `critic MLP depth (C3)`: {2, 3}
- `critic MLP width (C4)`: {256, 512}
- `RL activation (C5)`: {relu, tanh}
- `discount` $\gamma$ `(C6)`: {0.97, 0.99}
- `BC pretraining (C34)`: {False, True}
- `absorbing state (C32)`: {False, True}
- `discriminator replay buffer size (C43)`: {300000, 1000000, 3000000}
- `reward shaping (C39)`: {False, True}
- `discriminator input (C35)`: {s, sa, sas, ss}
- `discriminator MLP depth (C36)`: {1, 2, 3}
- `discriminator MLP width (C37)`: {16, 32, 64, 128, 256, 512}
- `discriminator activation (C38)`: {elu, leaky_relu, relu, sigmoid, swish, tanh}
- `discriminator last layer init scale (C41)`: {0.001, 1.0}
- `discriminator regularizer (C45)`: {GP, Mixup, No regularizer, PUGAIL, dropout, entropy, spectral norm, weight decay}
  - For the case "`discriminator regularizer (C45)` = GP", we further sampled the sub-choices:
    * `gradient penalty` $\lambda$ `(C47)`: {0.1, 1.0, 10.0}
    * `gradient penalty k (C46)`: {0.0, 1.0}
  - For the case "`discriminator regularizer (C45)` = Mixup", we further sampled the sub-choices:
    * `mixup` $\alpha$ `(C48)`: {0.1, 0.4, 1.0}
  - For the case "`discriminator regularizer (C45)` = PUGAIL", we further sampled the sub-choices:

      ∗ `PUGAIL` $\eta$ `(C49)`: $\{0.25, 0.5, 0.7\}$

      ∗ `PUGAIL` $\beta$ `(C50)`: $\{0.0, 0.7, \infty\}$

– For the case "`discriminator regularizer (C45)` = entropy", we further sampled the sub-choices:

      ∗ `entropy` $\lambda$ `(C54)`: $\{0.0003, 0.001, 0.003, 0.01, 0.03, 0.1, 0.3\}$

– For the case "`discriminator regularizer (C45)` = weight decay", we further sampled the sub-choices:

      ∗ `weight decay` $\lambda$ `(C53)`: $\{0.3, 1.0, 3.0, 10.0, 30.0\}$

– For the case "`discriminator regularizer (C45)` = dropout", we further sampled the sub-choices:

      ∗ `dropout input rate (C52)`: $\{0.0, 0.25, 0.5, 0.75\}$

      ∗ `dropout hidden rate (C51)`: $\{0.25, 0.5, 0.75\}$

- `observation normalization (C55)`: {fixed, none}

- `evaluation behavior policy type (C29)`: {average, mode, stochastic}

- `discriminator learning rate (C42)`: {1e-06, 3e-06, 1e-05, 3e-05, 0.0001, 0.0003}

- `replay ratio (C27)`: $\{64, 128, 256, 512, 1024\}$

- `batch size (C7)`: $\{64, 128, 256, 512, 1024\}$

- `discriminator to RL updates ratio (C44)`: $\{1, 2\}$

- `number of combined batches (C56)`: $\{1, 2, 4, 8, 16, 32, 64\}$

- `reward function (C30)`: {-ln(1-D), AIRL, ln(D)}

## I.2 Results

For each of the sampled choice configurations we compute the performance metric as described in Section 2. We report aggregate statistics of the experiment in Tables 11–14 as well as training curves in Figure 75. We further provide per-choice analyses in Figures 76-79.

Table 11: Quantiles of the *final* agent performance across HP configurations for OpenAI Gym tasks.

|       | Ant  | HalfCheetah | Hopper | Humanoid | Walker2d |
|-------|------|-------------|--------|----------|----------|
| 90%   | 0.81 | 1.04        | 1.18   | 0.14     | 0.97     |
| 95%   | 0.94 | 1.08        | 1.19   | 0.62     | 1.00     |
| 99%   | 1.04 | 1.15        | 1.22   | 0.98     | 1.03     |
| Max   | 1.15 | 1.41        | 1.31   | 1.05     | 1.16     |

Table 12: Quantiles of the *final* agent performance across HP configurations for Adroit tasks.

|       | Door expert | Door human | Hammer expert | Hammer human | Pen expert |
|-------|-------------|------------|---------------|--------------|------------|
| 90%   | 0.71        | 0.25       | 1.03          | 0.45         | 0.70       |
| 95%   | 0.89        | 0.71       | 1.25          | 1.19         | 0.86       |
| 99%   | 1.04        | 2.12       | 1.36          | 2.95         | 1.07       |
| Max   | 1.15        | 3.79       | 1.44          | 5.27         | 1.34       |

Table 13: Quantiles of the *average* agent performance during training across HP configurations for OpenAI Gym tasks.

|  | Ant | HalfCheetah | Hopper | Humanoid | Walker2d |
|---|---|---|---|---|---|
| 90% | 0.48 | 0.75 | 0.90 | 0.12 | 0.63 |
| 95% | 0.63 | 0.83 | 0.98 | 0.32 | 0.72 |
| 99% | 0.77 | 0.92 | 1.06 | 0.62 | 0.83 |
| Max | 0.89 | 1.00 | 1.10 | 0.85 | 0.92 |

Table 14: Quantiles of the *average* agent performance during training across HP configurations for Adroit tasks.

|  | Door expert | Door human | Hammer expert | Hammer human | Pen expert |
|---|---|---|---|---|---|
| 90% | 0.38 | 0.26 | 0.54 | 0.39 | 0.50 |
| 95% | 0.53 | 0.49 | 0.71 | 0.65 | 0.63 |
| 99% | 0.74 | 1.02 | 0.91 | 1.21 | 0.82 |
| Max | 0.94 | 2.05 | 1.17 | 2.13 | 1.01 |

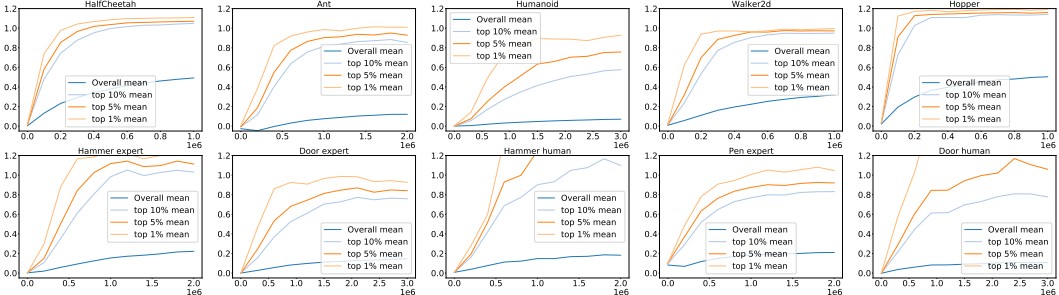

Figure 75: Training curves.

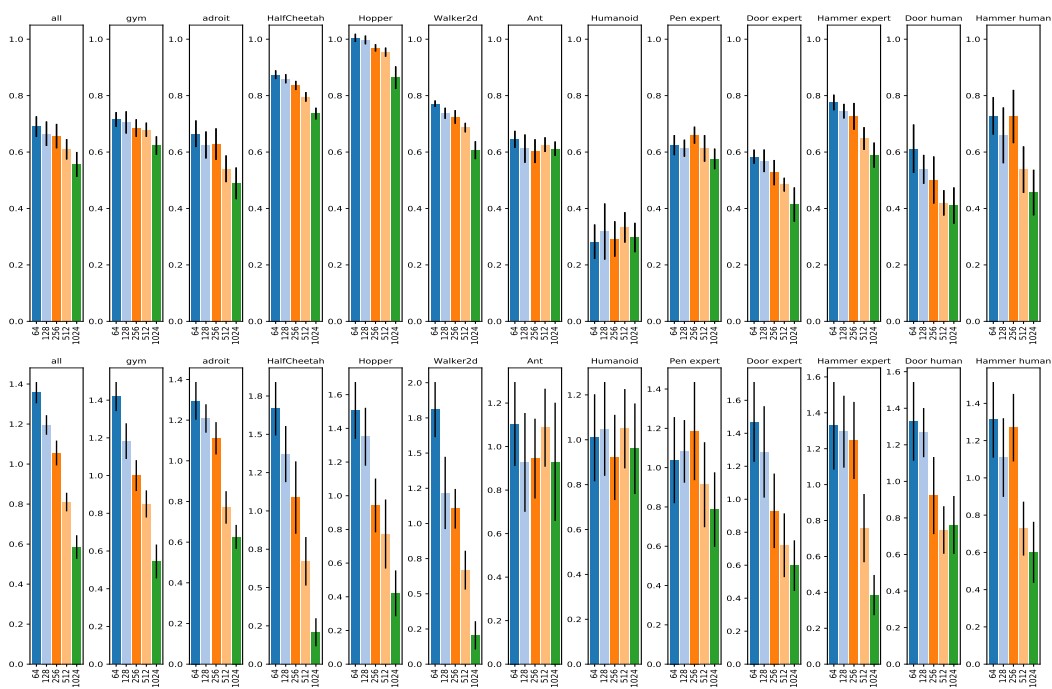

Figure 76: Analysis of choice `batch size` (C7): 95th percentile of performance scores conditioned on choice (top) and distribution of choices in top 5% of configurations (bottom).

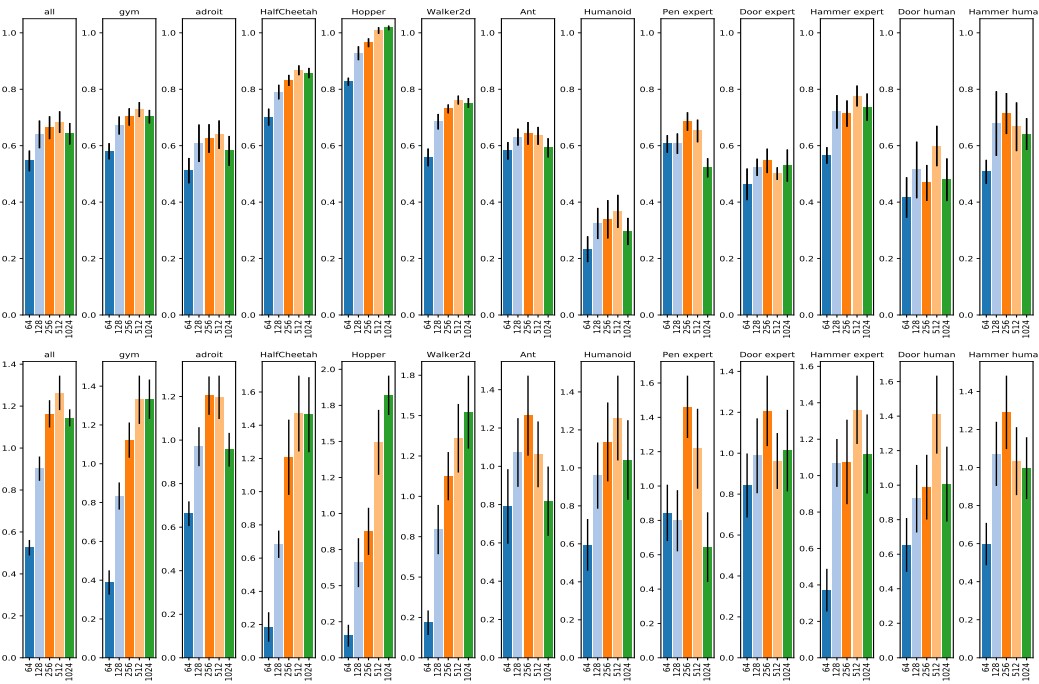

Figure 77: Analysis of choice `replay ratio` (C27): 95th percentile of performance scores conditioned on choice (top) and distribution of choices in top 5% of configurations (bottom).

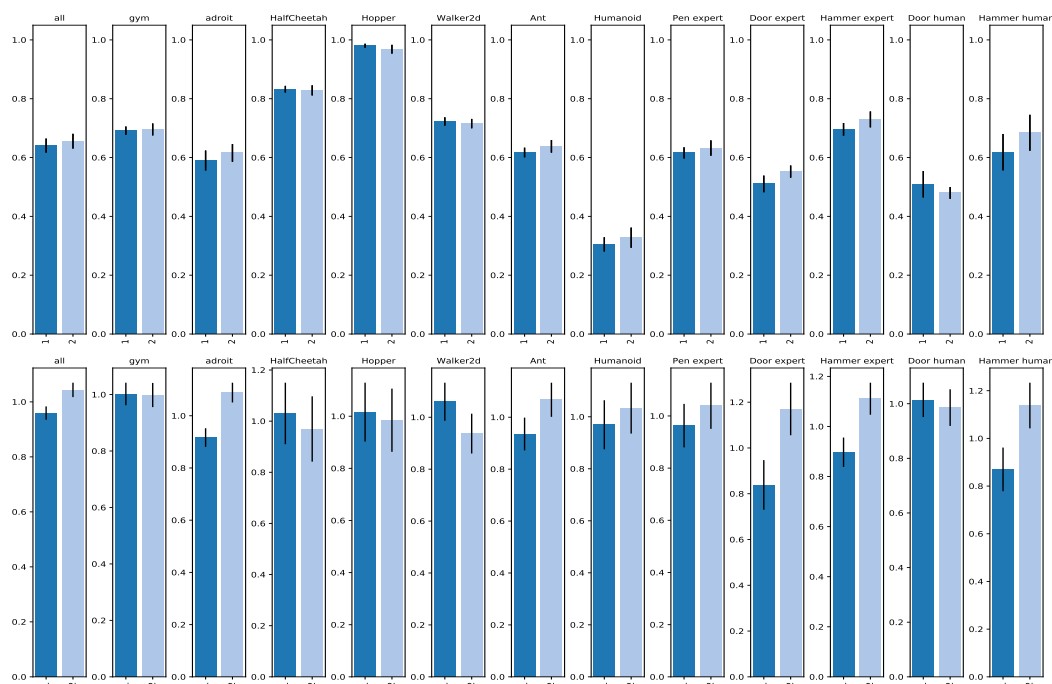

Figure 78: Analysis of choice `discriminator to RL updates ratio (C44)`: 95th percentile of performance scores conditioned on choice (top) and distribution of choices in top 5% of configurations (bottom).

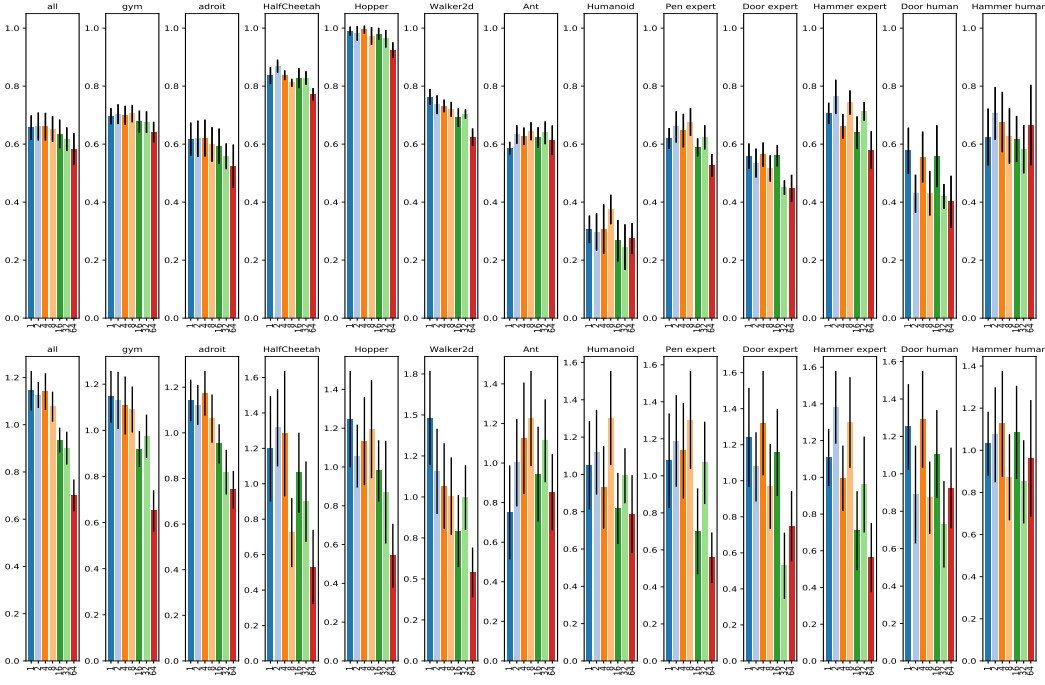

Figure 79: Analysis of choice `number of combined batches (C56)`: 95th percentile of performance scores conditioned on choice (top) and distribution of choices in top 5% of configurations (bottom).



Figure 80: 95th percentile of performance scores conditioned on `batch size (C7)`(subplots) and `replay ratio (C27)`(bars).

# J    Additional experiments

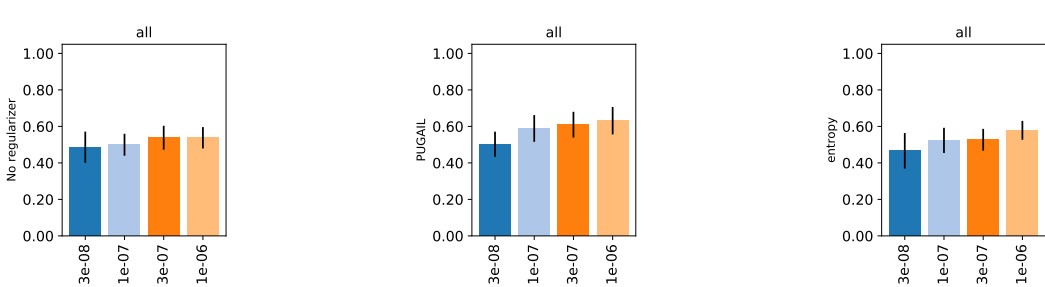

Figure 81: 95th percentile of performance scores conditioned on `discriminator regularizer` `(C45)`(rows) and `discriminator learning rate (C42)`(bars). The data comes from an experiment similar to the main one but with smaller values of `discriminator learning rate (C42)`.

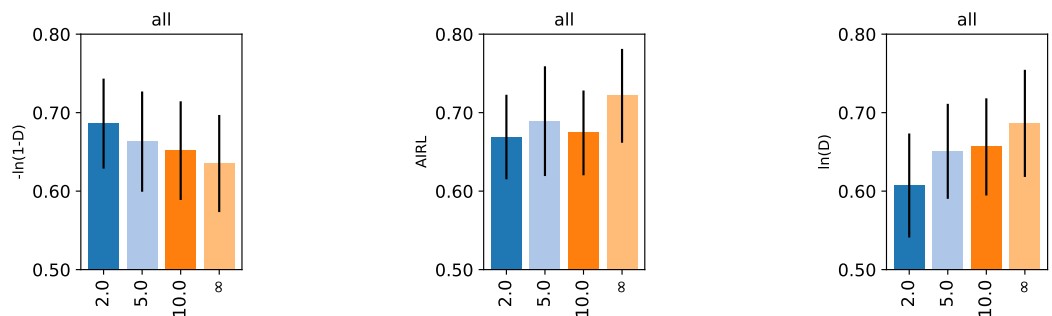

Figure 82:   95th percentile of  performance  scores  conditioned  on  `reward function` `(C30)`(subplots) and `max reward magnitude (C31)`(bars). The data comes from an experiment similar to the main one but with `max reward magnitude (C31)` swept.

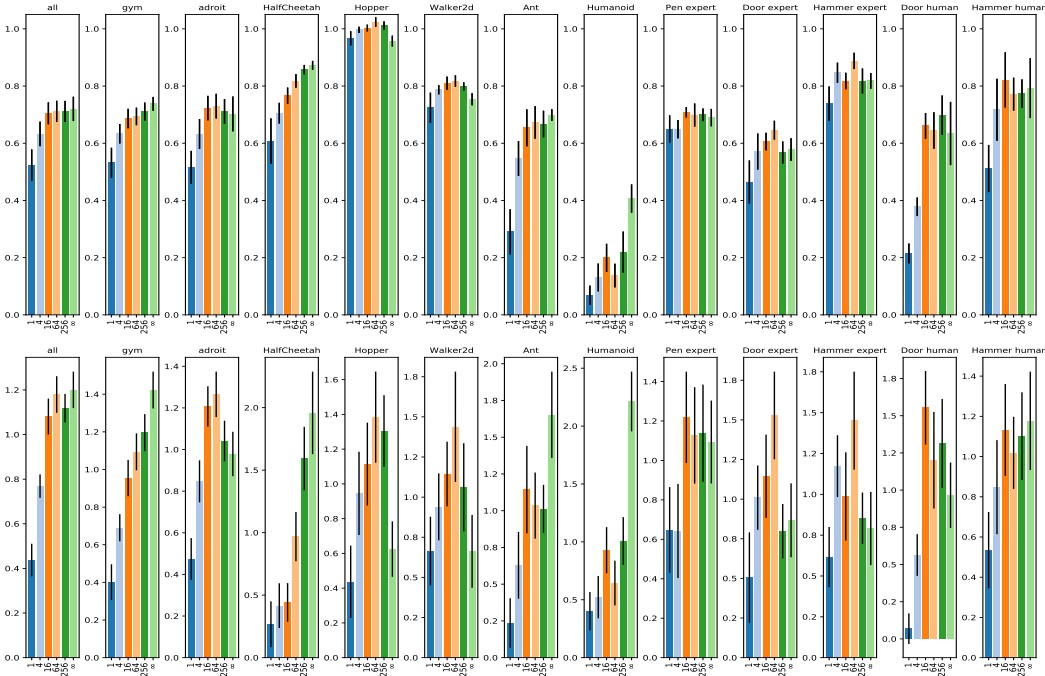

Figure 83: Analysis of choice `policy-to-expert replay ratio` (C33): 95th percentile of performance scores conditioned on choice (top) and distribution of choices in top 5% of configurations (bottom). The data comes from an experiment similar to the main one but in which we also sweep `policy-to-expert replay ratio` (C33). All other experiments do not replay expert data.