# OpenReview forum: "What Matters for Adversarial Imitation Learning?"
_NeurIPS.cc/2021/Conference — NeurIPS 2021 Poster_

### Official Review · Reviewer_YvcC · 2021-07-12

**Rating:** 8
**Confidence:** 4

**Summary:**

The work conducts a detailed empirical analysis of a range of design decisions in the domain of adversarial imitation learning. The imitation learning framework is sufficiently general to encompass prior work like GAIL, AIRL, various extensions to these algorithms (e.g. regularizers), different RL algorithms used for the generator training. These decisions are tested in a comprehensive hyperparameter sweep involving over 500k trained agents across eight continuous control tasks. The results include many observations that will be useful to practitioners, such as the importance of observation normalization. It also in some cases casts doubt on received wisdom, such as the various AIL-specific methods for discriminator regularization, which seem to be little better than standard SL approaches like dropout.

**Limitations And Societal Impact:**

Limitations discussed in context but not adequately pushed through to the summary. No real discussion of societal impact of work which is a missed opportunity but I expect it to be robustly positive.

**Main Review:**

With the proliferation of adversarial imitation learning algorithms and the sadly often not-so-careful evaluation in these papers, testing what actually works is timely and important. The paper is clearly written and the experiments are carefully conducted. I expect practitioners to find this a helpful aid to tuning their imitation learning algorithm. For those active in AIL research, this paper serves both as a useful wake-up call (with, in effect, several negative results contradicting prior work) and as a guide to areas where further research is needed (e.g. more human data benchmarks).

Although I am overall excited by this paper, I did find some of the experiments hard to interpret and potentially confounded. In particular:
  1. The performance measure that approximates area under the learning curve differs from the metric used in much prior work, of episode return at convergence. I think both of these metrics could be appropriate depending on the context. Perhaps you could also report episode return at the end of the training, at least for the experiments where this most significantly changes results? In particular, this would allow the reader to establish whether seemingly negative results about prior work are due to the differing performance measure (perhaps prior methods are better at convergence but worse during early training) or a true replication failure.
  2. I am glad that you highlighted the issue that variable-length episodes cause in evaluation, an issue that has plagued this field. I was also pleased to see that inclusion of the absorbing state does seem to help with this -- at least the difference between reward function choice in Figure 72 becomes a lot less significant with an absorbing state. But how confident can we be the absorbing state has really de-biased this? I would have favored removing the termination condition from these environments, as discussed in Appendix A of [Christiano et al (2017)](https://arxiv.org/pdf/1706.03741.pdf) and followed in much of the other work in this area. I understand re-running the experiments may not be practical -- perhaps consider a break-down of environments with fixed-length episodes vs variable-horizon episodes given your test suite contains a mix of both?
  3. There are often correlations between the hyperparameters. I think the paper mostly does a good job highlighting these limitations in context, but these caveats have tended to be dropped in summaries. In particular, I think the conclusion that synthetic demos may not predict human demos is significantly undermined by the human demos in question being quite strange, in the sense that the agent *cannot* possibly replicate the human demos. I think you either need to argue that this is a normal or unavoidable situation when working with human data, or modify the environment to allow the agent to transition to an absorbing state early, or at least add some significant hedging to the conclusion.

I also have one question: you write that “Following prior work [14, 31, 46], we subsample expert demonstrations by only using every 20th state-action pair to make the tasks harder.” However, I do not see any mention of this in the GAIL paper [14]. Can you refer me to where in [14] this is discussed?

As said, the paper was generally well written, but I noticed a few areas that were confusing:
- Line 29: “and never these changes were all compared together” – ungrammatical. Perhaps “and these changes have never been compared simultaneously”?
- Line 30: “the low-level”->”low-level”. “might be not even” -> “might not even be”.
- Line 220: “As noticed in prior work” – this made me expect you were going to confirm that result but you actually find it doesn't deplicate, consider rephrasing? E.g. “Prior work has claimed the initialization ...”
- Line 223: “An overfitting” -> “An overfit”?
- Some capitalization issues in references, e.g. "Openai gym"->"OpenAI Gym".

# Update after rebuttal

I thank the authors for their response. It is nice to know the results mostly do not change between final performance rather than AUC, but I agree that including a brief analysis of this in the paper would be worthwhile.

I continue to have some concerns about the results being confounded by variable-length episodes and the quirks in the human data, but I do sympathize that given the scale of the experiments and the already significant length of the paper, these issues are difficult to correct. I'd suggest the authors consider adding a more explicit discussion of these limitations to the manuscript, however.

**Time Spent Reviewing:**

2.5

---

> ### Author Response · Authors · 2021-08-10
> **Response to Reviewer YvcC**
>
> We thank the reviewer for the feedback. We are happy they found the study **“comprehensive”**, **“detailed”** and **“useful”**. We are pleased that they think **“the presentation of the results is good and the paper is neatly structured”**, given the size of the study. We address their concerns and suggestions below.
>
> *“1. The performance measure that approximates area under the learning curve differs from the metric used in much prior work, of episode return at convergence. I think both of these metrics could be appropriate depending on the context.”*
>
> We thank the reviewer for their important comment. We indeed computed all our results for both the area under curve as well as the final score. We decided, for the study to remain readable, to include only the former. However, we agree with the reviewer that some interesting findings could be developed in an additional paragraph. When looking at the final score rather than the area under curve (AUC), here are the noticeable differences.
>
> In order to maximize final performance rather than AUC, one should:
>
> - Decrease all learning rates by an order of magnitude (direct RL agents’ ones as well as discriminator’s).
>
> - Regularize the discriminator harder, e.g., use smaller layers, higher weight decay.
>
> - Use TD3 rather than SAC.
>
> - Don’t use observation normalization.
>
> Although the first two findings were expected, they confirm the intuition that regularization induces a trade-off between learning speed and final performance. We find the last two findings more surprising, in particular the one showing that observation normalization is not desirable to maximize final performance for almost all environments.
> We will add these findings in an additional paragraph in the main text of the paper as well as corresponding plots in the appendix. We thank the reviewer for bringing this up.
>
>
> *“2. I understand re-running the experiments may not be practical -- perhaps consider a break-down of environments with fixed-length episodes vs variable-horizon episodes given your test suite contains a mix of both?”*
>
> We thank the reviewer for noticing that we highlighted this issue of variable length episodes and tried to de-bias the experimental results (typically in Fig.1). Although we agree that this break-down would be useful and interesting, we think it would add an additional complexity to this large study. However, we are in the process of open-sourcing the result data as well as the notebook we use for analysis. Any interested reader should be able to re-run the analysis (without running the full experiment), breaking down the results for different sets of environments.
>
> *“3. [...] I think you either need to argue that this is a normal or unavoidable situation when working with human data, or modify the environment to allow the agent to transition to an absorbing state early, or at least add some significant hedging to the conclusion.”*
>
> We agree with your conclusions and advocate strongly for more human-data benchmarks. We will add argumentation to the main text.
>
> *“Can you refer me to where in [14] this is discussed?”*
>
> It is in Sec.6 (Experiments): "The trajectories constituting each dataset each consisted of about 50 state-action pairs."
>
> OpenAI Gym trajectories are of size 1000 (at least successful ones), so 1000/20=50. We definitely agree with the fact that this information is suspiciously hidden. It is explicitly said in the [DAC] paper: “We replicate the experimental setup of Ho & Ermon (2016): expert trajectories are sub-sampled by retaining every 20 time steps starting with a random offset (and fixed stride)”.
>
> We thank the reviewer for their feedback on the text. We took it into consideration in an updated version of the manuscript.
>
> [DAC]: Kostrikov, Ilya, et al. "Discriminator-actor-critic: Addressing sample inefficiency and reward bias in adversarial imitation learning." arXiv preprint arXiv:1809.02925 (2018).

---

### Official Review · Reviewer_mnBs · 2021-07-16

**Rating:** 6
**Confidence:** 5

**Summary:**

The paper presents an extensive study on hyperparameters and design choices for adversarial imitation learning, including the choice of divergence (reward function), RL algorithm and whether marginals $p(s,a)$, $p(s)$, $p(s,s')$ or $p(s,a,s')$ are to be matched. Evaluations are conducted on MuJoCo and Adroit environments --- the latter experiments also made use of human demonstrations.
The main findings are that demonstrations collected by an RL agent are not good proxies for human demonstrations and that more standard discriminator regularizers like dropout perform similarly well as gradient penalty.


**Ethical Concerns:**

no ethical concerns

**Limitations And Societal Impact:**

Limitations and potential negative societal impact were adequately addressed.

**Main Review:**


Significance
----------------
Studies like these can be very helpful for researchers and practitioners. The paper was interesting to read, but afterall I did not get any major new insights. The fact that the differences between human demonstrations and demonstrations from an RL agent affect the performance of the algorithm and the choice of parameters was to be exected. But of course, learning about which parameters might be preferable in a human-demonstration setting is still useful, even though I think that the number of experiments is too small to conclude that JS divergence performs better than KL for example.

Originality
--------------
I don't think that a similar study on adversarial imitation learning exists, so in that sense the work is original. However, it does not present any intriguing ideas and also the interpretations of the results seem a bit shallow.

Quality
-----------
Overall, the study was soundly conducted. However, I do have a few criticisms:

* All experiments seem to use a fixed number of demonstrations. That would be a major oversight, given that some algorithms that work well for many demonstrations (for example behavioral cloning) might not work very well for few demonstrations. Evaluating how design choices relate to the number of demonstrations would have been very interesting. On a minor note, also evaluations w.r.t. to noisy / mixed data might have produced more meaningful insights.

* The performance is evaluated based on the agent's return. I think it would be important to also evaluate the performance w.r.t. the different divergences, since these are the actual objectives of the algorithms. It is not at all clear, whether lower divergences always achieve higher rewards.

* The authors say that they plan to release the code, however, I am not overly optimistic since not even the reviewers have been given access to the code. Also, I would expect a study to publish the underlying data, which would be very useful, for example, to conduct the missing evaluations w.r.t. the different divergences (maybe even MMD).

Clarity
--------
The presentation of the results is good and the paper is neatly structured.

**Time Spent Reviewing:**

4

---

> ### Author Response · Authors · 2021-08-10
> **Response to Reviewer mnBs**
>
> We thank the reviewer for the feedback. We are happy they found our work **“interesting to read”** and that they acknowledge it **“can be very helpful for researchers and practitioners”**. We are pleased that they think **“the presentation of the results is good and the paper is neatly structured”**, given the size of the study. We address their concerns and suggestions below.
>
> *“The fact that the differences between human demonstrations and demonstrations from an RL agent affect the performance of the algorithm and the choice of parameters was to be expected”*
>
> We argue that, although this could have been suspected, it was more "folklore" than grounded knowledge and we could not find rigorous references defending it. We thus propose solid experimental results to confirm it. We argue it is highly relevant as the research community continues to draw conclusions from experiments which use almost entirely synthetic demonstrations and we hope that our results will encourage more work on human demonstrations.
>
> *“I think that the number of experiments is too small to conclude that JS divergence performs better than KL for example.”*
>
> We would like to remind the reviewer that over 500k agents were trained and that, although only 2 task/dataset pairs were considered for human demonstrations, the results we present (with error bars) are statistically significant. We agree that generalizing these results to other environments will require additional studies but we claim this is a first step towards understanding more about how to imitate humans. This study provides a basis, in terms of methodology as well as initial results, for further comparisons.
> Note we provided hypotheses on why JS would perform better than KL, (l.332-333 & 345-347) to help the reader build an intuition and an understanding of these results.
>
> *“Evaluating how design choices relate to the number of demonstrations would have been very interesting.”*
>
> We agree this is interesting. We actually ran the experiments for a single subsampled trajectory as demonstrations (so 50 state-action pairs), instead of 11 trajectories. This is one of the “hardest” setup proposed in previous work.
> Most conclusions were unchanged, we thus decided not to include the results in the paper out of conciseness. The main conclusions we could extract from the comparison of these two experiments are the following:
>
> - On Gym Mujoco results remain unchanged in terms of overall performance as well as of relative importance of components/hyperparameters.
>
> - On Adroit environments, on the other hand, performance suffers from going down to a single demonstration. The overall performance of the algorithm on these environments decreases, yet, the relative importance of the different components/hyperparameters remains mainly unchanged.
>
> To be more precise, when using a single demonstration, 95th percentile average performance over training drops by 15% on Gym Mujoco environments while it drops by 51% on Adroit environments.
>
> The few noticeable differences for the Adroit environments are the following:
>
> - Observation normalization using demonstration is less performant when using a single demonstration. This could be expected as the statistics (mean, std) computed on 11 trajectories were naturally more precise.
>
> - The reward function $-ln(1-D)$ is, relatively to others, more performant when a single demonstration is present.
>
> - When using a single demonstration, Mixup regularization is on par with the best regularizers (dropout, spectral norm), while it was not when given 11 trajectories.
>
> We will add these conclusions to the paper if the reviewer considers them valuable.
>
> *“The performance is evaluated based on the agent's return. I think it would be important to also evaluate the performance w.r.t. the different divergences”*
>
> This point is very important but we argue it is not as easy as the reviewer states. Defining a distance between states in an MDP is a complicated problem [1,2,3]. Defining a similarity between “behaviors” is even harder. Using, for example, the L2-Wasserstein is still a proxy for how “similar” two behaviors are. Somehow, the return is also a proxy for the behavior.
>
> Recent work [4] discussed this problem and showed correlations between these metrics. Overall we agree it is not the perfect performance score but we kindly remind the reviewers that what we propose here is an extensive benchmark of previously developed methods that were all compared with respect to this performance metric.
> We will add a discussion on this topic in the main text. We thank the reviewer for bringing it up.
>
> *“The authors say that they plan to release the code, however, I am not overly optimistic since not even the reviewers have been given access to the code.”*
>
> We are in the process of open-sourcing the code. We would like the code to be usable not only to reproduce our experiments but to run additional experiments on top of it. This requires some work but we are engaged in making this possible, for the sake of reproducibility. We also are open-sourcing both the demonstrations we used and the notebook we use to analyse the results.
>
>
>
> [1] Mahadevan, S. and Maggioni, M. Proto-value functions:
> A laplacian framework for learning representation and
> control in markov decision processes. Journal of Machine
> Learning Research, 2007
>
> [2] Castro, P. S. Scalable methods for computing state similarity
> in deterministic markov decision processes. In AAAI
> Conference on Artificial Intelligence, 2020
>
> [3] Ferns, N., Panangaden, P., and Precup, D. Metrics for finite
> markov decision processes. In Uncertainty in Artificial
> Intelligence (UAI), 2004
>
> [4] Hussenot L, Andrychowicz M, Vincent D, Dadashi R, Raichuk A, Ramos S, Momchev N, Girgin S, Marinier R, Stafiniak L, Orsini M. Hyperparameter Selection for Imitation Learning. In International Conference on Machine Learning 2021 Jul 1 (pp. 4511-4522). PMLR.

---

> > ### Comment · Reviewer_mnBs · 2021-08-24
> > **Thank you for your reply**
> >
> > Thank you for your reply. I stick to my initial assessment that the paper is above the acceptance threshold.
> >
> > However, I have one further question. You said that you will share the code, python notebook and the demonstrations. Would it also be possible to share the learned data (policies). I think this would be very useful to further analyze the results, because it will be difficult for many researchers to rerun the experiments due to computational budgets.

---

> > > ### Author Response · Authors · 2021-08-27
> > > **Thanks for the reply**
> > >
> > > Thanks again for answering our rebuttal.
> > > Releasing the policies would mean sharing over 350GBs of data, which is quite complicated. However, along with the notebook, we will share the raw result data of the experiments, which will allow users to analyze them in very different ways than we did if they want to. We hope this to already be very useful.

---

### Official Review · Reviewer_UNYV · 2021-07-17

**Rating:** 6
**Confidence:** 2

**Summary:**

This paper presents a large-scale empirical study for implementation choices on adversarial imitation learning. The contributions of this work can be summarized as follows:

a. implement a high-configurable adversarial imitation learning algorithm and conduct a large-scale empirical study

b. from the empirical study, the authors argue that they find two important results:

1. a choice of discriminator regularizers is not critical

2. artificial demonstrations are not a good proxy for human demonstrations

**Limitations And Societal Impact:**

Yes, the authors have adequately addressed the limitations and potential negative societal impact of their work.

**Main Review:**

This paper presents interesting results about algorithmic choices on adversarial imitation learning by conducting rigorous large-scale empirical studies. The paper is clearly written and findings, such as discriminator regularization techniques (e.g. gradient penalty, .. ) are not as crucial as we considered in literature, are interesting. Yet, I have following concerns and comments about this paper:

1. The authors argue that synthetic demonstrations are not a good proxy for human data.
But I concern this argument is too obvious to be one of main findings of this empirical study.

2. The authors may argue that unexpected human behaviors make significant differences in optimal hyperparameter settings in Figure 4. But it is not clear to me where those differences come from. Isn’t it simply because of the difference in performance? According to Table 2 in Appendix F, performances of door-expert-v0 and door-human-v0 are significantly different (2882 and 796 respectively), hammer-expert-v0 and hammer-human-v0, too (12794 and 3071 respectively).

3. Hyperparameter selections should be more carefully considered especially for human data. The agent getting higher performance on the evaluation environment is not always better, because the goal of imitation learning is imitating the given expert demonstration.

4. Since expert demonstrations define each problem, varying the quality of expert datasets could be a valuable extension of this empirical study. For example, D4RL dataset contains different versions of each task such as hopper-expert-v0 and hopper-medium-v0, etc.
.

Although I have some concerns on this paper, lessons from this work can be helpful to future imitation learning researchers and practitioners. Hence, I vote for acceptance to this paper.


**Time Spent Reviewing:**

12

---

> ### Author Response · Authors · 2021-08-10
> **Response to Reviewer UNYV**
>
> We thank the reviewer for the feedback. We are happy they found our work **“clearly written”** and presenting **“interesting results”**. We address their concerns and suggestions below.
>
> *“1.The authors argue that synthetic demonstrations are not a good proxy for human data. But I concern this argument is too obvious to be one of main findings of this empirical study.”*
>
>
> We argue that, although this could have been suspected, it was more "folklore" than grounded knowledge. We could not find rigorous references defending it. We thus propose solid experimental results to confirm it. We argue it is highly relevant as the research community continues to draw conclusions from experiments which use almost entirely synthetic demonstrations and we hope that our results will encourage more work on human demonstrations.
>
> *“2.[...] Isn’t it simply because of the difference in performance?”*
>
> Scores are (0,1) normalized using the fed-demonstration performance and the random behavior performance. So the difference in Fig.4 cannot be explained by the differences in the demonstration performance. We insist on the statistical significance of our results given the number of agents that were trained (>500k agents). We state, in the paper two hypothesis as for why such difference:
>
> - "We suspect that it might be caused by the mentioned issue with demonstration lengths which forces the policy to repeat a similar movement but with a different speed than the demonstrator." (l. 332-333)
>
> - “We suspect that this boundedness is beneficial for learning with human demonstrations because it may not be possible to exactly match the human distribution for the reasons explained earlier.” (l. 345-347)
> If the reviewer thinks these paragraphs are not clear enough, we can clarify them in the revision.
>
> *“3. [...] The goal of imitation learning is imitating the given expert demonstration”*
>
> This point is very important. First, we argue that Imitation Learning aims at learning the policy of the demonstrator using the demonstrations, and not only to reproduce the demonstrations with high fidelity. Second, defining a distance between states in an MDP is a complicated problem [1,2,3]. Defining a similarity between “behaviors” is even harder. Using, for example, the L2-Wasserstein is still a proxy for how “similar” two behaviors are. Somehow, the return is also a proxy for the behavior.
>
> Recent work [4] recently discussed this problem and showed correlations between these metrics. Overall we agree it is not the perfect performance score but we kindly remind the reviewers that what we propose here is an extensive benchmark of previously developed methods that were all compared with respect to this performance metric. To give the reviewer more information concerning this matter, we also computed the very same histograms and plots considering the following performance score $1-abs(1-score)$ instead of the normalized score, somehow implementing a distance to expert behavior. This led to similar results and we decided it was an additional degree of complexity that was not worth introducing.
>
> We will add a discussion on this topic in the main text. We thank the reviewer for bringing it up.
>
>
> *“4. Since expert demonstrations define each problem, varying the quality of expert datasets could be a valuable extension of this empirical study.”*
>
> We fully agree with the reviewer that studying how the results we present vary when confronting different demonstrations would be extremely valuable. We kindly remind the reviewer that the empirical study is already quite large (>500k agents) and that varying one more axis would also make it harder to present results in a comprehensive manner (note that we already provide a 53-page appendix with detailed results and already try to distill the most important results within the main text).
> We still wanted to mention that we actually ran the experiments for a single subsampled trajectory as demonstrations (so 50 state-action pairs), instead of 11 trajectories. This is one of the “hardest” setup proposed in previous work.
> Most conclusions were unchanged, we thus decided not to include the results in the paper out of conciseness. The main conclusions we could extract from the comparison of these two experiments are the following.
>
> - On Gym Mujoco results remain unchanged in terms of overall performance as well as of relative importance of components/hyperparameters.
>
> - On Adroit environments, on the other hand, performance suffers from going down to a single demonstration. The overall performance of the algorithm on these environments decreases, yet, the relative importance of the different components/hyperparameters remains mainly unchanged.
>
> To be more precise, when using a single demonstration, 95th percentile average performance over training drops by 15% on Gym Mujoco environments while it drops by 51% on Adroit environments.
>
> The few noticeable differences for the Adroit environments are the following:
>
> - Observation normalization using demonstration is less performant when using a single demonstration. This could be expected as the statistics (mean, std) computed on 11 trajectories were naturally more precise.
>
> - The reward function $-ln(1-D)$ is, relatively to others, more performant when a single demonstration is given.
>
> - When using a single demonstration, Mixup regularization is on par with the best regularizers (dropout, spectral norm), while it was not when given 11 trajectories.
>
> We will add these conclusions to the paper if the reviewer considers them valuable.
>
> [1] Mahadevan, S. and Maggioni, M. Proto-value functions:
> A laplacian framework for learning representation and
> control in markov decision processes. Journal of Machine
> Learning Research, 2007
>
> [2] Castro, P. S. Scalable methods for computing state similarity
> in deterministic markov decision processes. In AAAI
> Conference on Artificial Intelligence, 2020
>
> [3] Ferns, N., Panangaden, P., and Precup, D. Metrics for finite
> markov decision processes. In Uncertainty in Artificial
> Intelligence (UAI), 2004
>
> [4] Hussenot L, Andrychowicz M, Vincent D, Dadashi R, Raichuk A, Ramos S, Momchev N, Girgin S, Marinier R, Stafiniak L, Orsini M., Bachem O, Geist M, Pietquin O. Hyperparameter Selection for Imitation Learning. In International Conference on Machine Learning 2021 Jul 1 (pp. 4511-4522). PMLR.

---

### Author Response · Authors · 2021-08-10
**General Comment**

We thank all three reviewers for their comments. We are pleased to hear that they find this empirical study valuable and well structured. We answer below, in detail, each of the reviews specifically.

---

### Decision · Program_Chairs · 2021-09-27

**Decision:**

Accept (Poster)

**Comment:**

This paper presents an extensive experimental study on implementation choices in adversarial imitation learning. All reviewers were leaning towards acceptance of the paper since many observations made in the paper could be useful to practitioners. I am also recommending acceptance of the paper.